**Simulated Hydrologic Response to Projected Changes in Precipitation and**
**Temperature in the Congo River Basin**
Noel Aloysius[1,2] and James Saiers[1]
[1] School of Forestry and Environmental Studies, Yale University, New Haven,
Connecticut, USA
[2] now at Department of Food, Agriculture & Biological Engineering and

7         Aquatic Ecology Laboratory, Department of Evolution, Ecology and Organismal

8         Biology

9         Ohio State University, Columbus, Ohio, USA

**Abstract**
Despite their global significance, the impacts of climate change on water resources and
associated ecosystem services in the Congo River Basin (CRB) have been understudied.
Of particular need for decision makers is the availability of spatial and temporal
variability of runoff projections. Here, with the aid of a spatially explicit hydrological
model forced with precipitation and temperature projections from 25 global climate
models (GCMs) under two greenhouse gas emission scenarios, we explore the variability
in modeled runoff in the near (2016-2035) and mid (2046-2065) century. We find that
total runoff from the CRB is projected to increase by 5% [-9%; 20%] (mean [min and
max] across model ensembles) over the next two decades and by 7% [-12%; 24%] by
midcentury. Projected changes in runoff from sub-watersheds distributed within the CRB
vary in magnitude and sign.  Over the equatorial region and in parts of northern and
southwestern CRB, most models project an overall increase in precipitation and,
subsequently, runoff. A simulated decrease in precipitation leads to a decline in runoff
from head-water regions located in the northeastern and southeastern CRB. Climate
model selection plays an important role in future projections, for both magnitude and
direction of change. The multi-model ensemble approach reveals that precipitation and
runoff changes under business-as-usual and avoided greenhouse gas emission scenarios
(RCP8.5 vs. RCP4.5) are relatively similar in the near-term, but deviate in the mid-term,
which underscores the need for rapid action on climate change adaptation. Our
assessment demonstrates the need to include uncertainties in climate model and emission
scenario selection during decision making processes related to climate change mitigation
and adaptation.

## 1. Introduction

Sustainable management of water resources for food production, supply of safe drinking water, and provision of adequate sanitation presents immense challenges in many countries of Central Africa where the Congo River Basin (CRB) is located [*IPCC*, 2014; *UNEP*, 2011; *World Food Program*, 2014]. The economies of the nine countries that share the waters of the CRB are agriculture-based [*World Bank Group*, 2014] and, therefore, are vulnerable to the impacts of climate change. Despite the abundant water and land resources and favorable climates, the basin countries are net importers of staple food grains and are far behind in achieving Millennium Development Goals [*Bruinsma*, 2003; *Molden*, 2007; *UNEP*, 2011]. Appropriation of freshwater resources is expected to grow in the future as the CRB countries develop and expand their economies. At the same time, climate change related risks associated with water resources will also increase significantly [*IPCC*, 2014].

Historical, present and near-future greenhouse gas emissions in the CRB countries constitute a small fraction of global emissions; however, the impacts of climate change on water resources are expected to be severe due to the region's heavy reliance on natural resources (e.g. agriculture and forestry) [*Collier et al.*, 2008; *DeFries and Rosenzweig*, 2010; *Niang et al.*, 2014]. The limited adaptation capacity in the CRB region is expected to cause water and food security challenges, which, in turn, can lead to ecosystem degradation and increased greenhouse gas emissions [*Gibbs et al.*, 2010; *IPCC*, 2014; *Malhi and Grace*, 2000].

Strategies for addressing stresses on CRB water resources, including revival of

rural economies (largely agriculture based), achieving millennium development goals and
environmental conservation, would benefit from detailed information on the spatial and
temporal variability of water balance components under different climate projection
pathways. The effect of climate change on water resources can be investigated by
incorporating climate change projections (e.g. precipitation and temperature) in
simulation models that reliably represent the spatial and temporal variability of CRB's
hydrology. Such a framework could be applied to project changes in storage and runoff,
and hence freshwater availability, under different socioeconomic pathways that affect
climate trajectories.

A predictive framework of the CRB's hydrology is hindered by insufficient data

and too few evaluations of models against available data [*Beighley et al.*, 2011; *Wohl et*
*al.*, 2012]. Basin scale water budgets estimated from land-based and satellite-derived
precipitation datasets reveal significantly different results, and modeled runoff shows
only qualitative agreement with corresponding observations [*Alsdorf et al.*, 2016;
*Beighley et al.*, 2011; *Lee et al.*, 2011; *Schuol et al.*, 2008]. *Tshimanga and Hughes*
[2012; 2014] recently developed a semi-distributed hydrologic model capable of
simulating runoff in CRB.  This work crucially identified approaches suitable for
approximating runoff generation at the basin scale, although the spatial resolution of the
model predictions is rather coarse for supporting regional water management and
regional-planning efforts.  These regional planning efforts must take into account
variablity and uncertainties stemming from climate-model selection and projected
greenhouse gas emissions, but, with respect to freshwater runoff projections for the CRB,
these issues have been inadequately addressed.

The goals of this study are to i) develop a spatially explicit hydrology model that

uses downscaled output from general circulation models (GCMs) and is suitable for
simulating the spatiotemporal variability of runoff in the CRB; ii) test the ability of the
hydrological model to reproduce historical data on CRB river discharges using both
observed and GCM-simulated climate fields; (iii) quantify the sensitivity and uncertainty
of modeled runoff projections to GCM selection; (iv) use the hydrologic model with
individual GCMs and multi-GCM ensembles to project near-term (2016-2035) and mid-
term (2046-2065) changes in runoff for two greenhouse-gas emission scenarios. We
focus on the runoff projections because streams and rivers will serve as the primary
sources of freshwater targeted for human appropriation [*Burney et al.*, 2013; *Molden*,
2007].

**2. Materials and Methods**
***2.1 The Congo River Basin***

The Congo River Basin, with a drainage area of 3.7 million $km^2$, is the second

largest in the world by area and discharge (Figure 1, average discharge of ~41,000 $m^3s^{-1}$)
[*Runge*, 2007]. The basin extends from 9$^o$N to 14$^o$S, while the longitudinal extent is 11$^o$E
to 35$^o$E. Nine countries share the water resources of the basin. Nearly a third of the basin
area lies north of the equator. Due to its equatorial location, the basin experiences a range
of climate regimes. The northern and southern parts have strong dry and wet seasons,
while the equatorial region has a bimodal rainy season [*Bultot and Griffiths*, 1972]. Much
of the rain in the northern and southern CRB occurs in Jun-Jul-Aug (JJA) and Dec-Jan-
Feb (DJF), respectively. The primary and secondary rainy seasons in the equatorial
region are Sep-Oct-Nov (SON) and Mar-Apr-May (MAM, see *Bultot and Griffiths*
[1972] and Supplemental Information (SI) Figure S1). The mean annual precipitation is
about 1,500 mm. Rainforests occupy nearly 45% of the basin and are minimally disturbed
compared to the Amazon and Southeast Asian forests [*Gibbs et al.*, 2010; *Nilsson et al.*,
2005]. Grassland and savannah ecosystems, characterized by the presence of tall grasses,
closed-canopy woodlands, low-trees and shrubs, occupy another 45% [*Adams et al.*,
1996; *Bartholomé and Belward*, 2005; *Hansen et al.*, 2008; *Laporte et al.*, 1998]. Water
bodies (lakes and wetlands) occupy nearly 2% of the area and are concentrated mostly in
the southeastern and western equatorial parts of the CRB (Figure 1). Soils of the CRB
vary from highly weathered and leached Ultisols to Alfisols, Inceptisols and Oxisols
[*FAO/IIASA*, 2009; *Matungulu*, 1992]. Most soils are deep and well-drained, but they are
very acidic, deficient in nutrients, have low capacity to supply potassium and exhibit a
low cation exchange capacity [*Matungulu*, 1992].
In order to compare regional patterns in precipitation and runoff, we divided the
basin into four regions: i) Northern Congo (NC), ii) Equatorial Congo (EQ), iii)
Southwestern Congo (SW), and iv) Southeastern Congo (SE). The EQ region covers most
of the rainforest. The SE region consists of numerous interconnected lakes and wetlands.
Most of the CRB's population is concentrated in the NC, SE and SW regions [*Center for*
*International Earth Science Information Network (CIESIN) Columbia University et al.*,
2005].

### 2.2 Hydrologic model for the Congo River Basin


We used the Soil Water Assessment Tool (SWAT), a physically based, semi-

distributed watershed-scale model that operates at a daily time step [*Arnold et al.*, 1998;
*Neitsch et al.*, 2011]. The hydrological processes simulated include evapotranspiration,
infiltration, surface and subsurface flows, streamflow routing and groundwater recharge.
The model has been successfully employed to simulate river basin hydrology under wide
variety of conditions and to investigate climate change effects on water resources
[*Faramarzi et al.*, 2013; *Krysanova and White*, 2015; *Schuol et al.*, 2008; *Trambauer et*
*al.*, 2013; *van Griensven et al.*, 2012].

We delineated 1,575 watersheds within the CRB based on topography [*Lehner et*

*al.*, 2008]. Watershed elevations varied between 15 m and 2,700 m with a mean value of
680 m above mean sea level. Each watershed consisted of one stream section, where
near-surface groundwater flow and overland flow accumulated before being transmitted
through the stream channel to the watershed outlet.  Watersheds were further divided into
Hydrologic Response Units (HRUs) based on land cover (16 classes, *Bartholomé and*
*Belward* [2005]), soils (150 types, *FAO/IIASA* [2009]) and topography. The runoff
generated within each watershed was routed through the stream network using the
variable storage routing method. The average watershed size and the number of HRUs
within each watershed were 2,300 km$^2$ and 5, respectively. We also included wetlands
and lakes as natural storage structures that regulated the hydrological fluxes at different
locations within CRB (Figure 1). Detailed information was not available for the all the
lakes; therefore, we incorporated the largest 16 lakes (SI Table S1).
Simulated runoff, estimated for each HRU and aggregated at the watershed level,
was generated via three pathways: overland flow, lateral subsurface flow through the soil
zone and release from shallow groundwater storage. The Curve Number and a kinematic
storage routing methods were used to simulate overland and lateral subsurface flows, and
a nonlinear storage-discharge relationship was used to simulate groundwater contribution
(see *Arnold et al.* [1998]; *Neitsch et al.* [2011] and SI). A power law relationship was
employed to simulate the lake area-volume-discharge (see SI and *Neitsch et al.* [2011]).
The potential evapotranspiration was estimated using the temperature-based Hargreaves
method [*Neitsch et al.*, 2011]. The actual evapotranspiration was estimated based on
available soil moisture and the evaporative demand (i.e. potential evapotranspiration) for
the day. Additional details on model development and calibration are provided in the
Supplementary Information.
***2.3 Model simulation of historical hydrology with observed climate data***
We ran the hydrology model for the period 1950-2008. Estimates of observed
daily precipitation, and minimum and maximum temperatures needed to calculate
potential evapotranspiration were obtained from the Land Surface Hydrology Group at
Princeton University [*Sheffield et al.*, 2006]. In addition, measured monthly stream flows
were obtained at 30 gage locations (Figure 1) that had at least 10 years of records [*Global*
*Runoff Data Center.*, 2011; *Lempicka*, 1971; *Vorosmarty et al.*, 1998].
The model was calibrated using observed streamflows for the period 1950-1957 at
20 locations. The locations of streamflow gages and time period were chosen such that
they adequately captured climatic, land cover and topographic variability within the
CRB. The number of model parameters estimated by calibration varied from 10 to 13,
depending on the location of flow gages (e.g. gages with lakes within their catchment
area have more parameters). The calibration involved minimizing an objective function
defined as the sum-of-squared errors between observed and simulated monthly average
total discharge, baseflows (estimated by the baseflow separation method of *Nathan and*
*McMahon* [1990]) and water yield. The Gauss-Marquardt-Levenberg algorithm, as
implemented in a model independent parameter estimation tool [*Doherty*, 2004], was
used to adjust the fitted parameters and minimize the objective function. Parameter
estimation was done in two stages. First, parameters for the watersheds in the upstream
gages were estimated. Then the parameters for the downstream gages were estimated. To
test the calibrated model, simulated stream flows were compared to stream flows
measured at the same 20 locations, but during a period outside of calibration (i.e., 1958-
2008), as well as at 10 additional locations that were not used in the calibration.
**2.4 Hydrologic Simulations with Simulated Climate**

Historical climate simulations for the period 1950-2005 and climate projections

to 2065 for two greenhouse gas emission scenarios (Representative Concentration
Pathway – RCP), mid-range mitigation emission (RCP4.5) and high emission (RCP8.5),
were used to drive the hydrologic model. The RCP4.5 scenario employs a range of
technologies and policies that reduce greenhouse gas emissions and stabilize radiative
forcing at 4.5 W m$^{-2}$ by 2100, whereas the RCP8.5 is a business-as-usual scenario, where
greenhouse gas emissions continue to increase and radiative forcing rises above 8.5 Wm$^{-}$
$^{2}$ [*Moss et al.*, 2010; *Taylor et al.*, 2012]. We used monthly precipitation and temperature
outputs provided by 25 GCMs (Table 1) for the Fifth Assessment (CMIP5) of the
Intergovernmental Panel on Climate Change (IPCC).
GCM outputs may exhibit biases in simulating regional climate. These biases,
which are attributable to inadequate representation of physical processes by the models,
prevent the direct use of GCM output in climate change studies [*Randall et al.*, 2007;
*Salathé Jr et al.*, 2007; *Wood et al.*, 2004]. Hydrological assessments that use GCM
computations as input inherit the biases [*Salathé Jr et al.*, 2007; *Teutschbein and Seibert*,
2012]. To mitigate this problem, we implemented a statistical method [*Li  et al.*, 2010] to
bias-correct the monthly historical precipitation and temperature data. In brief, the
method employs a quantile-based mapping of cumulative probability density functions
for monthly GCM outputs onto those of gridded observations in the historical period. The
bias correction is extended to future projections as well. The observed data used in the
modeling and bias-correction has some limitations. That is, the number of precipitation
gages decreased over the period from 1950 to 1990, and the density of the gages is sparse
compared to the size of the river basin (see Section 3.4 and SI). However, we assumed
that the available ground-based observations combined with satellite-based and reanalysis
data adequately captured the spatiotemporal variability in precipitation. Studies by
*Munzimi et al.* [2014] and *Nicholson* [2000] draw similar conclusions.
The simulated monthly precipitation and temperature values were temporally
downscaled to daily values for use in the CRB hydrology model. We used the three-
hourly and monthly observed historical data developed for the Global Land Data
Assimilation System [*Rodell et al.*, 2004; *Sheffield et al.*, 2006] and the bias-corrected
monthly simulations to generate three-hourly precipitation and temperature data, which
were subsequently aggregated to obtain daily values (see SI Methods). The hydrological
model was forced with the bias-corrected and downscaled daily climate for the period
1950-2065. Due to the lack of information on the effect of $CO_2$ on the 16 land cover
classes simulated, the ambient $CO_2$ concentration was maintained at 330 ppm throughout
the simulation period. A recent study suggests that, in tropical rainforest catchments,
elevated $CO_2$ has little impact on evapotranspiration, but results in increased plant
assimilation and light use efficiency [*Yang et al.*, 2016].

A total of 50 projections (25 RCP4.5 and 25 RCP8.5 projections, see Table 1)

were compiled and analyzed. Results of individual and multi-model means (un-weighted
average of all models (MM) and an average of select models (SM)) for the near-term
(2016-2035) and mid-term (2046-2065) projections are presented.

Accessible flows (AF), which exclude surface runoff associated the storm events,

were estimated by applying a baseflow separation method described in *Nathan and*
*McMahon* [1990].
**3. Results and Discussion**
***3.1 Historical simulations***

Historical observations of average annual precipitation vary from 1,100 mm in the

southeastern portion of the CRB to 1,600 mm in the CRB's equatorial region. We
compared the GCM-simulated annual precipitation and its inter-annual variability during
the historical period with observations from 30 locations within the CRB (Figure 2). The
simulated inter-annual variability among the climate models (vertical bars in Figure 2)
lies within the range of the observed variability (horizontal bars in Figure 2). The linear-
regression slope of 1.16 ($p < 0.001$, Figure 2) between the annual observed and the multi-
model mean shows that bias-corrected precipitation is slightly over-estimated, but not
significantly so.  Observations of seasonal precipitation are reproduced similarly well by
the GCM models (SI Figure S2 and Table S2). The good agreement between GCM-
simulated and observed rainfall is expected given our bias correction of the GCM output.

We compared the simulated monthly runoff at 30 locations with observations

(Figure 3A and SI Table S3). The colored points compare observed mean annual runoff
at the 30 gage locations with historical simulations (hydrological model forced with
observed climate), while the vertical and horizontal bars show the modeled and observed
inter-annual variability, respectively. The shades of colors (from light-green to yellow
and red) reveal the model's skill in simulating the monthly flows in the historical period.
The Nash-Sutcliff coefficient of efficiency (NSE), a measure of relative magnitude of
residual variance compared to the monthly observed streamflow variance [*Legates and*
*McCabe*, 1999; *Nash and Sutcliffe*, 1970], varies between 0.01 and 0.86 (color scale in
Figure 3A). The NSE can range from negative infinity to 1, with values between 0.5 and
1 considered satisfactory [*Moriasi et al.*, 2007].  Seventeen of the 30 gages show NSE
greater than or equal to 0.5. Higher NSE values at locations on both sides of the equator,
particularly at major tributaries (NSE ~ 0.60, gages 1 to 8 in Figure 1 and SI Figure S3)
suggest that the model reliably simulates stream flows under different climatic
conditions. High NSE values also indicate that the seasonal and annual runoff
simulations, including the inter-annual variability in the historical period, are in good
agreement with observations. The catchment areas of the 30 gages vary between 5,000
$km^2$ and 900,000 $km^2$ (excluding the last two downstream gages, SI Table S3) and
encompass a range of land cover and climatic regions on both sides of the equator; thus,
the hydrology model exhibits reasonable skill in simulating runoff over a wide range of
watershed conditions.
Comparison of modeled runoff forced with GCM-simulated and observed climate
(Figure 3B) reveals generally acceptable runoff simulations in the CRB. The black dots
and red (blue) vertical bars in Figure 3B show multi-model mean and maximum
(minimum) range of inter-annual variability in the 25 historical GCM simulations. The
results suggest that model-data agreement in precipitation translates to similarly
acceptable runoff simulations.
Runoff patterns reflect seasonal rainfall that varies asymmetrically on either side
of the equator (see SI Figure S1). For example, the observed peak runoff at streamflow
gages 2 and 6 located north and south of the equator (see Figure 1) occur near the end of
the rainy seasons – during Sep-Oct and Mar-Apr, respectively (Figure 4).  Augmented by
flows from northern and southern tributaries (e.g. gages 1, 2, 4 and 6) and by high
precipitation in the tropical equatorial watersheds during the two wet seasons (MAM and
SON), the main river flows (downstream of gage 3 in Figure 1) show low variability
(Figure 4).  Differences in stream-flow variability between the main river and its
tributaries are illustrated through comparison of the coefficient of variation, which equals
only 0.23 at the basin outlet (gage 8), but 0.77 and 0.40 along the northern tributary (gage
2) and southern tributary (gage 4), respectively.
Runoff in the northern (NC) and southern (SW and SE) watersheds is strongly
seasonal with long dry seasons, but this is not the case in the equatorial region (Figure 5).
Average watershed runoff varies between 20-70 mm during dry seasons to 100-140 mm
during wet seasons in the NC, SW and SE. In the equatorial region, seasonal runoff varies
between 100-150 mm with the highest in SON. Overall, the precipitation-runoff ratio is
about 0.30 in the CRB. The accessible runoff (AF) that can be appropriated for human
use, and hence excludes runoff associated with flood events, is about 70% of the total
runoff.
***3.2 Future projections in precipitation and runoff***
***3.2.1 Precipitation***
*Aloysius et al. [2016]* showed that GCM projections of temperature generally
increase under both emission scenarios in line with the historical upward trend for Africa
[*Hulme*, 2001]; however, precipitation projections contain large uncertainties. The
modeled near-term (2016-2035) precipitation projections in the CRB vary between -4%
and 6% with a multi-model mean (MM) change of 1% under the two emission scenarios
relative to the reference period (1986-2005). Regionally, the northern CRB shows the
largest annual increase in precipitation followed by southwestern and equatorial regions.
However, the inter-model variability is larger than the MM in all regions, indicating
greater projection uncertainties in both emission scenarios (Table 2). The mid-term
(2046-2065) projections of annual precipitation vary between -5% and 9%, with the MM
of 1.7% and 2.1% for RCP4.5 and RCP8.5, respectively. More than 70% of the
ensembles in both RCPs project an increase in annual precipitation in the CRB over the
mid-term. The multi-model mean of all ensembles that project an increase (decrease) in
precipitation is 2.7% (-2.4%) for RCP4.5 and 4.0% (-2.9%) for RCP8.5.
The GCMs project considerable spatial and seasonal variations in precipitation
(Table 2 and Figure 6). However, the standard deviation of annual and seasonal
projections within the four regions exceed or equal the MM, indicating little agreement
on the direction of change. The spatial patterns (Figure 6), on the other hand, show
regions where modeled projections strongly agree on increasing or decreasing
precipitation. For example, decreasing precipitation is projected in most of the headwater
catchments in the southern and parts of northern CRB.

In general, the GCMs project decreasing precipitation in the driest parts of the

southern CRB (mostly in Southeastern CRB, but portions of Southwestern as well).
Under the RCP8.5 scenario, parts of northeastern CRB also experiences a reduction in
precipitation in the near-term (regions in Figure 6 with fewer GCMs projecting an
increase in precipitation). The areas of decreased precipitation shrink in the southeast and
southwest in the mid-term; however, drying expands in parts of northern CRB under the
two emission scenarios. Most GCMs (14-20) project a precipitation increase outside of
southeastern CRB.

Inter-model variability in precipitation projections are sensitive to seasons and

climate region (Figure 7A-D). At monthly scale, the northern and southern regions
receive less than 50 mm of precipitation for at least three months, which persist in the
future under both emission scenarios. The dry season is more prolonged in the southeast
compared to the rest of the CRB. The inter-model variability is larger in the rainy seasons
under RCP8.5, compared to RCP4.5. Larger variability under RCP8.5 highlights that
GCMs may have limited skill in simulating precipitation under high greenhouse gas
emissions.
*3.2.2 Runoff*

In general, modeled runoff increases, and its inter-annual variability within GCMs

is larger during high flow periods compared to low flow periods, except in the equatorial
region (Figure 7E-H, see Figure 1 for regions).  The model projection uncertainty
increases towards the middle of century, particularly under the RCP8.5 emission
scenario. The temporal patterns of runoff in the near- and mid-terms are similar to the
precipitation patterns, but with a time lag. As with precipitation, the monthly runoff
shows prolonged periods of low values in the northern and southern CRB in both
projection periods. Parts of northern, southeastern, and southwestern CRB also show
reduced runoff projections relative to the reference period under both RCPs; these
reductions are predominantly in the areas where fewer GCMs agree on the increase in
modeled precipitation (see Figure 6 and SI Tables S4 and S5). The area of decreasing
runoff expands in the northern CRB under both emission scenarios in the mid-term (see
Figure 6, which shows that more models agree on decreasing precipitation in parts of
northern CRB that subsequently results in decreasing runoff). Although the northern and
equatorial CRB show an overall increase in precipitation, the decrease in runoff in certain
parts in the northern and equatorial CRB is caused by reduction in seasonal precipitation
(e.g. JJA and SON, see SI Table S4). A larger reduction – up to 15% – in the southeastern
CRB covering most of northern Zambia is due to an overall decrease in precipitation
simulated by more the half of the GCMs (see Figure 6).

The multi-model mean of total runoff from the CRB shows an increase of 5%

($\pm6\%$, one standard deviation, n = 25) and 7% ($\pm8\%$) in the near- and mid-terms under
both RCPs relative to the reference period (1986-2005). Annual runoff in the equatorial
region, which receives the highest precipitation, is projected to increase by up to 5%
(±7%) in the near-term to 6% (±8%) and 7% (±9%) in the mid-term for RCP4.5 and
RCP8.5, respectively. The increases are greater in the secondary rainy season (MAM)
than the primary (SON, Figure 7 B and F).  The majority of the ensembles project an
increase in monthly runoff within the equatorial CRB, with the RCP8.5 ensembles
exhibiting larger variability (Figure 7F).

Runoff that can be appropriated for human use is generated mostly in the

northern, southeastern and southwestern CRB, which at present varies from 130 mm/year
in the southeastern CRB to 250-400mm/year in the northeastern and southwestern CRB.
Runoff is projected to increase in all three of these regions. However, the inter-model
variability is greater than twice the MM in nearly all the regions and during all four
seasons (Figure 8 and Table 3). In most cases, the largest uncertainties are in non-rainy
seasons and under high emission RCP8.5 scenario (e.g. DJF in the northern CRB, Figure
8B, and JJA in the southeastern CRB, Figure 8H).
*3.3 Variability in accessible flows*

Only part of the runoff may be appropriated for human use. In the CRB, the

accessible runoff (AF), excluding runoff associated with flood events, is about 70%. The
AF is largely under-utilized, but its appropriation is expected to increase in the future,
mostly in the populated areas of northern, southwestern and southeastern CRB. We
present the uncertainty associated with GCM and scenario selection by quantifying
seasonal and inter-model variability in AF at eight major tributaries (identified in Figure
1) that drain watersheds across a range of climatic regions on both sides of the equator
(Figure 9). Modeled AF exhibits substantial inter-model spread in the near-term and
widens in the mid-term (SI Figure S4). The inter-model variability is larger during high
flow periods compared to low flow periods.

Following the general pattern of increasing precipitation and runoff in the

northern and southwestern watersheds, we find that AF increases with greater model
agreement in tributaries that drain these watersheds (e.g. gages 1, 2 and 6 in Figure 9). A
closer look at tributaries in the northern and southwestern CRB reveals better agreement
of increased AF during low flow periods compared to high flow periods (compare gages
1, 2, 6 and 7 in Figure 9). In contrast, tributaries that drain southeastern watersheds
exhibit greater variability in modeled AF with majority of the ensembles projecting a
reduction (e.g. gages 4 and 5 in Figure 7). Overall, the AF in the main tributary (gages 3
and 8) is projected to increase, partly due to the contributions from the northern and
southwestern tributaries. The decrease in modeled precipitation and AF in the
southeastern CRB appears to have marginal effect on downstream flows in the main
river.

The spatial and temporal variations in the projected AF have consequence for

water resources development and management. For example, projections of increased
AF near the proposed Grand Inga Hydropower project (near gage 8, *Showers* [2009]) is
robust compared to the large variations near the proposed trans boundary water diversion
in the southeast (near gage 5, *Lund et al.* [2007]). Reductions in high and low flows in
streams in the southeastern region will have implications to aquatic life, channel
maintenance and lake and wetland flooding.

### 3.4 Sources of uncertainty


The sources of uncertainty encountered in this work can be broadly categorized
into i) observational uncertainty, particularly the sparse and declining network of
precipitation and stream flow gages and ii) model uncertainty, which in the GCMs
includes model structure, model initialization, parameterization and climate sensitivity
(i.e., the response of global temperature to a doubling of CO2 relative to pre-industrial
levels). We used only one hydrological model, which is also a source of uncertainty.
However, variation in climate signals between GCMs and emissions scenarios,
particularly precipitation projections, may be a larger source of uncertainty than the
choice of hydrology model [*Thompson et al.*, 2014; *Vetter et al.*, 2016].
The climate data used for bias-correction and for historical hydrologic simulations
has its own uncertainties. Gage-based, satellite derived data and reanalysis outputs are
used to develop the historical observations [*Sheffield et al.*, 2006]. Precipitation gages
were more numerous at the beginning of the simulation period and declined in number
toward the end of the 20th century [*Mitchell and Jones*, 2005; *Washington et al.*, 2013].
Available gage data varied both spatially and temporally (SI Figure S5 and S6). For
example, the equatorial region – nearly a third of CRB – had about 70 rain gages through
early 1990s, but only 10% of these were functioning by 2005 (SI Figure S5). The
southeastern and parts of northern CRB also had good rainfall-gage coverage, which has
similarly decreased since the 1990s [*Mitchell and Jones*, 2005]. However, satellite-based
and sparsely distributed gage data has been used to demonstrate that spatiotemporal
distribution of precipitation can be sufficiently described in the CRB region [*Munzimi et*
*al.*, 2014; *Nicholson*, 2000; *Samba et al.*, 2008]. We assume that, even with these
limitations, the available historical data are adequate to model the hydrology of the CRB.
In addition to climate data, observed runoff data are another limitation that could
restrict proper validation of hydrological models. However, we utilized a time period
(1950-1959) when the CRB had maximum coverage of both precipitation and runoff data
to calibrate and test the hydrology model (for example see evidence in *L'vovich* [1979]).
Where available, we used additional runoff data to further test model outputs during the
historical period. The runoff gage locations are distributed within the CRB (see Figure 1)
such that they adequately capture climatic, land cover and topographic variability.
For future projections, the largest sources of uncertainty arise from the GCMs and
emission scenarios. GCMs do not consistently capture observed rainfall seasonality and
heavy rainfall in regions of the central CRB, and in most cases do not show key features
such as seasonality and heavy rainfall regions of central CRB [*Aloysius et al.*, 2016;
*Washington et al.*, 2013]. The biases in the GCM-simulated precipitation, particularly in
the tropical regions, have been attributed to multiple factors including poorly resolved
physical processes such as the mesoscale convection systems, inadequately resolved
topography due to the coarse horizontal resolution and inadequate observations to
constrain parameterization schemes. These limitations are unavoidable in the current set
of CMIP5 projections. We assume that the combination of GCM outputs used in our
work, and the bias-correction method, which maintains key statistical properties in the
original GCM outputs (see *Aloysius et al.* [2016] and *Li  et al.* [2010]), adequately
captures the uncertainties in GCM and emission scenarios. Based on monthly
precipitation climatology, *Aloysius et al.* [2016] found no significant shift in seasonality
in modeled future precipitation projections.

The range of projections presented here for the two emission scenarios also

highlight the uncertainties planners would encounter when making climate-related
decisions. For example, broader agreement on increase in runoff in parts of the CRB
would help make robust decisions, whereas weaker agreement in the southern CRB calls
for greater scrutiny of regional climate. Generally, the MM approach reduces the
uncertainty because averaging tends to offset errors across models. However, one could
also ask whether this approach would work with fewer models.

*Washington et al.* [2013] and *Siam et al.* [2013] presented evidence that

evaluating atmospheric moisture flux (which is modulated by wind patterns and
humidity) and soil water balance is a better way to diagnose GCM performance in data
scarce regions like the CRB. *Balas et al.* [2007], *Hirst and Hastenrath* [1983] and
*Nicholson and Dezfuli* [2013] have shown that sea surface temperature (SST) anomalies
in the Atlantic and Indian ocean sectors could partly explain precipitation in the CRB
region. Along the same lines, *Aloysius et al.* [2016] identified five models as suitable
candidates. We examined this subset of GCM projections (M6, M7, M18, M23 and
M24), which we refer to as the select model average, or SM (see refs. *Giorgetta et al.*
[2013]; *Good et al.* [2012]; *Jungclaus et al.* [2013]; *Meehl et al.* [2013]; *Siam et al.*
[2013]; *Voldoire et al.* [2012]; *Yukimoto et al.* [2006] and *Aloysius et al.* [2016] for
further comparison of GCM performance). By evaluating seasonal atmospheric moisture
and soil water balance in 11 CMIP5 GCMs in the CRB and Nile River basin regions,
*Siam et al.* [2013] identified M7, M18 and M24 as good candidates for climate change
assessment.

Focusing on the northern, southeastern and southwestern CRB, where human

appropriation of runoff is expected to increase, we find that the projected increase of
annual runoff in SM is more than that of MM (20% to 50% higher in the SM compared to
MM). And, the extent of reduction in runoff in the south is concentrated in the
southeastern upstream watersheds in both MM and SM, although the magnitude of
decrease is smaller in SM (SI Table S4 and S5).

From the viewpoint of water resources for human appropriation, the changes by

seasons are also important. Future changes and uncertainties in modeled seasonal runoff
averaged over the four regions are presented Figure 8. In comparison with the CRB
projections, the uncertainties in sub-regions are larger. Nearly all the MM and SM
projections show an increase in runoff in all the four seasons; however, there is
substantial inter-model variability. The uncertainties increase under the high emission
RCP8.5 scenario during the mid-century. Considering the southeastern region as an
example, under RCP8.5 emission scenario, uncertainties reported as one inter-model
standard deviation in the mid-term are $\pm20\%$, $\pm27\%$, $\pm26\%$ and $\pm13\%$, respectively for
DJF, MAM, JJA and SON seasons, and are greater than the MM and SM. Further, the
deviation of uncertainty within the sub-regions of CRB increases under high emission
RCP8.5 scenario. For example, the inter-model projection ranges are larger in the
northern and southeastern CRB (Figure 8 B and H) compared to the equatorial and
southwestern CRB (Figure 8 D and F). Finally, the uncertainty assessment presented here
represents climate model uncertainty arising from emission scenarios, different response
to the same external forcing, different model structures and parameterization schemes.
While these uncertainties in projections pose challenges for robust decision making, they
also provide insights into where further research might be most valuable.
**4. Conclusions**

From the point of view of climate change adaptation related to water resources,

agriculture, and ecosystem management, the challenge faced by CRB countries is
recognizing the value of making timely decisions in the absence of complete knowledge.
In some settings, climate change presents opportunities as well as threats in the CRB. The
projected increases in accessible runoff imply new opportunities to meet increasing
demands (e.g. drinking water, food production and sanitation), while the enhanced flood
runoff would pose new challenges (e.g. flood protection and erosion control). On the
other hand, water managers could face different challenges in the southeast where
precipitation and runoff are projected to decrease.

GCM-related variability in regional climate projections could be constrained by a

subset of models based on attributes that modulate large-scale circulations (see *Knutti*
*and Sedlacek* [2013] and *Masson and Knutti* [2011]). This approach is particularly useful
because regions like the CRB lack complete coverage of observational data but the
mechanisms that moderate the climate system, particularly precipitation, are fairly well
understood [*Hastenrath*, 1984; *Nicholson and Grist*, 2003; *Washington et al.*, 2013]. Yet,
the span in rainfall predictions among the MM, SM, and individual GCMs suggest that,
despite the advances in climate modeling, significant uncertainties in precipitation
projections for CRB persist.
Rather than providing a narrow pathway for decision-making, our results, for the
first time for CRB, provide a framework to i) assess implications under various climate
model assumptions and uncertainties, ii) characterize and expose vulnerabilities and iii)
provide ways to guide the search for impact-oriented and actionable policy alternatives,
as emphasized by *Weaver et al.* [2013]. Projections and associated uncertainties vary
widely by region within the CRB, and therefore diverse but robust planning strategies
might be advisable within the river basin. We emphasize that projections provided here
could be considered as part of the process of incorporating multiple stressors into climate
change adaptation and engaging stakeholders in the decision making process.
**Acknowledgements**
We would like to thank Nadine Laporte, Innocent Liengola, Peter Umunay, Greg Fiske
and Melanie Burr for help with data and literature search. We acknowledge the World
Climate Research Program's Working Group on Coupled Modeling, which is responsible
for CMIP, and we thank the climate modeling groups (Table 1) for producing and making
available their model output. For CMIP, the U.S. Department of Energy's Program for
Climate Model Diagnosis and Inter-comparison provides coordinating support and led
development of software infrastructure in partnership with the Global Organization for
Earth System Science Portals.  We gratefully acknowledge the efforts of three
anonymous reviewers who made thoughtful comments that substantially improved the
manuscript. This work was supported in part by the facilities and staff of the Yale
University Faculty of Arts and Sciences High Performance Computing Center, and by the
National Science Foundation under grant CNS 08-21132 that partially funded acquisition
of the facilities.

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

# 1   Figures in the main text


Figure 1 Congo River Basin: the river basin boundary, the extent of the rainforest, locations of lakes and wetlands, and the locations of streamflow gages are shown. The "all other vegetation" category includes grasslands and savanna ecosystems, and all managed areas. Bartholome et al., (2005) provide further details on land cover in the Congo River basin.

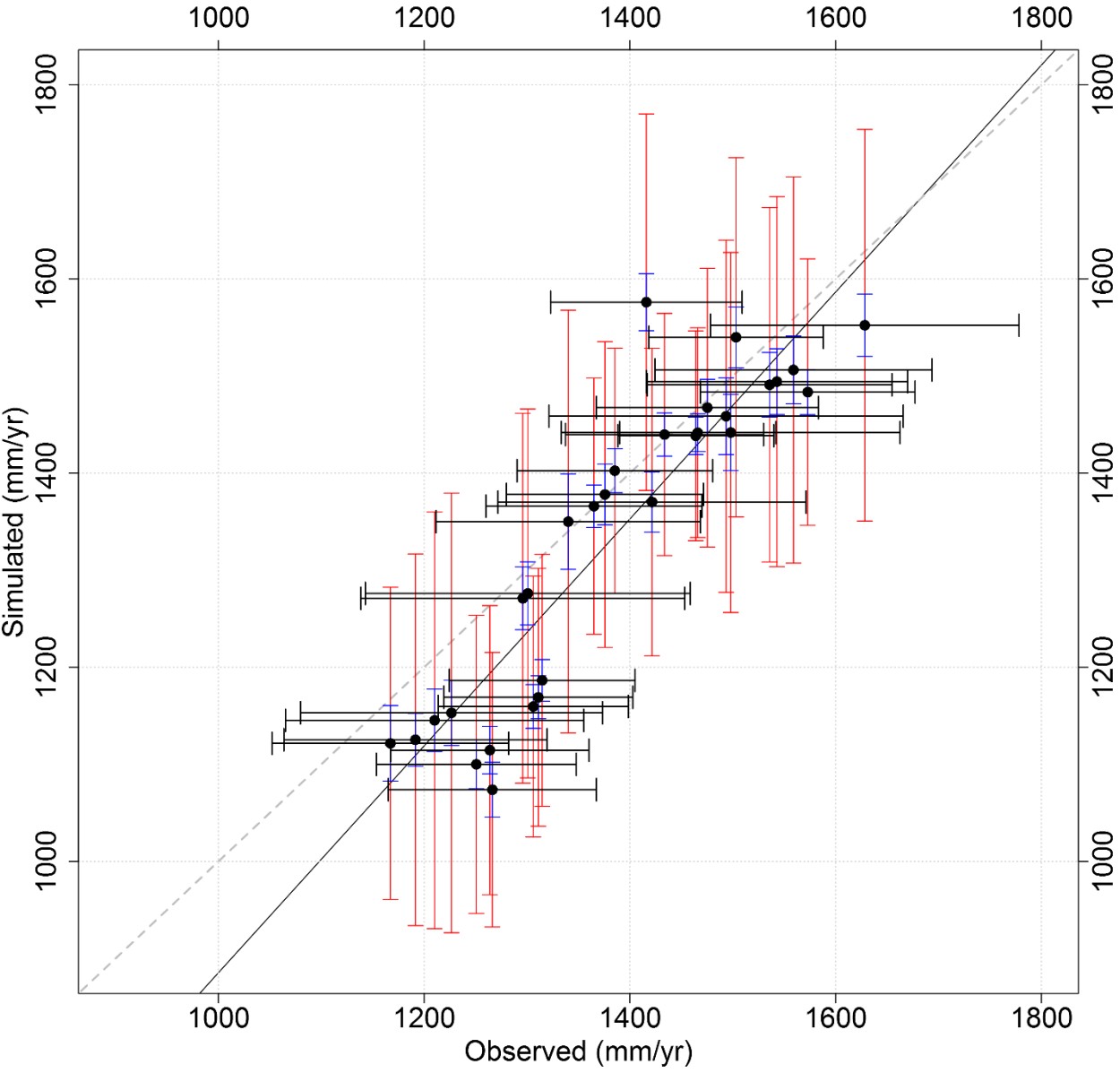

779

Figure 2 Comparison of observed and bias-corrected GCM-simulated average annual precipitation for 30 catchments with stream-flow gages (shown in Figure 1) in the historical period (1950-2005). Y-axis values are statistically downscaled GCM-simulated precipitation. Black dots compare multi-model means with observed precipitation, black horizontal bars show observed inter-annual variability ($\pm$ one standard deviation), and red (blue) vertical bars show maximum (minimum) range of modeled inter-annual variability ($\pm$ one standard deviation) within the 25 climate model outputs. The black line is linear regression fit between observed and multi-model mean of simulated precipitation ( $y = 1.16 \pm 0.204x - 283.4, p < 0.001, R^2 = 0.825$); parameter bounds are 95% confidence interval. The gray dashed line is the 1:1 line.


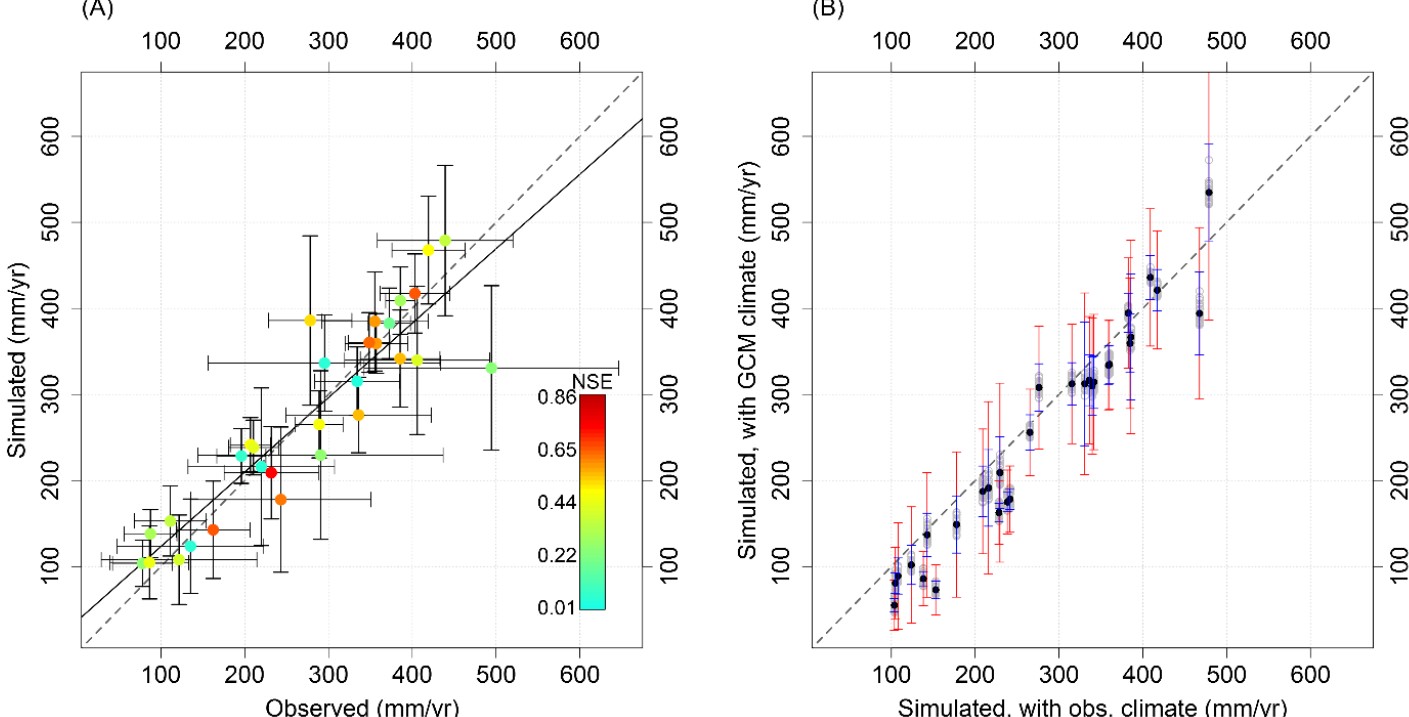


Figure 3. Comparison of observed and simulated annual runoff at the 30 streamflow gage locations (shown in
Figure 1). (A) Historical simulations with observed climate: the positions of the colored dots compare annual
values of observed and simulated historical runoff; the dots' colors (see legend) show the Nash-Sutcliff
coefficient of efficiency (NSE) of observed vs. simulated monthly stream flows; and the black horizontal and
vertical bars show observed and modeled inter-annual variability ($\pm$ one standard deviation), respectively. The
black line is linear regression fit between annual simulated and observed runoff ($y = 0.865 \pm 0.158x +$
$36.63, p < 0.001, R^2 = 0.82$), parameter bounds are the 95% confidence interval. (B) Simulations in the
historical period with GCM-simulated climate: black dots show the multi-model mean; red (blue) vertical bars
show modeled (forced with GCM-simulated historical climate) maximum (minimum) inter-annual variability ($\pm$
one standard deviation) within the 25 simulations; and gray circles show multi-year mean of individual GCM
simulations. The gray dashed lines in A and B are 1:1 line. The GCM-simulated outputs are statistically
downscaled and bias-corrected.

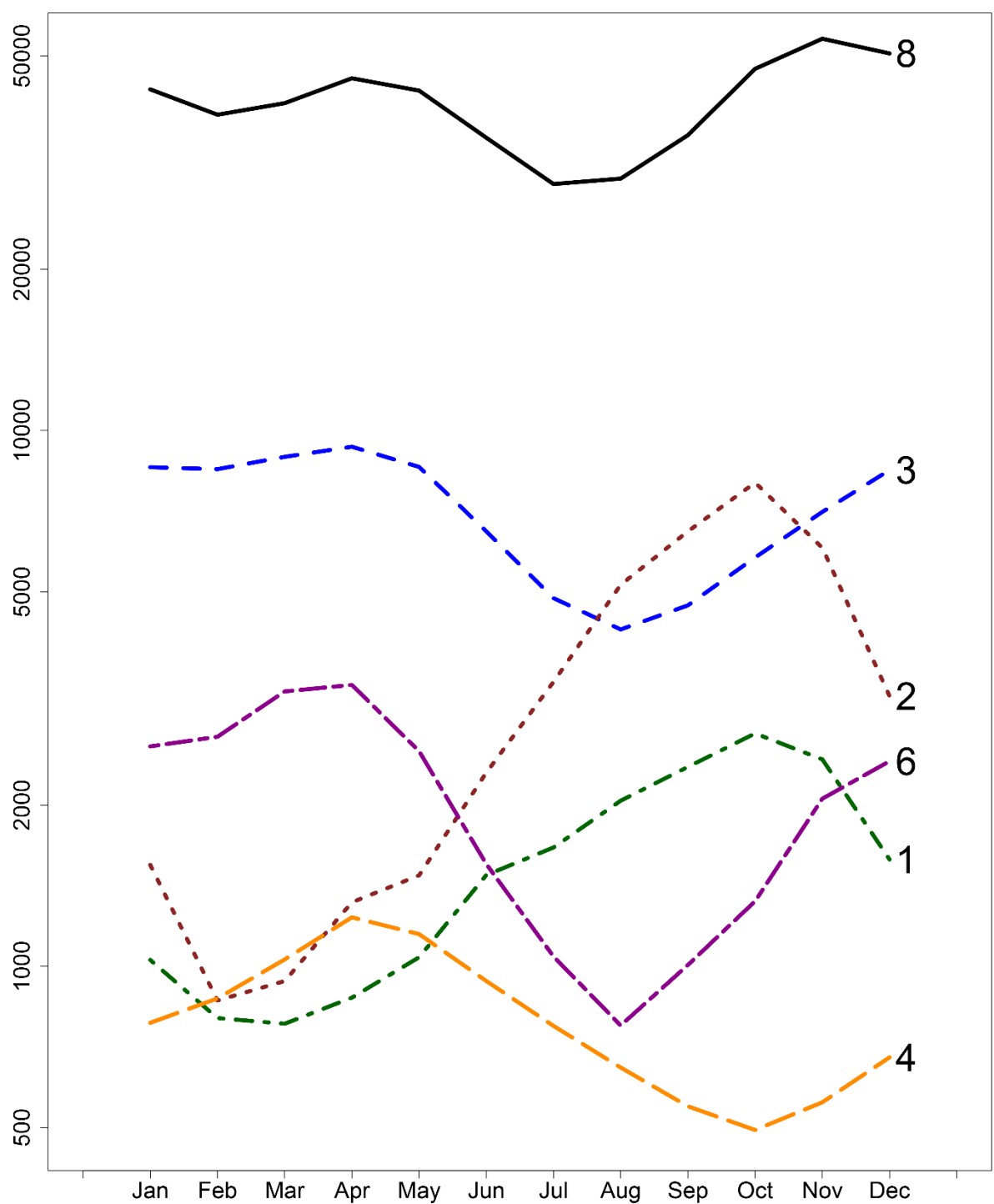


Figure 4 Mean monthly flows at selected tributaries in the CRB. Flows are in $m^3$/s and gage numbers are
identified in Figure 1. Monthly values are based on simulated flows (forced with observed precipitation) for the
period 1950-2005.



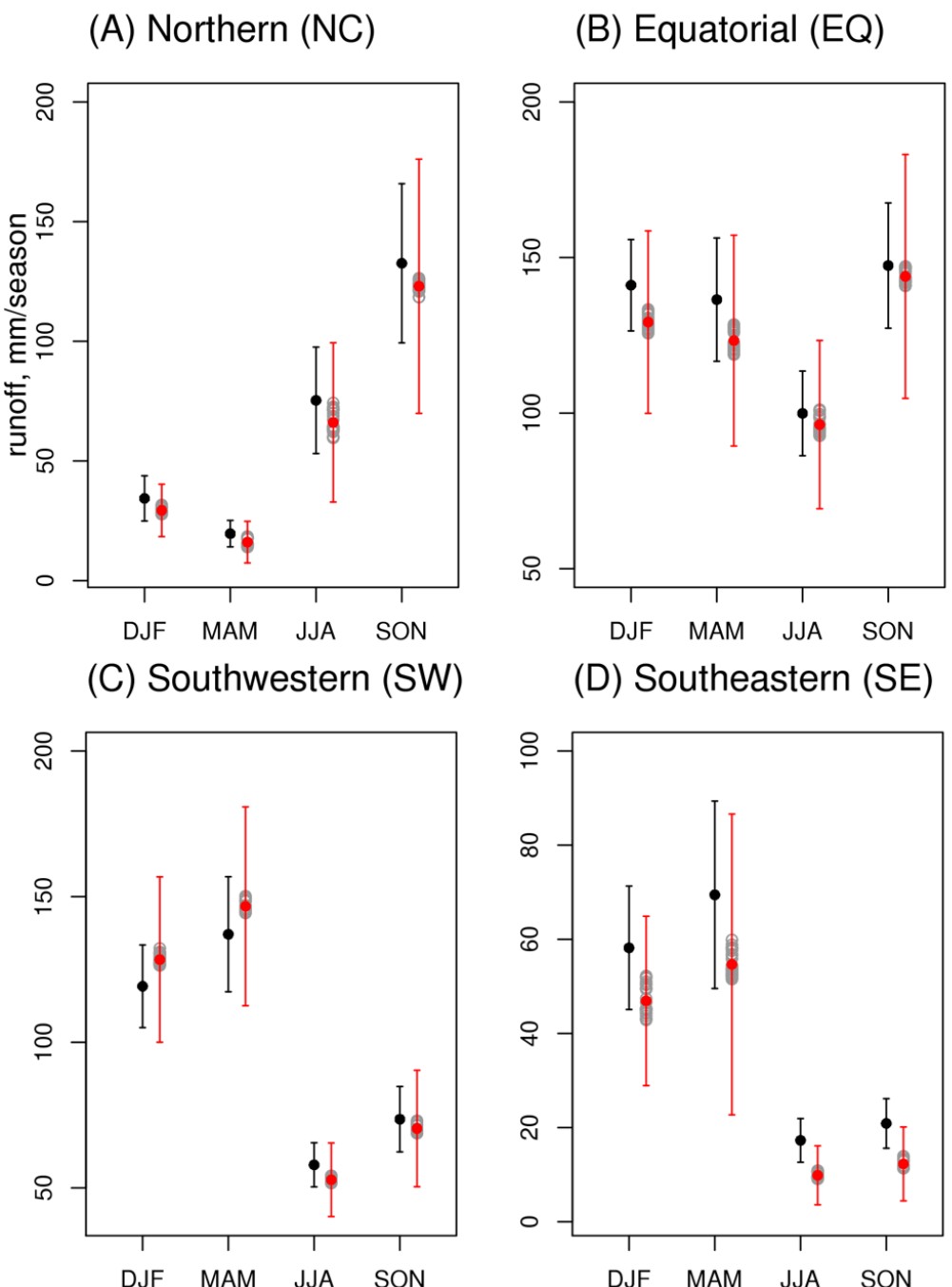


Figure 5 Seasonal variation in runoff in (A) Northern, (B) Equatorial, (C) Southwestern and (D) Southeastern
Congo River Basin for the historical period, 1950-2005. The seasonal total runoff are calculated for Dec-Jan-
Feb (DJF), Mar-Apr-May (MAM), Jun-Jul-Aug (JJA) and Sep-Oct-Nov (SON). Black dots and vertical bars
show the modeled inter-annual variability forced with observed climate, red dots show the multi-model mean
(MM) forced with GCM-simulated climate, red vertical bars show the maximum range of inter-annual
variability within the 25 models and the grey open circles show the mean of individual models. Y-axis scale is
different for each plot.

817

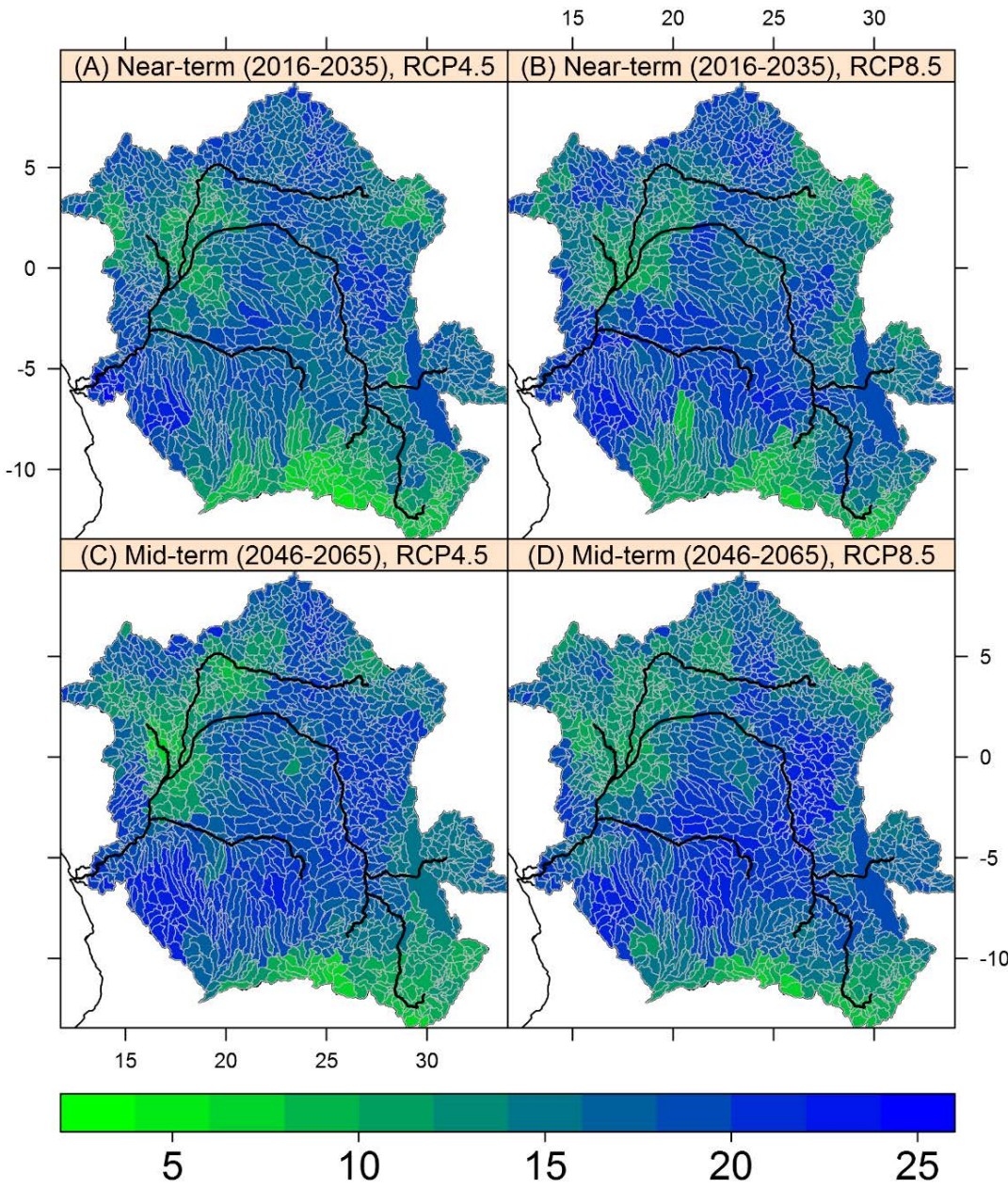

Figure 6 Number of climate model outputs projecting an increase in precipitation in the (A) near-term, 2016-2035, RCP4.5, (B) near-term RCP8.5, (C) mid-term, 2046-2065, RCP4.5 and (D) mid-term RCP8.5. The number of modeled precipitation outputs considered is 25. Main rivers and lakes are shown.

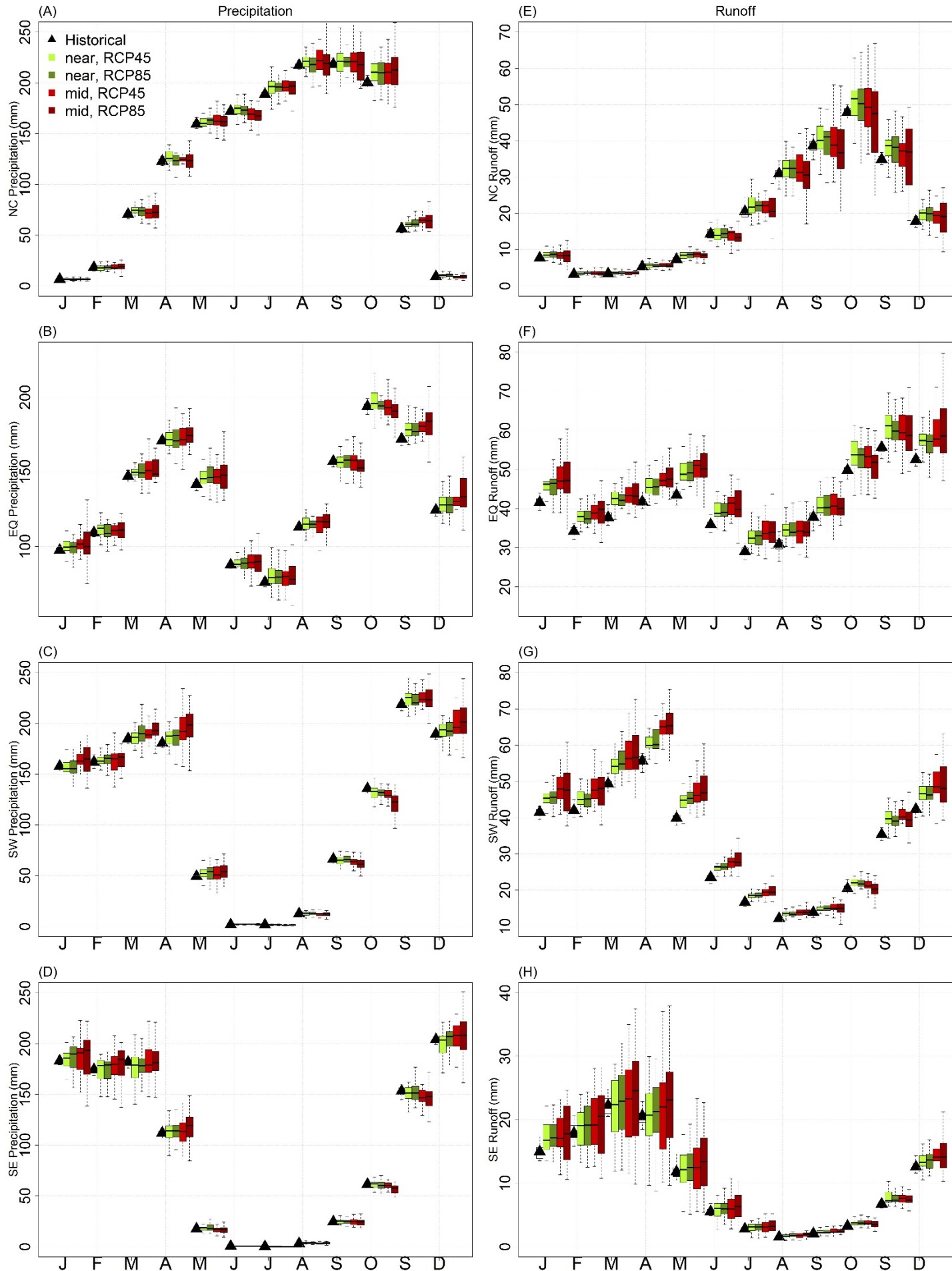


Figure 7 Monthly variation of precipitation (A-D) and runoff (E-H) in the four regions shown in Figure 1. Box-
and-whiskers for each month shows the inter-model variability for the historical period (black), near-term
RCP4.5 (light green), near-term RCP85 (dark green), mid-term RCP4.5 (red) and mid-term RCP8.5 (brown).
The upper and lower end of the boxes show the 75th and 25th quartiles, the bar inside each box shows the
median, and the whiskers cover approximately 90% of the values. The multi-model mean value for the
reference period is shown as triangles for clarity. All values are in mm/month. NC – northern, EQ – equatorial,
SE – southeast and SW – southwest, see Figure 1 for locations.

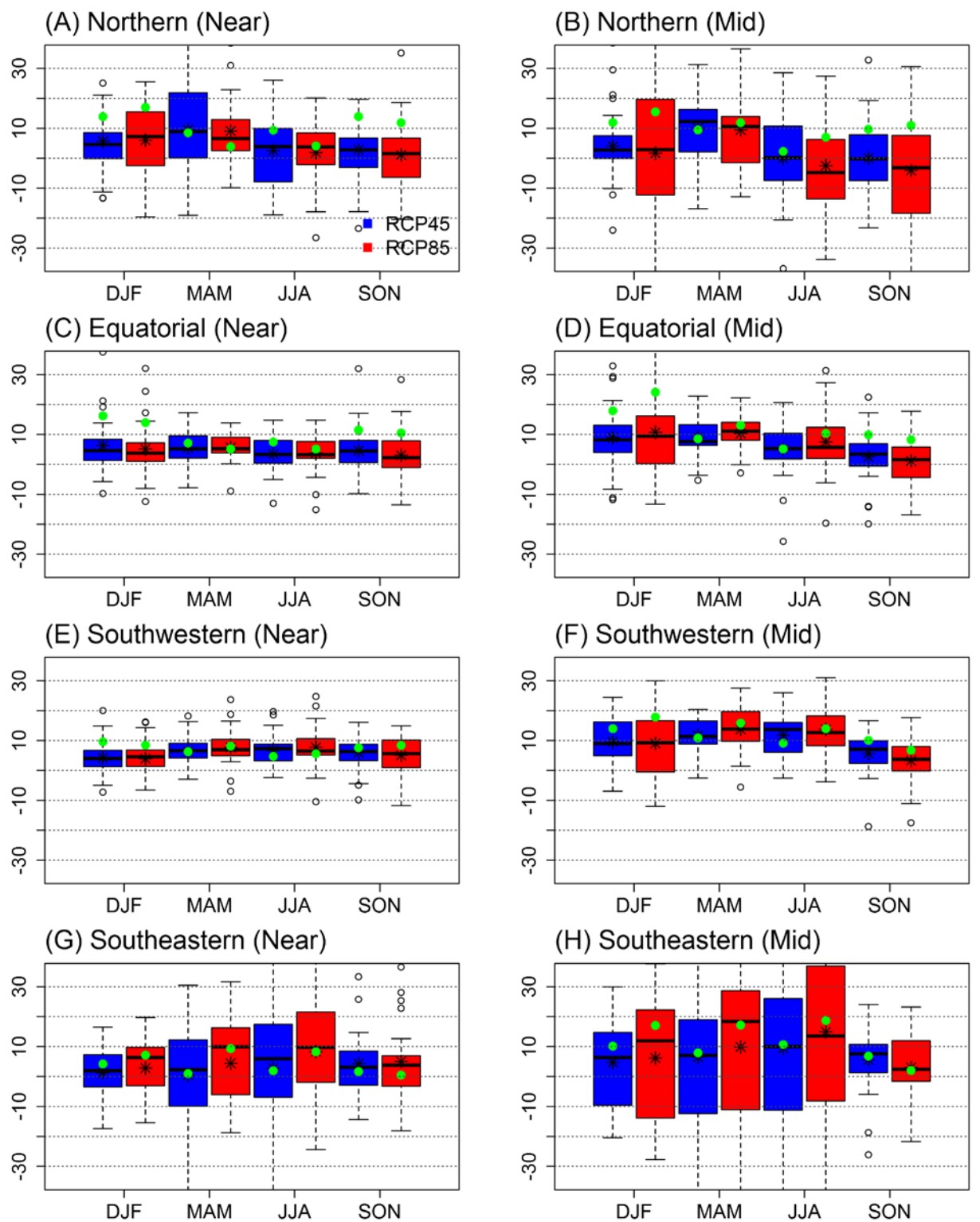


Figure 8 Seasonal runoff projections (as percent relative to the reference period 1986-2005) for the near-term
(2016-2035) and mid-term (2046-2065) projection periods for northern (A-B), equatorial (C-D), southwestern
(E-F) and southeastern (G-H) regions. Boxes show the $25^{th}$ and $75^{th}$ percentiles, the horizontal line within the
boxes show median value and the whiskers mark the $5^{th}$ and $95^{th}$ percentiles. The multi-model mean (asterisks)
and the select-model mean (green dots) are also shown. The y-axis range is limited to show the smaller boxes.
Y-axis values are in percentages.


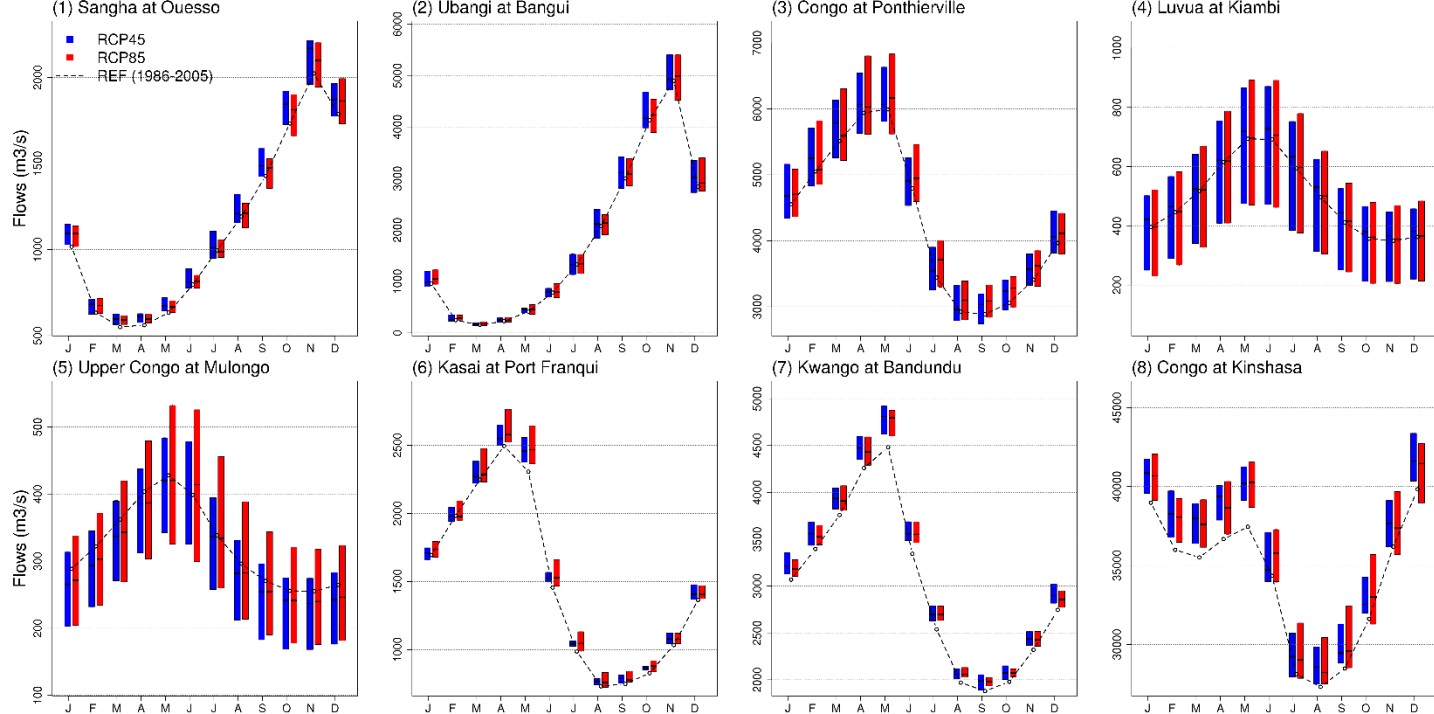


Figure 9 Accessible streamflow hydrographs in the near-term at selected locations shown in Figure 1A. Blue
and red bars (RCP 4.5, RCP 8.5, respectively) show the inter-model variability. The dotted black line shows the
hydrograph in the reference period (1986-2005). Plot numbers 1-8 coincide with the gage numbers in Figure 1.


**2 Tables in the main text**

Table 1 Global Climate Models whose outputs are used in this study. Further details about comparison of model
outputs and key references for GCMs are given in Aloysius et al., 2016.

| Model Number | Model Name |
| --- | --- |
| M1 | ACCESS1-3 |
| M2 | bcc-csm1-1 |
| M3 | BNU-ESM |
| M4 | CanESM2 |
| M5 | CCSM4 |
| M6 | CESM1-CAM5 |
| M7 | CNRM-CM5 |
| M8 | CSIRO-Mk3-6-0 |
| M9 | EC-EARTH |
| M10 | FIO-ESM |
| M[11-13]* | GISS-E2-H* |
| M[14-16]* | GISS-E2-R* |
| M17 | HadGEM2-CC |

| M18 | HadGEM2-ES |
| M19 | INM-CM4 |
| M20 | IPSL-CM5A-LR |
| M21 | MIROC5 |
| M22 | MIROC-ESM |
| M23 | MPI-ESM-LR |
| M24 | MRI-CGCM3 |
| M25 | NorESM1-M |

\* These climate models provide outputs from three different physics ensembles. We treat each a separate model.

Table 2 Multi-model mean (MM) of projected changes in precipitation (%) in the four regions within the Congo River Basin (see Figure 1) for the near-term (2016-2035) and the mid-term (2046-2065) relative to the reference period of 1986-2005. The regions are identified in Figure 1. The standard deviation values across the 25 GCM-simulations are provided in parenthesis. DJF: Dec-Jan-Feb, MAM: Mar-Apr-May, JJA: Jun-Jul-Aug and SON: Sep-Oct-Nov.

| | Northern (NC) | | Equatorial (EQ) | | Southwestern (SW) | | Southeastern (SE) | |
|---|---|---|---|---|---|---|---|---|
| | RCP4.5 | RCP8.5 | RCP4.5 | RCP8.5 | RCP4.5 | RCP8.5 | RCP4.5 | RCP8.5 |
| Near future (2016-2035) | | | | | | | | |
| Annual | 1.6 (3.0) | 1.3 (2.9) | 1.3 (2.9) | 1.1 (2.7) | 1.3 (2.3) | 1.5 (2.6) | -0.4 (3.7) | 0.1 (4.2) |
| DJF | 3.3 (13.3) | 5.4 (21) | 2.0 (4.9) | 1.4 (4.7) | 1.6 (3.2) | 1.8 (4.0) | -0.3 (3.7) | 0 .04 (4.8) |
| MAM | 1.4 (4.5) | 1.1 (3.7) | 0.5 (2.9) | 0.8 (2.8) | 1.5 (4.2) | 2.5 (5.2) | -0.5 (7.8) | 0.9 (8.3) |
| JJA | 1.3 (3.3) | 0.4 (4.2) | 1.3 (4.2) | 1.3 (4.7) | -0.7 (14.6) | -0.3 (15.7) | 19.6 (32.0) | 18.7 (31.6) |
| SON | 2.3 (4.6) | 2.3 (4.7) | 1.7 (4.1) | 1.1 (4.0) | 0.9 (3.6) | 0.2 (3.8) | -0.6 (5.4) | -1 (4.8) |
| Mid-term (2046-2065) | | | | | | | | |
| Annual | 1.6 (3.8) | 1.2 (4.9) | 1.7 (3.4) | 2.4 (3.9) | 2.9 (2.9) | 3.3 (4.0) | 0.2 (5.4) | 0.3 (7.4) |
| DJF | 1.1 (15.2) | 3.9 (18.8) | 3.5 (6.3) | 5.3 (9.4) | 4.8 (5.1) | 5.4 (7.4) | 1.5 (6.4) | 1.4 (9.6) |
| MAM | 0.9 (4.4) | 0.6 (5.4) | 1.5 (3.5) | 2.4 (3.5) | 4.1 (5.1) | 6.9 (5.8) | 0.4 (9.6) | 2 (11.0) |
| JJA | 0.6 (4.3) | 0.1 (5.5) | 0.7 (5.8) | 2.2 (7.3) | -6.1 (14.8) | -5.9 (19) | 6.7 (30.6) | 9.7 (32.0) |
| SON | 3.4 (6.2) | 2.9 (7.3) | 1.3 (4.0) | 0.6 (4.1) | -0.3 (4.2) | -2.5 (4.6) | -3.2 (5.2) | -4.6 (5.8) |

857

Table 3 Multi-model mean (MM) of projected changes in runoff (%) in the four regions within the Congo River Basin for the near-term (2016-2035) and the mid-term (2046-2065) relative to the reference period of 1986-2005. The regions are identified in Figure 1. The standard deviation values across the 25 GCM-simulations are provided in parenthesis. The asterisks (*) show the degree of agreement that projected runoff > 0 in more than 50% of the ensembles. DJF: Dec-Jan-Feb, MAM: Mar-Apr-May, JJA: Jun-Jul-Aug and SON: Sep-Oct-Nov.

| | Northern (NC) | | Equatorial (EQ) | | Southwestern (SW) | | Southeastern (SE) | |
|---|---|---|---|---|---|---|---|---|
| | RCP4.5 | RCP8.5 | RCP4.5 | RCP8.5 | RCP4.5 | RCP8.5 | RCP4.5 | RCP8.5 |
| **Near future (2016-2035)** | | | | | | | | |
| Annual | 3.6 (12.1) | 2.5 (11.2) | 5.0 (7.0)* | 4.3 (6.7)* | 5.6 (4.8)* | 6.0 (5.4)* | 1.4 (12.8) | 4.2 (12.1) |
| DJF | 5.7 (13.3) | 6.0 (14.1) | 6.2 (9.8)* | 5.1 (9.5)* | 4.2 (6.1)* | 3.9 (6.4)* | 1.3 (9.3) | 2.8 (9.8) |
| MAM | 9.4 (15.0)* | 9.1 (11.1)* | 5.5 (6.3)* | 5.7 (4.9)* | 6.3 (5.1)* | 7.7 (6.3)* | 0.4 (18.4) | 4.4 (17.3) |
| JJA | 2.6 (12.1) | 1.9 (10.2) | 3.4 (6.3)* | 3.8 (6.9)* | 6.7 (5.5)* | 7.7 (7.1)* | 2.8 (20.7) | 8.3 (19.6) |
| SON | 2.8 (13.5) | 1.1 (13.3) | 4.6 (9.1)* | 3.1 (9.4) | 6.0 (6.4)* | 5.0 (6.4)* | 4.3 (10.7) | 5.0 (12.6) |
| **Mid-term (2046-2065)** | | | | | | | | |
| Annual | 1.2 (15.4) | -2.0 (17.1) | 6.3 (8.1)* | 7.2 (8.5)* | 9.9 (5.9)* | 10.4 (8.2) | 6.1 (18.8) | 8.3 (20.6) |
| DJF | 4.0 (18.0) | 1.7 (21.9) | 8.9 (11.2)* | 10.7 (14.7)* | 9.6 (7.9)* | 9.0 (12.4) | 4.7 (14.9) | 6.2 (20.3) |
| MAM | 10.1 (13.4)* | 9.5 (17.1) | 8.9 (7.1)* | 10.3 (6.2)* | 11.7 (6.1)* | 13.7 (8.0)* | 6.5 (26.2) | 9.9 (26.6) |
| JJA | -0.02 (14.5) | -2.5 (15.8) | 5.2 (9.8)* | 7.5 (11)* | 11.8 (7.1)* | 13.7 (8.6)* | 9.5 (25.9) | 14.9 (25.7) |
| SON | 0.04 (17.7) | -4.1 (19.4) | 2.5 (9.3)* | 1.1 (8.5) | 5.7 (7.2)* | 3.3 (7.7) | 5.6 (11.2)* | 3.1 (12.6) |