# Peer review of "1. Introduction"

_Hydrology and Earth System Sciences, 2016_

## Short Comment (SC1) · 23 Apr 2016

the discharge of congo river is 41000 m3/s, no 41000 km3/s

---

## Referee Comment (RC1) · Anonymous Referee #1 · 4 May 2016

This manuscript aims to elucidate the spatiotemporal variability of runoff under the climate changes the climate changes in the Congo River Basin, where long-term data are currently unavailable. Output of 25 global climate models (GCMs) for two representative concentration pathways (RCPs), combined with downscaling method, are used as input of Soil Water Assessment Tool (SWAT) model. This paper is a valuable contribution to existing literature and also suitable for the HESS scope. However, the resolutions of the topography, precipitation, temperature, land use and soil data used for the modelling with SWAT are not clear in current manuscript. A detailed description of the basin in its current stage (land use, climatic conditions, soli, topography etc.) is needed. The bias-corrected precipitation is slightly over-estimated by the statistical

method proposed by Li et al. (2010), what about the bias- temperature? Why have these two RCP scenarios (RCP 8.5 and RCP 4.5) been selected? Why not including the other two RCP scenarios as well (RCP 2.6 and RCP 6.0)? The language could be polished in various places in order to facilitate understanding.

Specific comments

Page2 Line 25: "with the projected decrease" should be "with the projected decrease in accessible runoff"??

Page3 Line41: "due the region's heavy..." should be "due to the region's heavy..."

Page6 Line92: "∼41,000km3/s" should be "∼41,000m3/s"

Page6 Line96: "a strong dry and wat seasons" or "a strong dry and wet season"? Please check it.

Page8 Line 135: "The Curve Number and ... to predict the first two". This is not clear.

Page8 Line136-137: "A power law relationship is employed to simulate to the lake area-volume-discharge". Reference?

Page9 Line164: "W m-2"; "m3/s" should be used by negative exponents. Units should be displayed using exponential formatting.

Page11 Line190: "1,450 mm/year" should be "1,450 mm"

Page11 Line194: The linear-regression slope (1.16) should be illustrated in Figure 2.

Page11 Line195: "show that" should be "shows that"

Page11 Line197: "and within the four regions identified in Figure 1 (SI Table S3)". I could not draw the conclusion from SI Table S3.

Page11 Line204-205: "Seventeen of the 30 gages show NSE greater than or equal to 0.5" This sentence is not clear.

Page12 Line212: "indicating the hydrology model's skill in simulating runoff satisfactorily over a wide range in watershed areas". ?? This sentence is not clear.

Page13 Line239: "vary" should be "varies"

Page13 Line243: delete ", with indications of spatial patterns"

Page14 Line249-250: "Most GCMs (>15) predict ... in the SE". I could not draw the conclusion from all figures in the manuscript and the supplemental material. A figure showing the percentage agreement in the increase runoff at each unit would be helpful. The percentage of ensemble members that agree on the sign of change for projected change in runoff could be calculated.

Page14 Line267: "Although northern and equatorial CRB" should be "Although the northern and equatorial CRB"

Page15 Line274: "The variability is larger NC and SE ... during the rainy seasons." This sentence is not clear.

Page15 Line278: "in SE" should be "in the SE"

Page15 Line282: "most part of EQ" should be "most part of the EQ"

Page15 Line287: "produce compatible ... and provide their ..." should be "produces compatible ... and provides their ..."

Page17 Line316-317: "Figure 8 also shows moderate increase in the SW to decrease or no-change in the SE during the rainy season (DJF)." should be "Figure 8 also shows moderate increase in the SW and decrease or no-change in the SE during the rainy season (DJF)."

Page18 Line349: "reveal" should be "reveals"

Page21 Line393: ";" should be ","

Page21 Line394-395: "... of Historical and Future Simulations of Precipitation and

Temperature in Central Africa from CMIP5 Climate Models". The initial letters should be lowercase.

Page21 Line400: "GLC2000: a" should be "GLC2000: A"

Page21 Line405: "World agriculture: towards 2015/2030: an FAO perspective" should be "World agriculture: Towards 2015/2030: An FAO perspective"

Page22 Line430: "Giorgetta, M. A., et al." Please add all authors.

Page22 Line441: "cycle: mechanisms . . ." should be "cycle: Mechanisms . . ."

Page22 Line452: "Nature Clim. Change" Please do not use the abbreviation of the journal.

Page23 Line461-464: The initial letters of the paper title should be lowercase. And please do not use the abbreviation of the journal.

Page23 Line469; Page24 Line501; Page25 Line549 and Line 558: Please do not use the abbreviation of the journal.

Page23 Line477-478: "Climate Change Projections in CESM1(CAM5) Compared to CCSM4" should be "Climate change projections in CESM1(CAM5) compared to CCSM4"

Page23 Line479: "life: a" should be "life: A"

Page23 Line482; Page24 Line503 and Line508; Page25 Line550 and Line558: "et al." Please add all authors.

Page24 Line496: "The Seasonal Evolution of the Atmospheric Circulation over West Africa and Equatorial Africa" should be "The seasonal evolution of the atmospheric circulation over west Africa and equatorial Africa"

Page24 Line508: "The Global Land Data Assimilation System" should be "The global land data assimilation system"

[Figure]

Page24 Line519: "scheme: linking" should be "scheme: Linking"

Page24 Line521: "Hydrological Cycles over the Congo and Upper Blue Nile Basins: Evaluation of General Circulation Model Simulations and Reanalysis Products" should be "Hydrological cycles over the Congo and Upper Blue Nile Basins: Evaluation of general circulation model simulations and reanalysis products"

Page25 Line550: "model: description" should be "model: Description"

Page25 Line556: "climatology: can" should be "climatology: Can"

Page25 Line 559-560: The initial letters of the paper title should be lowercase except the first word.

Page30 Line582: "water yield" is the same as "runoff"??

Page30 Line585: "show" should be "shows"

Page37 Line606 and Page40 Line624: "Figure 1A" don't exist. Please check it.

Page38: What the unit of Figure 8 is??

Some specific comments in the supplemental material

Page1 Line11: "H Lehner et al." should be "Lehner"

Page1 Line17; Page9 Line77; Page10 Line84; Page14 Line100; Page15 Line104; Page17 Line109; Page19 Line119 and 121: "Figure 1A" don't exist. Please check it.

Page9 Line76; Page10 Line83; Page11 Line89: "projected" should be "Projected"

Page13: What the unit of Figure S1 is??

Page16 Line104: "(D) Sep-Oct-Nov)." should be "(D) Sep-Oct-Nov."

Page18: The legend in each sub-figure could be deleted.

Page19 Line119: "accessible" should be "Accessible"

Page21 Line128: "GLC2000: a new" should be "GLC2000: A new"

Page21 Line 141: The initial letters of the paper title should be lowercase except the first word.

Page21 Line149: "et al." Please add all authors.

Page21 Line153-154: The initial letters of the paper title should be lowercase. And please do not use the abbreviation of the journal.

Please also note the supplement to this comment:
http://www.hydrol-earth-syst-sci-discuss.net/hess-2016-152/hess-2016-152-RC1-supplement.pdf

---

## Referee Comment (RC2) · Anonymous Referee #2 · 22 May 2016

General comments

The authors address future change in water availability in the Congo Basin. This topic is welcomed given the relative lack of research for this important region. The authors have embarked on a thorough analysis using projections from 50 climate model experiments which are bias corrected and downscaled and run through a hydrologic model. As with any impacts study of this nature, there are a host of uncertainties and methodological choices which can influence the outcomes, and it is challenging to distill information about future impacts in this context. There are also different views amongst scientists as to the best way to approach these uncertainties. However, personally I feel that the balance of emphasis on uncertainties is not quite right in this study, and

would like to see more discussion/emphasis on the climate model uncertainty (and less emphasis on the multi-model mean), as well as more analysis of observational uncertainty. Therefore I suggest major revisions. Please note that my background is in climate science so I will mainly comment on this component of the study, and do not have the relevant expertise to comment on the hydrological modelling.

1. Model uncertainty: Given the uncertainties associated with future climate, I think that the comments on the implications of the findings, particularly in the abstract and conclusions, are too strong. The "challenges" described for planners in the abstract occur only if the projections are valid (which we wont know for 50 years). The authors also make several comments about the importance of providing "details" for planners. I disagree. I think it is more important that planners are aware of uncertainties in future climate, and would benefit more from information about the range of future projections than the multi-model mean. (This is in line with a body of researchers and literature discussing such issues e.g. Weaver et al. 2011; Dessai et al. 2009; Knutti et al. 2008).

The authors have quite a large ensemble of projections from their modelling which could be made much more useful in this regard. I think it would be more useful if they commented on the size of the uncertainty and what this means for planning – are there any regions for which there is not a great deal of model uncertainty, where planners can prepare for wetter or drier conditions? Or is there also uncertainty in the direction of change which might mean that adaptive/robust planning strategies are more appropriate? How does the uncertainty from climate models compare to other uncertainties e.g. if a different hydrological model were used?

In general I think they should put more emphasis on understanding the range (e.g. Figure 7 which is useful) and less on the multi model mean (e.g. Figure 6, which should include a measure of uncertainty).

2. Observational uncertainty. The first sentence of the paper states that efforts to understand the impacts of climate change in the Congo Basin are hindered by data

availability. However, the authors do not make clear in the paper how they have overcome this, or the extent to which their findings are valid given observational uncertainty. They use an observational dataset from Sheffield et al. and (I think) use this for (a) bias correction (b) temporal downscaling to daily data and (c) sub-selecting climate models based on their ability to represent the region. Therefore, the observations might have a very important influence on their findings.

It is generally accepted (e.g. Washington et al. 2013) that availability of observed climate data in this region is a huge problem which might prevent subselection of models or bias correction. How can we say which model is more valid when there are basic questions remaining about the quantity of precipitation or where the precipitation maximum occurs? What dataset should we use assess and correct biases when there are large differences between the observational datasets used? The Sheffield et al. dataset does sound like an impressive undertaking and an important initiative but in the absence of rain gauge records it is difficult to validate it for this region, so it it still just one estimate of the observed state. I think the authors should, as a minimum, comment on the extent to which this dataset is reliable for the region and the extent to which their results might be influenced by observational uncertainties. They could also repeat their correction analyses with an alternative observational estimate and see whether this influences their results.

I am particularly concerned about the temporal downscaling to daily data, and think the authors should comment on the extent to which this is reliable, give that our understanding of day-to-day variability in precipitation/organisation of convection/meso-scale convective systems in this region is just beginning.

Specific comments

p. 4. Line 50. “require detailed information” – perhaps rephrase. If the information is not credible then details could be counterproductive. So I think better to say “would benefit from detailed information”

p. 4. Line 54. “predictive” and “forecast” – suggest change to “project” since we cannot forecast or predict on these timescales, only “project” what if under certain emissions scenarios. Suggest changing throughout.

p. 9. Line 162 – I am not sure what is meant by “medium mitigation” for RCP4.5.

p. 11. Line 190 – does this refer to bias corrected precip? If so I think this should be highlighted. Does it mean much if bias corrected precip fits with observations? Since it has been corrected using these observations?

p. 11. Line 193. “The modeled inter-annual variability among the climate models (vertical bars in Figure 2) lies within the range of the observed variability” – Looking at the figure I am not sure this is strictly true. I can see a few examples where the error bar for the models is larger than for the observed.

Figure 2 – please clarify the meaning of the modelled error bar. The caption states that it is based on the minimum and maximum range of interannual variability from the models. Is there anything to show the range of mean/climatological values for the models? And how does this compare? (similar comment for Figure 3b)

p. 13. Line 239 – why is the IQR used? What is the full range?

Figure 7 – nice figure. Is there a way to make historical plot clearer?

p. 16 Line 293 – I am not sure that MMEs reduce uncertainty. It’s more that they help explore and reveal uncertainty.

p. 16 Line 304 – I think it is overstating it to say that these models reliably simulate regional climate. We don’t have good enough observations of the regional climate to judge this. And, in any region, subselecting models is usually about taking the ones which most reliably simulate regional climate, rather than being confident that they are good enough.

p. 17 Line 326. This is quite an odd paragraph which starts of talking about implications of findings (from MM and SM?) and then finishes by saying we can reduce the range of projections from MMEs. Perhaps this should be reconsidered to suggest more nuanced conclusions about the implications of the findings which incorporate uncertainty? It is also not clear, when the author is discussing the potential to constrain model ensembles using knowledge of mechanisms that moderate the regional climate system, whether this is something they feel they have already done, or something that needs to be done. If the former, I'd suggest that their subselection procedure warrants further attention in the paper.

Figure 8 – quite a lot of information here. Could it be distilled to extract the main message?

p. 19 Line 363. "with sufficient details". I disagree. Providing details to planners may be misleading if there is too much uncertainty to give details. Better to help planners understand the uncertainty?

p. 20 Line 377. "The analyses presented in our work increase the degree of confidence in using the results for policy and management." This is unsubstantiated.

References

Dessai, S., Hulme, M., Lempert, R., & Pielke, R. (2009). Do We Need Better Predictions to Adapt to a Changing Climate? Eos, Transactions American Geophysical Union, 90(13), 111. doi:10.1029/2009EO130003

Knutti, R. (2008). Should we believe model predictions of future climate change? Philosophical Transactions. Series A, Mathematical, Physical, and Engineering Sciences, 366(1885), 4647–64. doi:10.1098/rsta.2008.0169

Weaver, C. P., Lempert, R. J., Brown, C., Hall, J. a., Revell, D., & Sarewitz, D. (2013). Improving the contribution of climate model information to decision making: the value and demands of robust decision frameworks. Wiley Interdisciplinary Reviews: Climate Change, 4(1), 39–60. doi:10.1002/wcc.202

---

## Author Comment (AC1) · 15 Aug 2016

This manuscript aims to elucidate the spatiotemporal variability of runoff under the cli- mate changes the climate changes in the Congo River Basin, where long-term data are currently unavailable. Output of 25 global climate models (GCMs) for two rep- resentative concentration pathways (RCPs), combined with downscaling method, are used as input of Soil Water Assessment Tool (SWAT) model. This paper is a valuable contribution to existing literature and also suitable for the HESS scope. However, the resolutions of the topography, precipitation, temperature, land use and soil data used for the modelling with SWAT are not clear in current manuscript. A detailed description of the basin in its current stage (land use, climatic conditions, soli, topography etc.) is needed. The bias-corrected precipitation is slightly over-estimated by the statisticalmethod proposed by Li et al. (2010), what about the bias- temperature? Why have these two RCP scenarios (RCP 8.5 and RCP 4.5) been selected? Why not including the other two RCP scenarios as well (RCP 2.6 and RCP 6.0)? The language could be polished in various places in order to facilitate understanding.

The main and supplemental text are revised to include information about the river basin's physiographic information.

We used the outputs from the Coupled Model Inter-comparison Project phase 5 (CMIP5). The CMIP5 experimental design guidelines [*Taylor et al.*, 2012] recommend the use of RCP4.5 and RCP8.5 simulations as they provide high-interest information about future climate change. We followed these guidelines and decided to present only these two future scenarios.

Overall temperature bias is 0.15 °C. An assessment of how climate models simulate precipitation and temperature in the Central African region is presented in a separate paper [*Aloysius et al.*, 2016]. Figure 5 e-h in that manuscript (see below) compares the overall biases in temperature. I have also attached the full manuscript for easy reference.

[Figure]

Figure 1 Comparison of observed and bias-corrected annual (e,f) and monthly (g-h) climatology of temperature in the historical period. The units are °C.

Specific comments

Page2 Line 25: "with the projected decrease" should be "with the projected decrease in accessible runoff"??

Revised as suggested

Page3 Line41: "due the region's heavy…" should be "due to the region's heavy…"

Revised as suggested

Page6 Line92: "~41,000km3/s" should be "~41,000m3/s" - corrected

Page6 Line96: "a strong dry and wat seasons" or "a strong dry and wet season"? Please check it. - corrected

Page8 Line 135: "The Curve Number and … to predict the first two". This is not clear.

Revised the text and clarified

Page8 Line136-137: "A power law relationship is employed to simulate to the lake area-volume-discharge". Reference?

Reference included

Page9 Line164: "W m-2"; "m3/s" should be used by negative exponents. Units should be displayed using exponential formatting.

All units have been revised as suggested

Page11 Line190: "1,450 mm/year" should be "1,450 mm"

Revised as suggested

Page11 Line194: The linear-regression slope (1.16) should be illustrated in Figure 2.

Regression details added to Figure 2.

Page11 Line195: "show that" should be "shows that" – revised as suggested

Page11 Line197: "and within the four regions identified in Figure 1 (SI Table S3)". I could not draw the conclusion from SI Table S3.

SI Figure S2 shows the seasonal precipitation. SI Table S3 provides the mean values within the regions identified in Figure S1. The text has been modified.

Page11 Line204-205: "Seventeen of the 30 gages show NSE greater than or equal to 0.5" This sentence is not clear.

Literature suggests that model simulations can be considered satisfactory if Nash-Sutcliff values are ≥ 0.5. This is clarified in the text and a reference is added to support [*Moriasi et al.*, 2007].

Page12 Line212: "indicating the hydrology model's skill in simulating runoff satisfacto- rily over a wide range in watershed areas". ?? This sentence is not clear.

Catchment areas of all gages considered vary between 5,000 to 900,000 km$^2$ and encompass a range climatic regions on both sides of the equator. In this context, our hydrology model performance is satisfactory. The text has been modified to clarify.

Page13 Line239: "vary" should be "varies" – text revised

Page13 Line243: delete ", with indications of spatial patterns" – sentence revised

Page14 Line249-250: "Most GCMs (>15) predict . . . in the SE". I could not draw the conclusion from all figures in the manuscript and the supplemental material. A figure showing the percentage agreement in the increase runoff at each unit would be helpful. The percentage of ensemble members that agree on the sign of change for projected change in runoff could be calculated.

A figure is added to reflect the author's comments.

Page14 Line267: "Although northern and equatorial CRB" should be "Although the northern and equatorial CRB" – sentence revised

Page15 Line274: "The variability is larger NC and SE . . . during the rainy seasons." This sentence is not clear.

The sentence is revised to clarify.

Page15 Line278: "in SE" should be "in the SE" – sentence revised

Page15 Line282: "most part of EQ" should be "most part of the EQ" – sentence revised

Page15 Line287: "produce compatible . . . and provide their . . ." should be "produces compatible . . . and provides their . . ." – revisions are incorporated in the revised version

Page17 Line316-317: "Figure 8 also shows moderate increase in the SW to decrease or no-change in the SE during the rainy season (DJF)." should be "Figure 8 also shows moderate increase in the SW and decrease or no-change in the SE during the rainy season (DJF)." - revised

Page18 Line349: "reveal" should be "reveals" Page21 Line393: ";" should be "," - revised

Page21 Line394-395: ". . . of Historical and Future Simulations of Precipitation and Temperature in Central Africa from CMIP5 Climate Models". The initial letters should be lowercase. - revised

Page21 Line400: "GLC2000: a" should be "GLC2000: A" - revised

Page21 Line405: "World agriculture: towards 2015/2030: an FAO perspective" should be "World agriculture: Towards 2015/2030: An FAO perspective" - revised

Page22 Line430: "Giorgetta, M. A., et al." Please add all authors. – This reference has 39 co-authors, therefore, we decided to include only the first author.

Page22 Line441: "cycle: mechanisms . . ." should be "cycle: Mechanisms . . ." - revised

Page22 Line452: "Nature Clim. Change" Please do not use the abbreviation of the journal. - revised

Page23 Line461-464: The initial letters of the paper title should be lowercase. And please do not use the abbreviation of the journal.

revised

Page23 Line469; Page24 Line501; Page25 Line549 and Line 558: Please do not use the abbreviation of the journal. –

revised

Page23 Line477-478: "Climate Change Projections in CESM1(CAM5) Compared to CCSM4" should be "Climate change projections in CESM1(CAM5) compared to CCSM4" - revised

Page23 Line479: "life: a" should be "life: A" - revised

Page23 Line482; Page24 Line503 and Line508; Page25 Line550 and Line558: "et al." Please add all authors. – These references have more than 10 co-authors

Page24 Line496: "The Seasonal Evolution of the Atmospheric Circulation over West Africa and Equatorial Africa" should be "The seasonal evolution of the atmospheric circulation over west Africa and equatorial Africa" - revised

Page24 Line508: "The Global Land Data Assimilation System" should be "The

global land data assimilation system" - revised

Page24 Line519: "scheme: linking" should be "scheme: Linking" - revised

Page24 Line521: "Hydrological Cycles over the Congo and Upper Blue Nile Basins: Evaluation of General Circulation Model Simulations and Reanalysis Products" should be "Hydrological cycles over the Congo and Upper Blue Nile Basins: Evaluation of general circulation model simulations and reanalysis products" - revised

Page25 Line550: "model: description" should be "model: Description" - revised

Page25 Line556: "climatology: can" should be "climatology: Can" - revised

Page25 Line 559-560: The initial letters of the paper title should be lowercase except the first word. - Revised

Page30 Line582: "water yield" is the same as "runoff"?? –revised the figure caption

Page30 Line585: "show" should be "shows" - revised

Page37 Line606 and Page40 Line624: "Figure 1A" don't exist. Please check it. Page38: What the unit of Figure 8 is??

Revised

Some specific comments in the supplemental material Page1 Line11: "H Lehner et al." should be "Lehner"

Revised

Page1 Line17; Page9 Line77; Page10 Line84; Page14 Line100; Page15 Line104; Page17 Line109; Page19 Line119 and 121: "Figure 1A" don't exist. Please check it.

Revised

Page9 Line76; Page10 Line83; Page11 Line89: "projected" should be "Projected"

Revised

Page13: What the unit of Figure S1 is??

The units are in mm month$^{-1}$. The figure caption is revised.

Page16 Line104: "(D) Sep-Oct-Nov)." should be "(D) Sep-Oct-Nov." Page18: The legend in each sub-figure could be deleted.

Revised. The square dots show projections of a subset of models outputs.

Page19 Line119: "accessible" should be "Accessible" - Revised

Page21 Line128: "GLC2000: a new" should be "GLC2000: A new" – revised

Page21 Line 141: The initial letters of the paper title should be lowercase except the first word. – revised a suggested

Page21 Line149: "et al." Please add all authors.

This reference has 15 co-authors

Page21 Line153-154: The initial letters of the paper title should be lowercase. And please do not use the abbreviation of the journal.

All references have been revised as suggested

**References**

Aloysius, N., J. Sheffield, J. E. Saiers, H. Li, and E. F. Wood (2016), Evaluation of historical and future simulations of precipitation and temperature in Central Africa from CMIP5 climate models, *Journal of Geophysical Research - Atmospheres*, *121*(1), 130-152, doi: 10.1002/2015JD023656.

Moriasi, D. N., J. G. Arnold, M. W. Van Liew, R. L. Bingner, R. D. Harmel, and T. L. Veith (2007), Model evaluation guidelines for systematic quantification of accuracy in watershed simulations, *Transactions of the ASABE*, *50*(3), 885-900.

Taylor, R. Stouffer, and G. Meehl (2012), An overview of CMIP5 and the experiment design, *Bulletin of the American Meteorological Society*, *93*(4), 485, doi: 10.1175/BAMS-D-11-00094.1.

[Figure]

[Figure]

**Journal of Geophysical Research: Atmospheres**

**RESEARCH ARTICLE**
10.1002/2015JD023656

**Key Points:**
- Evaluation of precipitation and temperature simulation in global climate models in central Africa
- Climate models exhibit limited skills in precipitation simulations
- Climate model selection for regional impact studies is evaluated

**Supporting Information:**
- Figure S1
- Figure S2
- Figure S3
- Figure S4
- Figure S5
- Figure S6
- Figure S7
- Figure S8
- Figure S9
- Figure S10
- Figure S11
- Figure S12
- Figure S13
- Figure S14
- Figure S15

**Correspondence to:**
N. R. Aloysius,
aloysius.1@osu.edu

**Citation:**
Aloysius, N. R., J. Sheffield, J. E. Saiers, H. Li, and E. F. Wood (2016), Evaluation of historical and future simulations of precipitation and temperature in central Africa from CMIP5 climate models, *J. Geophys. Res. Atmos., 121*, 130–152, doi:10.1002/2015JD023656.

Received 10 MAY 2015
Accepted 31 OCT 2015
Accepted article online 4 NOV 2015
Published online 8 JAN 2016

**Evaluation of historical and future simulations of precipitation and temperature in central Africa from CMIP5 climate models**

Noel R. Aloysius[1,2], Justin Sheffield[3], James E. Saiers[1], Haibin Li[4], and Eric F. Wood[3]

[1]School of Forestry and Environmental Studies, Yale University, New Haven, Connecticut, USA, [2]Now at Department of Food, Agricultural, and Biological Engineering and Department of Evolution, Ecology, and Organismal Biology, Ohio State University, Columbus, Ohio, USA, [3]Department of Civil and Environmental Engineering, Princeton University, Princeton, New Jersey, USA, [4]Department of Earth and Planetary Sciences, Rutgers University, Piscataway, New Jersey, USA

**Abstract** Global and regional climate change assessments rely heavily on the general circulation model (GCM) outputs such as provided by the Coupled Model Intercomparison Project phase 5 (CMIP5). Here we evaluate the ability of 25 CMIP5 GCMs to simulate historical precipitation and temperature over central Africa and assess their future projections in the context of historical performance and intermodel and future emission scenario uncertainties. We then apply a statistical bias correction technique to the monthly climate fields and develop monthly downscaled fields for the period of 1948–2099. The bias-corrected and downscaled data set is constructed by combining a suite of global observation and reanalysis-based data sets, with the monthly GCM outputs for the 20th century, and 21st century projections for the medium mitigation (representative concentration pathway (RCP)45) and high emission (RCP85) scenarios. Overall, the CMIP5 models simulate temperature better than precipitation, but substantial spatial heterogeneity exists. Many models show limited skill in simulating the seasonality, spatial patterns, and magnitude of precipitation. Temperature projections by the end of the 21st century (2070–2099) show a robust warming between 2 and 4°C across models, whereas precipitation projections vary across models in the sign and magnitude of change (−9% to 27%). Projected increase in precipitation for a subset of models (single model ensemble (SME)) identified based on performance metrics and causal mechanisms are slightly higher compared to the full multimodel ensemble (MME) mean; however, temperature projections are similar between the two ensemble means. For the near-term (2021–2050), neither the historical performance nor choice of models is related to the precipitation projections, indicating that natural variability dominated any signal. With fewer models, the "blind" MME approach will have larger uncertainties in future precipitation projections compared to projections by the SME models. We propose the latter a better approach in regions that lack quality climate observations. Our analyses also show that the choice of model and emission scenario dominate the uncertainty in precipitation projections, whereas the emission scenario dominates the temperature projections. Although our analyses are done for central Africa, the final Bias-Corrected Spatially Downscaled data set is available for global land areas. The framework for climate change assessment and the data will be useful for a variety of climate assessment, impact, and adaptation studies.

**1. Introduction**

Adapting to variability and change in climate is one of the challenges that countries in sub-Saharan Africa face in the 21st century [*Boko et al.*, 2007; *Collier et al.*, 2008]. This challenge arises, in part, from the need to respond to the potentially detrimental effects of climate change on freshwater availability, food production, and other ecosystem services [*Bruinsma*, 2003; *Lobell et al.*, 2008; *Nkem et al.*, 2010; *Wilkie et al.*, 1999]. The continent is also important because of the tropical rainforests in central Africa (CA) that are a large carbon sink and play a role in mediating the impact of greenhouse gas emissions [*Baccini et al.*, 2008; *Fisher et al.*, 2013].

The climate of CA is complex. Processes that modulate precipitation over the region include isolated convection cells to mesoscale convective systems (MCSs) that span a range of spatial and temporal scales and interact with regional and global circulatory patterns [*Houze*, 2004; *Jackson et al.*, 2009; *Laing and Fritsch*, 1993; *Pearson et al.*, 2014]. Another phenomena that characterize the region is the Intertropical Convergence Zone (ITCZ) where the warm and moist trade winds converge to form a zone of increased convection and

©2015. American Geophysical Union.
All Rights Reserved.

cloudiness [*Schneider et al.*, 2014]. As such, the north-south movement of the ITCZ, its strength, and position strongly influence processes that generate rainfall in the CA, including the MCS [*Dezfuli and Nicholson*, 2013; *Farnsworth et al.*, 2011]. Further, topographic features (e.g., Cameroon and Rift Valley highlands in the northern and eastern CA, respectively) also trigger convection, which then propagates downstream and develops into larger MCS and contributes to the observed pattern of maximum mid-CA precipitation [*Janiga and Thorncroft*, 2014]. Other factors that influence CA's precipitation include departures in sea surface temperatures (SSTs) in the equatorial and south Atlantic, the African Easterly Jets and the Tropical Easterly Jets [*Farnsworth et al.*, 2011; *Hastenrath*, 1991; *Nicholson and Dezfuli*, 2013]. Moisture sourced from the southern Atlantic and Indian Oceans facilitated by the easterlies also brings precipitation to land, although their effects diminish farther onto the continent [*Giannini et al.*, 2008; *Nicholson and Grist*, 2003; *Suzuki*, 2011].

Few studies have assessed the impacts of climate variability and change on CA [*Intergovernmental Panel on Climate Change* (*IPCC*), 2007; *Washington et al.*, 2013]. Analysis of the interactions between the region's hydrological cycle and the functioning of its ecosystems are particularly lacking, owing to the dearth of reliable, long-term observations and projections of climate. As African countries strive to expand agriculture, increase hydropower generation, and improve freshwater supply and sanitation, it is essential that reliable climate change scenarios are made available for systematic evaluation of all plausible options [*Bruinsma*, 2003; *DeFries et al.*, 2010; *IPCC*, 2007; *United Nations Environment Programme*, 2006]. To this end, our goal is to provide an evaluation of how GCMs simulate the main drivers of the hydrological cycle in CA: precipitation and temperature. We also evaluate approaches for climate model selection and provide a set of climate change scenarios for the 21st century that can be used in climate change impact analyses related to water and natural resources.

We analyze monthly precipitation and temperature fields simulated by 25 GCMs that were submitted to the fifth phase of the Climate Model Intercomparison Project (CMIP5) in support of the Intergovernmental Panel on Climate Change (IPCC) Fifth Assessment. We compare GCM simulations with observed climate fields [*Mitchell and Jones*, 2005; *Sheffield et al.*, 2006] to document model performance in reproducing the historical climate, including interannual variability. We evaluate the relationship between precipitation and SST fields to elucidate the physical mechanisms that control precipitation. We then correct the biases by applying a statistical bias correction method [*Li et al.*, 2010] to monthly climate fields. The future projections are assessed in terms of multimodel mean changes and their uncertainties, as a function of the emission scenario, time period, and performance of models for the region.

**2. Data Sets and Methods**

**2.1. Study Area and Historical/Observed Data Set**

The historical surface climate fields are based on a combination of the National Centers for Environmental Prediction-National Center for Atmospheric Research (National Centers for Environmental Prediction (NCEP)–National Center for Atmospheric Research (NCAR)) reanalysis and a suite of remotely sensed and ground-based data sets. The development of this data set is described in *Sheffield et al.* [2006]. In brief, the data set is developed by using submonthly variability in the NCEP-NCAR reanalysis with corrections for biases applied at monthly time scale. We used precipitation, temperature, and diurnal temperature range (DTR), which are available at three hourly, daily, and monthly time steps from 1948 to present at a horizontal resolution of 1° latitude/longitude. The data set has been similarly used by others in studies of regional climate outside of CA (e.g., *Bohn et al.* [2007], *Demaria et al.* [2012], *Sheffield et al.* [2010], and *Wang et al.* [2011]). In addition, we use the University of East Anglia Climate Research Unit gridded monthly time series (1901–2008) of precipitation and temperature fields [*Mitchell and Jones*, 2005], and Hadley Center sea surface temperature (SST) data set (1870–2012) [*Hurrell et al.*, 2008], in order to compare the model simulations during the historical period in the CA.

We focus our analysis on the CA region that covers the Congo River basin (CB, 15°S to 15°N and 10°E to 36°E; Figure 1). The CB boundary extends to 10°N, but we expanded the latitude range to 15°N to account for the full extent of the movement of ITCZ in the northern hemisphere. Several ecosystems lie within the region, including evergreen tropical forests that comprise nearly 45% of the CB, vast freshwater wetlands in the central and southeastern parts of the basin, and grasslands in the peripheries of the basin [*Laporte et al.*, 2007; *Revenga et al.*, 1998]. The study area is divided into four subregions: Northern (NC, 5°N–1°5N and 10°E–36°E), equatorial west (EQW, 5°S–5°N and 10°E–27°E), equatorial east (EQE, 5°S–5°N and 27°E–36°E), southwestern

[Figure]

**Figure 1.** Study area (15°S–15°N, 10°E–35°E). The Congo River Basin boundary, extent of rainforest, and locations of large lakes within the Congo River Basin are also shown.

(SW, 5°S–15°S and 10°E–25°E), and southeastern (SE, 5°S–15°S and 25°E–36°E). A unimodal rainy season dominates the NC (June-July-August (JJA)), and SW, and SE (December-January-February (DJF)); the equatorial regions are characterized by bimodal rainy seasons: March-April-May (MAM) and September-October-November (SON). Most of the rainforest is concentrated in the EQW (Figure 1). Parts of EQE lie along the East African Rift system and the elevation here is around 1000 m above sea level. The southern region (below 5°S) is divided into two (SW and SE) in order to distinguish the presence of extensive lakes and wetlands in the SE.

**2.2. Twentieth and 21st Century GCM Simulations**

We selected outputs from 25 GCMs archived at the Program on Climate Model Diagnosis and Intercomparison (PCMDI) website for monthly values of total precipitation and monthly values of mean, minimum, and maximum surface air temperature (Table 1). We also obtained monthly SST data for all but one model (M10). These data archives and the CMIP5 experimental design are documented in *Taylor et al.* [2012]. In general, the CMIP5 models have higher horizontal resolution than models from the previous CMIP phases and include more comprehensive treatments of physical processes, such as interactive vegetation, and external forcings, such as aerosols and land cover. All the historical simulations are for the period of 1850 to 2005. The monthly minimum and maximum temperatures are monthly means of the daily values. Precipitation includes liquid and solid phases from all types of precipitation simulated in the models. The historical simulations are forced

[Figure]
 **Journal of Geophysical Research: Atmospheres** 10.1002/2015JD023656

**Table 1.** List of Climate Models, Experiments, Ensemble Members, and Variables Whose 20th and 21st Century Simulations Are Analyzed in This Study

| No | Model Name | Institute | Horizontal Resolution (latitude × longitude) | No. of Ensembles | Key Reference |
|---|---|---|---|---|---|
| M1 | ACCESS1–3 | Commonwealth Scientific and Industrial Research Organisation and Bureau of Meteorology, Australia | 1.25 × 1.875 | r1i1p1 | *Bi* [2012] ES[a] = 3.50°C and TS[b] = 1.60°C |
| M2 | BCC-CSM1-1 | Beijing Climate Center | 2.8 × 2.8 | r1i1p1 | *Wu et al.* [2013]; *Xin et al.* [2013] ES = 2.8°C and TS = 1.85°C |
| M3 | BNU-ESM | GCESS, BNU, Beijing, China | 2.8 × 2.8 | r1i1p1 | ES = 4.1°C and TS = 2.6°C |
| M4 | CanESM2 | Canadian Center for Climate Modeling and Analysis | 2.8 × 2.8 | hist.and RCP45 – r[1–5]i1p1, RCP85 – r[1–3]i1p1 | *Arora et al.* [2011] ES = 3.7°C and TS = 2.3°C |
| M5 | CCSM4 | National Center for Atmospheric Research | 0.9 × 1.25 | r[1–5]i1p1 | *Gent et al.* [2011] ES = 3.2°C and TS = 1.73°C |
| M6 | CESM1-CAM5 | National Center for Atmospheric Research, (NCAR) Boulder, CO, USA | 0.9 × 1.25 | r[1–3]i1p1 | *Meehl et al.* [2013] ES = 4.1°C and TS = 2.33°C |
| M7 | CNRM-CM5 | Centre National de Recherches Meteorologiques | 1.4 × 1.4 | hist. and RCP45 – r1i1p1, RCP85 – r[1,2,4,6,10]i1p1 | *Voldoire et al.* [2012] ES = 3.3°C and TS = °C |
| M8 | CSIRO-Mk3-6-0 | CSIRO Marine and Atmospheric Research | 1.8 × 1.8 | hist and RPC85 – r[1–5]i1p1, RCP45 – r[1,3,5]i1p1 | *Rotstayn et al.* [2010] ES = 4.1°C and TS = °C |
| M9 | EC-EARTH | European Earth System Model | 1.1 × 1.125 | r[2,8,9]i1p1 | *Hazeleger et al.* [2012]; *Hazeleger et al.* [2010] ES = 2.56°C |
| M10 | FIO-ESM | The First Institution of Oceanography, SOA, Qingdao, China | 2.76 × 2.80 | r[1–3]i1p1 | *Bao et al.* [2012] |
| M11, M12, M13 | GISS-E2-H | Goddard Institute for Space Studies, New York, NY | 2.0 × 2.5 | hist and RCP45 – r[1–3]i1p[1–3], RCP85 – r1i1p[1–3] | *Schmidt et al.* [2006] ES = 2.3°C and TS = 1.7°C |
| M14, M15, M16 | GISS-E2-R | Goddard Institute for Space Studies | 2.0 × 2.5 | hist and RCP45 – r[1–3]i1p[1–3], RCP85 – r1i1p[1–3] | *Schmidt et al.* [2006] ES = 2.1°C and TS = 1.5°C |
| M17 | HadGEM2-CC | Met Office Hadley Centre, Fitzroy Road, Exeter, Devon, EX1 3PB, UK, | 1.25 × 1.875 | r1i1p1 | *Collins et al.* [2011] |
| M18 | HadGEM2-ES | Met Office Hadley Centre | 1.875 × 1.25 | r[2–4]i1p1 | *Collins et al.* [2011] ES = 4.6°C and TS = 2.5°C |
| M19 | INM-CM4 | Institute for Numerical Mathematics, Moscow, Russia | 2.5 × 2.0 | r1i1p1 | *Volodin et al.* [2010] ES = 2.1°C and TS = °C |
| M20 | IPSL-CM5A-LR | Institut Pierre Simon Laplace, Paris, France | 1.9 × 3.75 | r[1–3]i1p1 | *Dufresne et al.* [2013] ES = 3.6°C and TS = 2.1°C |
| M21 | MIROC5 | Japan Agency for Marine-Earth Science and Technology (JAMSTEC), Atmosphere and Ocean Research Institute, The University of Tokyo, and National Institute for Environmental Studies, Japan | 1.4 × 1.4 | r[1–3]i1p1 | *Watanabe et al.* [2010] ES = 2.7°C and TS = °C |
| M22 | MIROC-ESM | Japan Agency for Marine-Earth Science and Technology (JAMSTEC), Atmosphere and Ocean Research Institute, The University of Tokyo, and National Institute for Environmental Studies, Japan | 2.8 × 2.8 | r1i1p1 | *Watanabe et al.* [2011] ES = 4.7°C and TS = °C |
| M23 | MPI-ESM-LR | Max Planck Institute for Meteorology | 1.80 × 1.80 | r[1–3]i1p1 | *Giorgetta et al.* [2013]; *Notz et al.* [2013] ES = 3.6°C and TS = 2.0°C |
| M24 | MRI-CGCM3 | Meteorological Research Institute | 1.10 × 1.10 | r1i1p1 | *Yukimoto et al.* [2006] ES = 2.6°C and TS = °C |
| M25 | NorESM1-M | Norwegian Climate Centre | 1.875 × 2.50 | r1i1p1 | *Bentsen et al.* [2013]; *Iversen et al.* [2012] ES = 2.8°C and TS = 1.4°C |

[a]Equilibrium climate sensitivity.
[b]Transient climate response.

by anthropogenic changes ($CO_2$ and non-$CO_2$ greenhouse gases, aerosols, and land cover), and in most cases, each GCM provides multiple ensembles to represent model internal variability.

We use model output for two emission scenarios (the representative concentration pathways, RCPs), RCP45 and RCP85, for the period of 2006–2100. The RCP45 is a stabilization scenario in which the total radiative forcing is stabilized before 2100 by employing a range of technologies and policies to reduce greenhouse gas (GHG) and aerosol emissions. The RCP85, on the other hand, is characterized by increasing GHG and aerosol emissions leading to high concentrations beyond 2100. The approximate concentrations of radiatively active gases and aerosols for RCP45 and RCP85 by 2100 are 650 and 1370 ppm $CO_2$ equivalent, respectively [*International Institute for Applied Systems Analysis*, 2009; *Moss et al.*, 2010]. In terms of land use change, the RCP45 assumes land management strategies that result in expansion of forests and reduction in crop and pastureland for carbon storage, whereas the RCP85 assumes considerable loss of forest area and concomitant increases in crop and pastureland [*Lawrence et al.*, 2012].

One to five ensembles of each GCM were evaluated as available, with multiple ensembles available for 17 of the 25 models (Table 1). We use the data reference syntax of *Taylor et al.* [2009a, 2009b] to refer to the ensemble members: $r<N>i<M>p<L>$, where $r$ is the initial condition (time), $i$ is the initialization method, and $p$ is the perturbed physics ensemble, while $N$, $M$, and $L$ represent integer values that identify the different ensemble members from the same GCM (Table 1). Historical runs in the CMIP5 models are initialized (different $i$) from different times in the preindustrial control run; these control runs usually span a 500 year simulation and are characterized by the prescription of nonevolving, well-mixed atmospheric gases (including $CO_2$) and unperturbed land use [*Taylor et al.*, 2009a, 2009b]. Multiple ensembles of a single GCM were computed for a single initialization method but different initial conditions (different $r$ but the same $i$). We treat the three physics ensembles of GISS-E2-H and GISS-E2-R as six different models ($p1$, $p2$, and $p3$).

The horizontal resolution varies between GCMs, and we assigned the models into two groups: (a) high-resolution models less than 2° latitude/longitude (MMEhires, $n=11$) and (b) low-resolution models greater than 2° (MMElores, $n=14$). We used bilinear interpolation to regrid all the GCM climate fields to match the horizontal resolution of the observations (1°). These preprocessed outputs for the period of 1901–2100 are used in the analysis presented here.

**2.3. Bias Correction and Downscaling**

We applied the equidistant quantile-based mapping method (EDCDF) developed by *Li et al.* [2010] to bias correct the monthly precipitation and temperature fields. For each grid point, the probability density function (pdf) for the monthly model-simulated climate field was mapped to the observed pdf for the historical period (1951–2000), thus matching all statistical moments between the GCM simulations and observations. Standard quantile-based mapping methods (CDF) then apply the mapping to the projections assuming that the biases in the historical period persist in the future as well [*Ines and Hansen*, 2006; *Wood et al.*, 2004]. The EDCDF method extends the CDF method by taking into account changes in the model pdf between the historic and the future periods. *Li et al.* [2010] compared the performance of both methods and showed that the EDCDF is more efficient in reducing the model biases, which may be particularly important if changes in the extreme values (at monthly scale) occur in the future. Further comparison with several CDF-type methods, including *Li et al.* [2010], are provided in *Watanabe et al.* [2012].

To avoid interpolation between values of the empirical CDFs, parametric distributions were fitted to the precipitation and temperature data at each grid point, following *Li et al.* [2010]. A four-parameter beta function was fitted to the temperature data. The distribution range parameters were taken as the extreme values from the data extended by half of one standard deviation at each grid point. The distribution shape parameters were then determined by the method of maximum likelihood estimation. A two-parameter mixed gamma distribution was fitted to the precipitation fields to allow for periods of no rain. We applied this method—first to correct the biases in the historical period (1901–2000) and then to the future period (2001–2100)—to all the ensemble members of each GCM.

**2.4. Evaluation of Model Performance and Future Projections**

We used the mean square error (MSE) and spatial correlation to evaluate the model performance. The MSE and spatial skill score (SS) between a model-simulated and observed patterns of a field (precipitation or temperature) are defined as

$$\text{MSE} = \frac{1}{N} \sum_{i=1}^{N} (m_i - o_i)^2 \qquad (1)$$

and

$$\text{SS} = 1 - \frac{\text{MSE}(m, o)}{\text{MSE}(\overline{o}, o)} \qquad (2)$$

where $m$, $o$, and $\overline{o}$ are the simulated, observed, and mean of the observed fields, respectively, and $N$ is the number of spatial points [*Murphy*, 1988; *Pierce et al.*, 2009]. Further, the *MSE* can be decomposed as follows:

$$\text{MSE}(m, o) = (\overline{m} - \overline{o})^2 + \sigma_m^2 + \sigma_o^2 - 2\sigma_{m,o} \qquad (3)$$

where $\sigma_m^2$ and $\sigma_o^2$ are the variances of simulated and observed fields, respectively, and $\sigma_{m,o}$ is the covariance of simulated and observed fields. Because $\sigma_{m,o} = \sigma_m \sigma_o R_{m,o}$, the MSE can be rewritten as

$$\text{MSE}(m, o) = (\overline{m} - \overline{o})^2 + \sigma_m^2 + \sigma_o^2 - 2\sigma_m \sigma_o R_{m,o} \qquad (4)$$

where $R_{m,o}$ is the spatial pattern correlation coefficient—the potential skill of a model in replicating patterns of departure from a climatological mean [*Wilks*, 2006]. The spatial pattern correlation is defined as

$$R_{m,o} = \frac{\frac{1}{N} \sum_{i=1}^{N} (m_i - \overline{m})(o_i - \overline{o})}{\sigma_m \sigma_o} \qquad (5)$$

Subsequently, the *SS* can be expressed as a function of $R_{m,o}$, such that

$$\text{SS} = R_{m,o}^2 - \left[ R_{m,o} - \left( \frac{\sigma_m}{\sigma_o} \right) \right]^2 - \left[ \frac{(\overline{m} - \overline{o})}{\sigma_o} \right]^2 \qquad (6)$$

The three terms of *SS* in equation (6) quantify the influences of spatial pattern correlation, systematic biases, and nonsystematic biases in the model simulations compared to an observed climate field [*Murphy*, 1988; *Wilks*, 2006]. A model that exactly reproduces the observation will have a skill score of 1; a model that predicts the regional average well, but without any spatial features will have a skill score of 0. We used the root-mean-square difference (RMSD–square root of MSE) and the standard deviation (sd) of monthly, seasonal, and annual fields to compare biases and interannual variability with observations in the historical period.

The performance in the historical period and projections are evaluated for individual models and for combination of multimodel ensembles (MME: $n = 25$ and SME: $n = 5$, M6, M7, M18, M23, and M24). The selection of SME is described in the results section.

**3. Results**

**3.1. Precipitation**

**3.1.1. Historical Patterns**

The mean annual precipitation fields for the historical period (1971–2000) simulated by the 25 models exhibit the largest absolute differences in the equatorial region (Figures 2a–2d and Figure S1 in the supporting information). The RMSD between simulated and observed (Figure 2d) annual precipitation varies from 250 mm to 800 mm over the study area. The equatorial region (5°S–5°N) exhibits the largest (300 to 1000 mm) absolute biases among the models; however, the relative biases—measured as the coefficient of variation of RMSD ($\text{CV}_{\text{RMSD}}$)–are about the same (~0.4) over the study area. The relative biases are large during nonrainy seasons on all sub regions shown in Figure 1 (Figure S2 in the supporting information). The RMSD and the spatial patterns values do not vary much among models of the same family (M11–M16 and M17–M18; Table 1 and Figure S1 in the supporting information). The pattern correlation between the observed and simulated annual precipitation varies from 0.60 to 0.94, while the SS varies from −2.00 to 0.70. Thirteen model simulations result in negative SS, largely as a result of nonsystematic biases. Among the individual models, M17 shows the highest SS. The highest skill scores are obtained for the MMEhires average followed by the all-model average (Figures 2a and 2b). The SS for the five models in the SME (M6, M7, M18, M23, and M24) varies between 0.5 and 0.7. The performance in the historical period and other characteristics in the SME will be discussed in section 4. The seasonal analysis of the 1971–2000 climatology reveals that most of the models overestimate precipitation to a greater degree during DJF and SON (average RMSD 180 mm) than MAM and JJA

[Figure]

**Figure 2.** Average annual precipitation (mm/yr), temperature (°C), and DTR (°C) over the study area for the historical period of 1971–2000. For precipitation (a) all-, (b) high-resolution, and (c) low-resolution GCM averages and (d) selected (M6, M7, M18, M23, and M24) and (f) observed values. (g–r) Show the same for temperature and DTR, respectively.

(average RMSD = 150 mm). Notably, the overprediction of seasonal wetness is more pronounced in southeastern CA in DJF and in eastern equatorial CA in SON, which are also the dominant rainy seasons in these areas (Figures S3–S6 in the supporting information). The seasonal SS values are comparable with the annual values. Notably, the SS for the MME (0.7–0.85) and MMEhires (0.5–0.9) are within the range of the five models in the SME.

A prominent feature in the central African rain belt is the strong bimodal cycle near the equator (MAM and SON seasons) coinciding with the northward and southward movements of the ITCZ (Figure 3b). Many models, notably M6, M7, M18, M23, and M24, represent the bimodality and the intensity of precipitation satisfactorily (Figure S7 in the supporting information). Over the land surface, the northward movement of the heavy precipitation region starts in January near 10°S and reaches the peak near 10°N in August. The southward

[Figure]

**Figure 3.** Zonally averaged monthly precipitation (mm/month) along the latitudes, averaged over the study region (10.5°E to 35.5°E), (a) all-GCM average and (b) observed values. The black solid line in Figures 3a and 3b indicates the month of maximum precipitation along the latitudes for the all-GCM average and observed precipitation. The red and grey solid lines and dotted lines in Figure 3a indicate the same for high-resolution (high-res), low-resolution (low-res), and individual GCMs, respectively.

movement starts immediately and continues until December and brings heavier precipitation than the north-ward phase as shown in the observations (Figure 3b). The average monthly precipitation along the northward phase (January to June) of the ITCZ (black solid line in Figure 3b) is 170 mm, which is significantly different from the monthly precipitation of 190 mm during the wetter southward phase ($p < 0.05$). Models M1, M4, M9–10, M17–18, M21, M23–24 (see Table 1), and MMEhires capture the observed asymmetry ($p < 0.05$); the asymmetry is reversed in M10. The average difference between the northward and southward phases among the above models is 20 mm. Several models (e.g., M1 and M21) yield too much precipitation during the southward phase, while others (e.g., M2, M11, and M19) poorly reproduce the asymmetric precipitation structure. The pattern correlations of this space-time structure (Figure 3 and Figure S7 in the supporting infor-mation) are generally high (0.83–0.97, > 0.9 for the models in the SME) due to the models' ability to repro-duce the strong seasonality, which is characteristic in this region.

[Figure]

**Figure 4.** Taylor diagram showing the comparison of monthly (a) precipitation, (b) temperature, and (c) diurnal temperature range simulations over the region with observations based on 1971–2000 climatology. Multimodel ensembles (all-GCM, high-res: high resolution and low-res: low resolution) and selected GCMs (M6, M7, M18, M23, and M24) described in the text are marked in colors. All other models are shown as open circles.

**◈AGU** **Journal of Geophysical Research: Atmospheres** 10.1002/2015JD023656

[Figure]

**Figure 5.** Comparison of observed and bias-corrected annual and monthly climatology of (a–d) precipitation and (e–h) temperature for the period of 1941–1970. A difference map is shown in Figure S9 in the supporting information.

Agreement between the model-simulated and observed climatology is further evaluated in a Taylor Diagram (Figure 4a) [*Taylor*, 2001]. In this polar plot, the reference or observed data are plotted on the abscissa, and the model-simulated values lie in the first quadrant if the spatial correlation coefficient is positive. The radial dimension indicates the standardized deviation (ratio of standard deviation of simulated and observed fields, ratios >1 indicate that simulated values are more variable than observed) of the monthly climatology, and the angular dimension shows the pattern correlations. These statistics are computed using the 1971–2000 climatology. The similarity between model-simulated and observed precipitation is quantified in terms of their correlation and the amplitude of the variability. For example, models M7, M20, and M21 have a pattern correlation of about 0.85, but the standardized deviations are higher (>1.0) for M20 and M21 than M7. On the other hand, the pattern correlation of M23 is slightly higher than M9, but the standardized deviation is lower (<1) for M9. Many models show large variability even though the pattern correlations are high. The range of pattern correlations and standardized deviations of five models in the SME are 0.82–0.90 and 0.96–1.08, respectively. Also, the MME (n = 25), MMEhires average (MMEhires, n = 11), and MMElores average (MMElores, n = 14) outperform many of the individual models, but their performance measures are within the range of the SME models.

We compared modeled interannual variability using standardized anomalies for the period of 1950–2000 and coefficient of variation (Figure S8 in the supporting information) in CA and the subregions. The coefficient of variation of observed annual precipitation varies between 10 and 20% within CA; however, the modeled values (median across the 25 models) vary between 5 and 10%. Northern CA exhibits the largest variability, while the equatorial CA shows the smallest. All models capture this pattern qualitatively (see Figure S8 in the supporting information), although the variability is consistently low.

**AGU** **Journal of Geophysical Research: Atmospheres**  10.1002/2015JD023656

**Table 2.** Multimodel Mean Change in Precipitation (%) and Temperature (°C) for the Near Term (2021–2050) and the Long Term (2070–2099) in Relation to the Historical Period (1971–2000)[a]

| | Near Term (2021–2050) | | | | Long Term (2070–2099) | | | |
|---|---|---|---|---|---|---|---|---|
| | Precipitation (%) | | Temperature (°C) | | Precipitation (%) | | Temperature (°C) | |
| | RCP45 | RCP85 | RCP45 | RCP85 | RCP45 | RCP85 | RCP45 | RCP85 |
| Central Africa | 2.2 (3.7) | 2.7 (3.8) | 1.4 | 1.6 | 3.7 (5.6) | 7.0 (9.2) | 2.3 (2.4) | 4.1 (4.3) |
| Northern | 4.5 (6.9) | 5.5 (6.4) | 1.4 | 1.6 (1.7) | 6.6 (7.8) | 14.0 (15.2) | 2.3 (2.5) | 4.1 (4.4) |
| equatorial west | 1.7 (3.8) | 1.8 (4.3) | 1.3 | 1.5 | 2.6 (5.5) | 4.6 (8.4) | 2.1 (2.3) | 3.9 (4.1) |
| Equatorial east | 2.4 (0.6) | 2.6 (0.5) | 1.3 (1.4) | 1.2 (1.6) | 5.7 (5.1) | 10.1 (7.4) | 2.2 (2.3) | 3.9 (4.1) |
| Southwestern | 0.6 (2.7) | 0.8 (2.6) | 1.4 | 1.7 (1.6) | 1.4 (4.5) | 0.2 (3.8) | 2.3 (2.4) | 4.3 |
| Southeastern | −1.4 (−1.0) | −1.0 (0) | 1.5 (1.4) | 1.7 | −0.7 (2.2) | −1.5 (3.2) | 2.4 (2.5) | 4.4 (4.3) |

[a]Projected changes by the selected model ensemble (M6, M7, M18, M23, and M24) are given in parentheses. Only one value is given when they are the same for both.

The bias correction reduced the RMSD of annual precipitation to $30 \pm 4$ mm. The spatial-temporal patterns and magnitudes of the bias-corrected precipitation fields agree with the historical observed climatology (Figures 5a–5d and Figure S9 in the supporting information).

**3.1.2. Future Projections**

The change in precipitation in CA relative to the historical (reference) period (1971–2000) varies between −6% and 12% for the near term (2021–2050) and −9% and 27% for the long term (2070–2099) between the 25 models and the two RCPs. In the near term, the MME mean projects a 2.4% and 2.8% increase in precipitation relative to the reference period for RCP45 and RCP85, respectively. The projected changes for the long term are 4% and 7% for RCP45 and RCP85, respectively (Table 2). Overall, the MME average projections show larger precipitation increases in the northern latitudes (~6% in the near term for both RCPs and 8% and 15% in the long term for RCP45 and RCP85, respectively), moderate increases in the equatorial and southwestern regions and a decline in the southeast in the near and long terms for both RCPs (Table 2 and Figures 6a–6d). For a given model, the spatial patterns of precipitation in the projection periods are remarkably similar to the historical period, despite the differences in the emission scenarios (figure not shown). The spatial patterns of projected change between the RCPs are also similar within individual models. However, the magnitude of change and its spatial variation within CA are not uniform across the models (Figures 6e–6h). Large variations in the projected changes are noticeable in northern and eastern CA, particularly under the long-term projections, compared to moderate variations elsewhere. Seventeen models project an increase in precipitation in the northern and equatorial CA, whereas 14 models project an increase in southern CA (Figures 6i–6l). Projected changes are greater than twice that of the MME in at least eight models in the near-term and five models in the long term for both RCPs. The near-term projections do not vary much between the RCPs, but they diverge in the long-term (Figure S10 in the supporting information).

To further confirm that the projected changes are significantly different from the historical period, we compare the 30 year annual precipitation in the reference, near-, and long-term periods for the two RCPs by means of a $t$ test. We test the null hypothesis that the means in the respective periods are from the same population; rejection of this implies that they are not (at 5% level). The results reveal that not all the models' projections are significant, and substantial spatial variability exists between models and RCPs. The changes that are significant in the projection periods are highlighted with hatching in Figures 6a–6d. We only show (crosshatch) regions where at least half the models project a significant change. The changes, irrespective of the RCPs, are not significant in the near term in most parts of CA, except in the northern latitudes. In the long term, more models agree on the sign and magnitude of change under RCP85 than RCP45. Model consensus on the decrease in precipitation in southeastern CA is also uncertain in the near-term and long-term RCP45 scenario. The SME ensemble shows moderate reductions in the near term but an increase in the long term (see Table 2).

The projected changes in the spatiotemporal pattern of the CA rain belt show an intense rainy season from August to December (Figure 7). The wet season rainfall intensity is more pronounced in the long-term RCP85 compared to the RCP45. The location of maximum precipitation along the latitudes does not vary at monthly scale.

[Figure]

**Figure 6.** Multimodel mean change in precipitation in the Congo Region in relation to the reference period of 1971–2000. (a–d) Change in precipitation as percentage, (e–h) standard deviation of precipitation change across the models, and (i–l) number of models out 25 that projects an increase in precipitation as fraction. Cross hatching in Figures 6a–6d indicate regions where annual values in the projection and reference periods are significantly different ($p < 0.05$ in $t$ test) in more than 50% of the models.

[Figure]

**Figure 7.** Zonally averaged deviation of monthly precipitation (mm/month) from the reference (1971–2000) climatology along the latitudes, averaged over the study area (10.5°E to 35.5°E), (a–c) near term, 2021–2050, RCP45, (d–f) near-term, 2021–2050, RCP85, (g–i) long term, 2070–2099, RCP45, and (j–l) long term, 2070–99, RCP85.

**3.2. Temperature**

**3.2.1. Historical Patterns**

The temperature trends for CA vary from −0.03 to +0.19°C/decade during 1950–2000 across the 25 models. The observed and all-model average trends for the same period are +0.14 and +0.10°C/decade, respectively. Only one model (M8) shows a negative trend. The model-predicted spatial variation of average annual temperature for the historical period (1971–2000) shows good agreement with the observations (Figures 2f–2i and Figure S11 in the supporting information). The pattern correlation between the annual observed and simulated temperatures varies between 0.7 and 0.9 for different models. Pattern correlations are lowest in January, November and December for all the models, whereas the highest values are for April, May, June, and July (Figure S12 in the supporting information). The median SS for annual and seasonal simulations for the historical period (1971–2000) varies between 0.3 and 0.7, except for DJF for which majority of the models show limited skill. Nevertheless, the models simulate temperature better than precipitation as shown in the Taylor diagram (Figure 4b). The overall bias among the models is $2 \pm 0.8$°C (one standard deviation). Six models (M11–M16, variants from one modeling group), have an overall warm bias, whereas, the models M9, M19, and M25 have an overall cold bias.

While changes in maximum and minimum temperature strongly affect the changes in mean temperature, the difference between them provides added knowledge about regional climate change. The observed DTR (maximum-minimum temperature) values are high (~14°C) in the northern and southern latitudes compared to the equatorial region (~11°C, Figure 2o). Comparison of model-simulated annual DTR shows that the spatial structure is satisfactorily simulated (pattern correlations 0.74–0.89, Figures 2k–2n), although many models do not capture the observed variability (Figure 4c). The annual average DTR has an overall bias of $3.3 \pm 0.8$°C. Among the individual models, M7 has the lowest bias (1.8°C).

The bias-corrected temperature (Figures 5e–5h and Figures S9c and S9d in the supporting information) and DTR fields are in good agreement with observations. The overall annual temperature bias is reduced to 0.15°C.

[Figure]

**Figure 8.** Multimodel mean change in temperature and diurnal temperature range over central Africa in relation to the reference period of 1971–2000. (a–d) Temperature change (°C), (e–h) standard deviations of temperature change across models (*n* = 25), (i–l) average change in diurnal temperature range (°C), and (m–p) standard deviation of the change in diurnal temperature range across models for the near term (2021–2050), long term (2070–2099), and RCP45 and RCP85.

[Figure]

**Figure 9.** Skill scores for the (a) annual and (b) December-January-February, (c) March-April-May, (d) June-July-August, and (e) September-October-November seasons for precipitation (blue) and temperature (red). Numbers of GCMs used in the MME averages are shown in the *x* axis.

**3.2.2. Future Projections**

The temperature anomalies for the two RCPs follow similar warming trends until the 2030s, but diverge toward the end of the century. The rate of temperature change is greater for RCP85 than RCP45 after 2040. The uncertainties in the projections across the models are larger and more visible in RCP85 than RCP45. Toward the end of the 21st century, RCP85 shows continuous warming as a result of increasing GHG emissions, whereas RCP45 shows signs of stabilization (see Figure S13 in the supporting information).

The average, near-term temperature is projected to increase by 1.4°C and 1.6°C for RCP45 and RCP85, respectively (Table 2). Over the long term, the temperature projections diverge considerably due to differences in emission pathways between the two RCPs. The average, long-term (2070–2099) temperature increase is 2.3°C and 4.1°C for RCP45 and RCP85, respectively (Table 2).

The spatial distribution in the projected temperature change and the intermodel variability are similar for both RCPs in the near term (Figures 8a, 8b, 8e, and 8f). In the long term, RCP85 projections exhibit larger spatial and intermodel variability in average temperature than RCP45 projections (Figures 8c, 8d, 8g, and 8h). Near the end of the 21st century, the subequatorial region undergoes greater warming than the equatorial region for both emission scenarios, with warming in parts of southern CA exceeding 5°C for RCP85. The DTR values display a decreasing pattern for most of the CA, except the southern CA for both the RCPs (Figures 8i–8l). These spatial patterns remain the same for the near and long terms. However, the magnitude of change and the intermodel variability is large for the long-term RCP85 scenario (Figures 8m–8p). The long-term DTR projections under RCP85 are highly variable among models.

**3.3. Model Performance and Projection Uncertainties**

We examined the performance of models in simulating historical climate using skill scores. The highest skill scores for an individual model are 0.7 (M17 and M18) and 0.80 (M8, M21, and M24) for precipitation and temperature, respectively, but M8 shows limited skill for precipitation. Several combinations of models yield higher skill scores than those of the "best" individual models. For example, the average of 13 models (6 MMEhires and 7 MMElores) with pattern correlations greater than 0.6 and the standardized deviations between 0.9 and 1.1 (subjective measures of acceptable model-data agreement) yields a skill score of 0.76 for precipitation and 0.85 for temperature in the historical period. All models in the SME have an SS greater than 0.5.

We also examined the performance of multimodel ensemble averages (MME) in simulating the historical climate by computing the skill scores of different combination of models, starting from the individual models, average of two models, the average of three models, and so on (limited to 500 combinations). The skill scores are estimated for annual and seasonal precipitation and temperature fields (Figure 9). Temperature skill scores are better than precipitation skill scores for all model combinations (Figure 9a). The upper whiskers in Figure 9 indicate the best skill scores obtained for different model combinations. For precipitation the best skill score value peaks at five or six model averages and slightly reduces as more models are added that simulate precipitation poorly (Figure 9a, whiskers of blue curve). The skill scores approach an asymptote after approximately five models have been averaged, although the uncertainty of selecting the five models that give the best skill score remain large, except for the JJA season. The uncertainty is reduced by at least 50% after five more models have been added; it is further reduced by 50% with the addition of another ten models.

**4. Discussion**

We have examined historical simulations and future projections of two climate fields that exert considerable control on the functioning of terrestrial ecosystems and water resources. Overall, the GCMs simulate the spatial patterns and seasonal shifts of historical observations of temperature better than precipitation. The spatial variability and magnitude of the observed annual and seasonal precipitation vary considerably in model simulations. All but seven models produce too much precipitation in CA, particularly in the tropical Congo River Basin. The strong seasonality, northward and southward movement of the rain belt, its asymmetric nature, and the bimodal annual cycle near the equator are simulated well by some models but poorly in many (e.g., M2, M11, and M25). The MMEhires models have better skill in simulating the seasonality in precipitation than the MMElores models. The historical simulation of precipitation does not vary much between ensembles of the same models, which suggest that differences in model performance is primarily due to the differences in model physics rather than natural variability.

[Figure]

**Figure 10.** Correlation between seasonal precipitation and SSTs near West African coast (0°S–15°S, 0°E–10°E) at the beginning of the season. Calculations are based on 1950–2000 seasonal anomalies based on the reference period of 1971–2000. The grey open circles show individual model correlations, whereas colored circles show specific models mentioned in the text. The dark black squares show the correlation between observed precipitation and SSTs for the same period. Correlations for the main rainy season in the four regions, identified in parenthesis (see Figure 1) are reported.

We postulate that differences in model skill in simulating precipitation are partly dependent on how they replicate the teleconnections with SST departures [*Balas et al.*, 2007; *Hastenrath*, 1984; *Hirst and Hastenrath*, 1983; *Suzuki*, 2011]. Other mechanisms are also involved (e.g., global tropical circulations, Indian Ocean SSTs, inland moisture transport, and topographically induced controls on MCS), but the Atlantic teleconnection may be one of the dominant (see *Dezfuli and Nicholson* [2013], *Hirst and Hastenrath* [1983], *Jackson et al.* [2009], *Nicholson and Grist* [2003], and *Mitchell and Wallace* [1992]). Moreover, CA also lacks observed gage data [*New et al.*, 2000]; therefore, comparing outputs such as wind patterns and SST departures in the tropical Atlantic could provide more insight about climate model performance. We investigated the correlation between SST departures in the eastern Atlantic (0°S–15°S and 0°E–10°E) and precipitation during the rainy seasons in NC, EQW, and SW. The results show that only a few models agree with the observations (Figure 10). Many models show strong positive correlations with SST in the SON season in EQW, but not the observed relationship in MAM—the secondary rainy season in EQW. We have highlighted a few models that are able to capture the relationship reasonably; these models also exhibit better agreement with historical precipitation and temperature climatology (high values of SS, spatial pattern correlations, and the asymmetric structure of precipitation seasonality). Precipitation and SST linkage are further influenced by remote forcings, for example, by seasonal wind stress relaxation over the Equatorial Atlantic [*Hirst and Hastenrath*, 1983; *Todd and Washington*, 2004], which can partly explain the differences in the two rainy season peaks.

GCMs evaluated here and several of their predecessors exhibit a pattern of excessive precipitation off the equator but not sufficient on the equator, which is often referred to as the double ITCZ problem [*Bellucci et al.*, 2010; *Lin*, 2007]. Recent studies report that many CMIP5 models (e.g., M5, M7, M20, and M25) exhibit the double ITCZ structure [*Dufresne et al.*, 2013; *Gent et al.*, 2011; *Li and Xie*, 2014]. The presence of a strong double ITCZ in M15, M19, and M25 or the absence of it in M8, M18, and M21, as reported by *Li and Xie* [2014], does not reveal any differences in skill in simulating the historical annual precipitation. However, the latter models reproduce the monthly space time structure better (Figure S7 in the supporting information). Model parameterization schemes, their performance and potential inadequacies, and further research needs in tropical regions are further discussed in *Cook and Vizy* [2006], *Dai* [2006], *Monerie et al.* [2012], *Pearson et al.* [2014], *Phillips and Gleckler* [2006], and *Washington et al.* [2013]. *Gent et al.* [2011] and *Meehl et al.* [2013] provide evidence of improved representation of model processes (e.g., convection, plant functional type dependency on soil moisture, and evapotranspiration) and subsequent improvements in precipitation and temperature simulations in models M5 and its successor M6; however, the double ITCZ problem remains in M5. The southern Atlantic moisture flow, which is modulated by seasonal SSTs and trade winds, is an important source of atmospheric moisture and, subsequently precipitation in CA [*Gimeno et al.*, 2012; *Hastenrath*, 1991; *Kent et al.*, 2015]; however, many models exhibit too strong or weak coupling with Atlantic SSTs. The overestimation of ocean evaporation in M25, as reported in *Bentsen et al.* [2013], can be the cause of significant overestimation of precipitation in the CA. Evolution of Atlantic SSTs, which is an important driver of CA's precipitation, in several CMIP5 models is presented in *Richter et al.* [2014]. They illustrate that some model simulations (e.g., M18 and M24) have substantially improved compared to their earlier versions and propose that modeling groups should intensify their efforts to

[Figure]

**Figure 11.** Comparison of projected changes in (a–d) precipitation (*Pr*) and (e–h) temperature (*T*) with skill scores for 5, 10, 15, and 20 MME averages. Each MME combination consists of 500 samples. The red asterisks show the median of skill score on *x* axis and projection values on *y* axis; 50% of (*n* = 250) the samples lie within the inner contour line. These plots visualize location, spread, correlation, skewness, and tails of skill scores versus projections.

improve basic model performance. Most models, according to Richter et al., overestimate the Atlantic cold tongue SSTs, which have implications on the latitudinal position of the ITCZ and seasonality and magnitude of continental precipitation.

The MCSs also play significant role in spatiotemporal variability of precipitation in CA. The importance of topographic (e.g., Rift Valley and Cameroon highlands) and environmental (e.g., relative humidity and convective available potential energy, remote forcings such as SSTs and wind stress) controls on the growth and propagation of MCSs, and the lack of their representation and subsequent poor simulation of precipitation in GCMs are evaluated in recent studies (e.g., see *Farnsworth et al.* [2011], *Jackson et al.* [2009], *Janiga and Thorncroft* [2014], and [*Richter et al.*, 2014]). Therefore, we emphasize, for instance, that causal mechanisms related to the northward and southward phases of precipitation (*Jackson et al.* [2009] and *Hirst and Hastenrath* [1983], see Figure 3) and their representation in models require separate consideration.

Although, the ensemble average of MMEhires models reproduces observed precipitation better than the average of MMElores models, we do not find a consistent pattern of better performance among the MMEhires models. For example, M7 and M18 simulate precipitation better than M5 and M21, although all four simulate temperature and DTR about the same. The observed spatial patterns in precipitation such as the east-west distribution near the equator (see Figure 2 and Figure S1 in the supporting information) are

not captured by many models, with the eastern heavy rain zone (~20°E–25°E) appears to be drifted farther eastward than observed. This pattern is more apparent in the MMElores models. The SME compares well with historical observations than the all average.

The Taylor diagram shows that the pattern correlations of precipitation vary between 0.65 and 0.95. For comparison, the values reported for global correlations of precipitation in two previous CMIP studies (CMIP2 and CMIP3) are ~0.4–0.7 and ~0.5–0.8, respectively [*Phillips and Gleckler*, 2006]. Note that the present generation of GCMs includes time-varying GHG concentrations and aerosols and dynamic land use change, which many previous generations of GCMs did not include. The Taylor diagram also reveals that the majority of the models show large intraannual variability and greater model to observation mismatch for precipitation than temperature. However, the MME or the subsets of models (SME, MMEhires, and MMElores) consistently outperform individual models for both precipitation and temperature, which suggests that a smaller group of models can be used in regional climate change assessment. The SME is a subset of MMEhires, which suggests that a combination of skillful process representation and horizontal resolution improves model performance. However, precipitation projections of the SME (and most of the MMEhires) vary under both emission scenarios, indicating that model skill does not necessarily reduce projection uncertainty between models.

The performance of the multimodel ensemble means in the historical period improves as more models are added (Figure 9), irrespective of the performance of individual models. This is attributable to the contribution of different skill levels by individual models, which negates individual model errors, in agreement with conclusions drawn for other regions (see *Knutti et al.* [2010], *Pierce et al.* [2009], and *Phillips and Gleckler* [2006]). However, as more models are added to the MME, the historical performance becomes less important for projecting future changes. For example, with a five-member MME, the median value of projected change in precipitation (temperature) in the near- and long-term RCP85 are $2.9 \pm 1.9\%$ ($1.6 \pm 0.1°C$) and $7.3 \pm 4.0\%$ ($4.1 \pm 0.3°C$), respectively, whereas with 15-member MME, the projected changes are $2.8 \pm 0.8\%$ ($1.6 \pm 0.05°C$) and $7.1 \pm 1.6\%$ ($4.1 \pm 0.1°C$), respectively (the ranges are one standard deviation from the median); which indicates that projection uncertainty reduces with more models in the MME, particularly true for precipitation projections. To illustrate this further, we plot the historical skill scores against projected changes in precipitation (Figures 11a–11d) and temperature (Figures 11e–11h) for the near and long terms and for the two RCPs. The figure uses 5, 10, 15, and 20 MME averages of several model combinations. For a skill score range between 0.50 and 0.75, the near-term projections vary between 2% and 5% for the two RCPs when 15 or more models are used. The long-term RCPs still overlap with the near-term projections in many of the MME combinations. However, in the long-term scenarios, models with higher skill scores (e.g., 0.70–0.75 range) uniformly predict a smaller increase in precipitation than models with lower skill scores (e.g., 0.50–0.55, Figures 11a–11d). In contrast, as the number of models increases in the MMEs, the skill scores of historical temperature simulations improve and a well-defined warming signal emerges in the near and long terms for both RCPs. It is also clear that the choice of GCMs and emission scenarios will dominate the climate change projection toward the end of the century. Interestingly, the SME projections lie within the ranges reported above (see Table 2); however, the spatial patterns are slightly different in the precipitation projections (Figure S14 in the supporting information). Both ensembles project a reduction in precipitation in southern CA, but the increase in the north in the near term is slightly less. The SME also shows a region of reduced precipitation in the EQE along the Rift Valley (0°N–5°N and 30°E–35°E) in the near term.

The rate of change in precipitation during the northward and southward phases of the ITCZ is significantly different (Figure 7); the contrast appears prominently in the long-term RCP85. In the equatorial region, projected changes are higher in the second semester (SON) compared to the first semester (MAM); we expect that these disparities are modulated by different circulation patterns (e.g., tropical jets), SST anomalies in the Atlantic and Indian Oceans, and the contribution of MCS [*Dezfuli and Nicholson*, 2013; *Jackson et al.*, 2009; *Pearson et al.*, 2014] and therefore warrant separate investigation. Recent efforts exploring spatial resolution and representation of convection processes reveal that both improve model simulations in the tropical regions; however, processes representation is the main source of biases [*Birch et al.*, 2014; *Pearson et al.*, 2014].

Overall, precipitation increases during the wet and dry seasons in the northern and equatorial regions but decreases in the south. The increase (or decrease in the south) is more prominent under RCP85 than RCP45, indicating a stronger response of the climate system to anthropogenic warming. In addition to the drying, both maximum and minimum temperatures rise simultaneously in the south (DTR increases, see Figure 8), which will have implications on hydrology and ecosystem functioning in the region. Projected

changes in the near and long terms are consistent with projections reported in *Meehl et al.* [2012] for M5. We also find that overall projections lie within the range reported in the IPCC Fourth Assessment Report; notably, the long-term projections for RCP45 and RCP85 are comparable with that of the B1 and A1F1 scenarios in the fourth assessment [*Rogelj et al.*, 2012; *Solomon et al.*, 2007].

The comparisons above present an informed model selection process for regional climate change assessment. Our analyses show that the models in the SME have high SS, simulate the historical climatology well, and exhibit good correlations with likely mechanisms that dominate the regional climate. Recent work on mechanisms controlling precipitation variability in CA and model performance further support this [*Dezfuli and Nicholson*, 2013; *Giannini et al.*, 2008; *Richter et al.*, 2014; *Sandjon et al.*, 2012; *Washington et al.*, 2013]. We anticipate that models that simulate precipitation and temperature well exhibit better performance with other variables and teleconnections such as the ENSO and the Atlantic Multidecadal Oscillation.

The choice of emission scenarios will dominate the projection toward the end of the century as seen in Figures 6 and 8. Consensus among GCMs on the temperature and DTR projections are high and distinctly different between the two RCPs, but it is rather weak in the precipitation projections. The limited model skill for precipitation in the tropics is attributable to several factors including, though not limited to (a) inadequate process understanding, (b) physical processes representation and parameterization, (c) rather coarse horizontal resolution, and (d) insufficient observations in the region to better constrain models. Further, terrestrial vegetation response and the climate system are closely coupled; how tropical vegetation will respond to changes in climate still remains a large source of uncertainty [see *Huntingford et al.*, 2013]. Mechanisms through which vegetation influences climate include albedo, canopy conductance, photosynthesis, and water and energy fluxes [*Richardson et al.*, 2013]. Feedback between vegetation and climate are represented by plant functional types; however, representation of vegetation (particularly tropical vegetation) in GCMs that couples land surface to the atmosphere requires considerable improvements [*Bonan*, 2008; *Friend et al.*, 2014; *Randall et al.*, 2007]. Diagnostic analyses of these issues are beyond the scope of this study, but several of these are discussed in *Knutti and Sedlacek* [2013], *Giannini et al.* [2008], [*Pearson et al.* [2014], and [*Richardson et al.* [2013]. Based on a global analysis using several CMIP5 and CMIP3 models, *Knutti and Sedlacek* [2013] reports that projected changes in temperature and precipitation are remarkably similar in CMIP5 and CMIP3. Given that, by design, the CMIP5 models use the same GHG, land use and other input forcings, the main reason for divergent projections ought to be the representation of different physical processes. Attributes that modulate the space-time variability of precipitation, including the SST anomalies in the tropical oceans, movement of the ITCZ, zonal and meridional wind patterns, moisture flux, and cloudiness are presented in *Nicholson and Dezfuli* [2013], *Nicholson and Grist* [2003], *Sandjon et al.* [2012], and *Suzuki* [2011]. Detailed investigations of how algorithms represent these attributes will illuminate the limitations climate models have in simulating precipitation fields in tropical regions like CA.

Our analyses highlight that both precipitation and temperature simulated by all models exhibit biases that should prevent their direct use in climate change impact and adaptation studies. Even the individual models and the MME that yield better performance in the historical period contain systematic biases, which are attributable to the limited skills in simulating the spatial patterns and variability. Hence, the statistical bias correction methods, such as EDCDF, are employed to correct biases in the monthly climate fields. *Ehret et al.* [2012] argue that bias correction methods, similar to what is employed in this study, are not valid procedures that can be used to correct GCM simulation mismatch, as they are statistical methods that neither take into account the model's physical processes nor consider any feedback mechanisms associated with other variables. However, we emphasize the inherent limitations (discussed elsewhere in this paper) of climate models in simulating various climate fields, particularly in the tropical regions like the CA, where observations are severely lacking. The uncertainty associated with the choice of bias correction method is marginal compared to climate model and emission scenario selection [*Chen et al.*, 2011; *Li et al.*, 2010]. Any climate change assessment where the GCM outputs are used as input will inherit the biases in the subsequent outputs. Studies on climate change impacts on land surface hydrology and agriculture report that models forced with outputs directly from the GCMs or even regional climate models nested within GCMs result in unacceptably biased simulations, and therefore, the GCM (or regional model) outputs ought to be bias corrected [*Glotter et al.*, 2014; *Salathé et al.*, 2007; *Sulis et al.*, 2012; *Wood et al.*, 2004]. Since the model outputs cannot be used directly, we apply bias correction that corrects all moments of the climate fields' statistical distribution at monthly scale. We choose monthly outputs over daily because the region lacks good coverage of observed daily climate fields. We propose methods where historical model performance and future changes can be quantitatively assessed and develop bias-corrected

[Figure]
 **Journal of Geophysical Research: Atmospheres**  10.1002/2015JD023656

climate fields that can be used to analyze climate change response of ecosystem services in the future. Similar bias correction methods have been successfully implemented in several studies [*Maurer and Hidalgo*, 2008; *Piani et al.*, 2010; *Teutschbein and Seibert*, 2012; *Thrasher et al.*, 2012; *Watanabe et al.*, 2012]. Our analyses also demonstrate that the choice of climate models and emission scenario will dominate the variability in climate change projections. The impacts of the projected changes in precipitation and temperature on the hydrological cycle in the Congo River Basin will be explored in a companion study.

**5. Conclusions**

We present an evaluation of the performance of several CMIP5 models in CA and assess climate projections for two RCPs. The multimodel evaluation highlights the spatiotemporal variability of precipitation and temperature in CA and illustrates the uncertainties in 21st century projections. Although several GCMs simulate important features (in precipitation and temperature) and the climatology in the CA reasonably well, model outputs ought to be corrected before they can be used as practical planning tools. We identified a subset of models based on historical performance and causal mechanisms that can provide plausible future projections and highlighted uncertainties associated with model selection. The advances in climate model development in CMIP5 provide a better perspective on how global changes and circulations affect regional climate in central Africa (e.g., SST changes in tropical Atlantic). Future efforts on climate change mitigation and adaptation in CA should consider these global implications.

The climate fields developed in this study can be used in hydrologic and ecosystem modeling at regional and local scales. They can also be used to develop diagnostic tools to analyze threats to water and food security, as well as ecosystem sustainability. Even though the focus of our analysis is the CA, the data generated are available for the global land areas and provide spatially and temporally consistent precipitation and temperature fields for specific regions that end users seek. The precipitation, temperature, and DTR data for the period of 1950–2099 are available at 1° spatial resolution over global land areas in netCDF format from http://hydrology.princeton.edu/data.php. The monthly data set includes both before and after bias-corrected climate fields.

**Acknowledgments**
We thank the two anonymous reviewers for their valuable comments. We acknowledge the World Climate Research Program's Working Group on Coupled Modeling, which is responsible for CMIP, and we thank the climate modeling groups (listed in Table 1) for producing and making available their model output. We would like to thank Nadine Laporte at the Woods Hole Research Center, Falmouth, MA, and Ronald B. Smith at the Department of Geology and Geophysics at Yale University, New Haven, CT, for their valuable comments during the development of this manuscript. For CMIP, the U.S. Department of Energy's Program for Climate Model Diagnosis and Intercomparison provides coordinating support and led development of software infrastructure in partnership with the Global Organization for Earth System Science Portals. Noel Aloysius acknowledges the support provided by the School of Forestry and Environmental Studies, the Graduate School of Arts and Sciences at Yale University, and the Department of Civil and Environmental Engineering at Princeton University. This work was supported in part by the facilities and staff of the Yale University Faculty of Arts and Sciences High Performance Computing Center, the National Science Foundation under grant CNS 08–21132 that partially funded acquisition of the facilities, and NOAA grants NA10OAR4310130 and NA11OAR4310097.

**References**

Arora, V. K., J. F. Scinocca, G. J. Boer, J. R. Christian, K. L. Denman, G. M. Flato, V. V. Kharin, W. G. Lee, and W. J. Merryfield (2011), Carbon emission limits required to satisfy future representative concentration pathways of greenhouse gases, *Geophys. Res. Lett.*, 38(5), L05805.

Baccini, A., N. Laporte, S. J. Goetz, M. Sun, and H. Dong (2008), A first map of tropical Africa's above-ground biomass derived from satellite imagery, *Environ. Res. Lett.*, 3(4).

Balas, N., S. E. Nicholson, and D. Klotter (2007), The relationship of rainfall variability in West central Africa to sea surface temperature fluctuations, *Int. J. Climatol.*, 27(10), 1335–1349.

Bao, Y., F. L. Qiao, and Z. Y. Song (2012), The historical global carbon cycle simulation of FIO-ESM, paper presented at EGU General Assembly 2012, EGU, Vienna, Austria.

Bellucci, A., S. Gualdi, and A. Navarra (2010), The double-ITCZ syndrome in coupled general circulation models: The role of large-scale vertical circulation regimes, *J. Clim.*, 23(5), 1127–1145.

Bentsen, M., et al. (2013), The Norwegian Earth System Model, NorESM1-M – Part 1: Description and basic evaluation of the physical climate, *Geosci. Model Dev.*, 6(3), 687–720.

Bi, D. (2012), The ACCESS Coupled Model: Description, Control Climate and Evaluation.

Birch, C. E., J. H. Marsham, D. J. Parker, and C. M. Taylor (2014), The scale dependence and structure of convergence fields preceding the initiation of deep convection, *Geophys. Res. Lett.*, 41, 4769–4776, doi:10.1002/2014GL060493.

Bohn, T. J., D. P. Lettenmaier, K. Sathulur, L. C. Bowling, E. Podest, K. C. McDonald, and T. Friborg (2007), Methane emissions from western Siberian wetlands: Heterogeneity and sensitivity to climate change, *Environ. Res. Lett.*, 2(4).

Boko, M., I. Niang, A. Nyong, C. Vogel, A. Githeko, M. Medany, B. Osman-Elasha, R. Tabo, and P. Yanda (2007), Africa climate change 2007: Impacts, adaptation and vulnerability, in *Contribution of Working Group II to the Fourth Assessment Report of the Intergovernmental Panel on Climate Change*, edited by M. L. Parry et al., Cambridge Univ. Press, Cambridge, U. K.

Bonan, G. B. (2008), Forests and climate change: Forcings, feedback, and the climate benefits of forests, *Science*, 320(5882), 1444–1449.

Bruinsma, J. (2003), *World Agriculture: Towards 2015/2030: An FAO Perspective*, 520 pp., Earthscan/James & James, London, U. K.

Chen, J., O. Haerter, S. Hagemann, and C. Piani (2011), On the contribution of statistical bias correction to the uncertainty in the projected hydrological cycle, *Geophys. Res. Lett.*, 38, L20403, doi:10.1029/2011GL049318.

Collier, P., G. Conway, and T. Venables (2008), Climate change and Africa, *Oxford Rev. Econ. Policy*, 24(2), 337–353.

Collins, W., N. Bellouin, M. Doutriaux-Boucher, N. Gedney, P. Halloran, T. Hinton, J. Hughes, C. Jones, M. Joshi, and S. Liddicoat (2011), Development and evaluation of an Earth-system model-HadGEM2, *Geosci. Model Dev. Discuss.*, 4, 997–1062.

Cook, K. H., and E. K. Vizy (2006), Coupled model simulations of the West African monsoon system: Twentieth- and 21st century simulations, *J. Clim.*, 19(15), 3681–3703.

Dai, A. (2006), Precipitation characteristics in eighteen coupled climate models, *J. Clim.*, 19(18), 4605–4630.

DeFries, R. S., T. Rudel, M. Uriarte, and M. Hansen (2010), Deforestation driven by urban population growth and agricultural trade in the 21st century, *Nat. Geosci.*, 3(3), 178–181.

[Figure]

**Journal of Geophysical Research: Atmospheres**     10.1002/2015JD023656

Demaria, E. M. C., E. P. Maurer, J. Sheffield, E. Bustos, D. Poblete, S. Vicuña, and F. Meza (2012), Using a gridded global data set to characterize regional hydroclimate in central Chile, *J. Hydrometeorol.*, *14*, 251–265.

Dezfuli, A. K., and S. E. Nicholson (2013), The relationship of rainfall variability in western equatorial Africa to the tropical oceans and atmospheric circulation. Part II: The boreal autumn, *J. Clim.*, *26*(1), 66–84.

Dufresne, J. L., et al. (2013), Climate change projections using the IPSL-CM5 Earth System Model: From CMIP3 to CMIP5, *Clim. Dyn.*, *40*(9–10), 2123–2165.

Ehret, U., E. Zehe, V. Wulfmeyer, K. Warrach-Sagi, and J. Liebert (2012), HESS opinions "Should we apply bias correction to global and regional climate model data?", *Hydrol. Earth Syst. Sci. Discuss.*, *9*(4), 5355–5387.

Farnsworth, A., E. White, C. J. R. Williams, E. Black, and D. R. Kniveton (2011), Understanding the large scale driving mechanisms of rainfall variability over central Africa, in *African Climate and Climate Change: Physical, Social and Political Perspectives*, edited by C. J. R. Williams and D. R. Kniveton, pp. 101–122, Springer, New York.

Fisher, J. B., et al. (2013), African tropical rainforest net carbon dioxide fluxes in the twentieth century, *Phil. Trans. R. Soc. B*, *368*(1625).

Friend, A. D., et al. (2014), Carbon residence time dominates uncertainty in terrestrial vegetation responses to future climate and atmospheric $CO_2$, *Proc. Natl. Acad. Sci. U.S.A.*, *111*(9), 3280–3285.

Gent, P. R., G. Danabasoglu, L. J. Donner, M. M. Holland, E. C. Hunke, S. R. Jayne, D. M. Lawrence, R. B. Neale, P. J. Rasch, and M. Vertenstein (2011), The community climate system model version 4, *J. Clim.*, *24*(19), 4973–4991.

Giannini, A., M. Biasutti, I. Held, and A. Sobel (2008), A global perspective on African climate, *Clim. Change*, *90*(4), 359–383.

Gimeno, L., A. Stohl, R. M. Trigo, F. Dominguez, K. Yoshimura, L. Yu, A. Drumond, A. M. Durn-Quesada, and R. Nieto (2012), Oceanic and terrestrial sources of continental precipitation, *Rev. Geophys.*, *50*, RG4003, doi:10.1029/2012RG000389.

Giorgetta, M. A., et al. (2013), Climate and carbon cycle changes from 1850 to 2100 in MPI-ESM simulations for the coupled model intercomparison project phase 5, *J. Adv. Model. Earth Syst.*, *5*(3), 572–597.

Glotter, M., J. Elliott, D. McInerney, N. Best, I. Foster, and E. J. Moyer (2014), Evaluating the utility of dynamical downscaling in agricultural impacts projections, *Proc. Natl. Acad. Sci. U.S.A.*, *111*(24), 8776–8781.

Hastenrath, S. (1984), Interannual variability and annual cycle: mechanisms of circulation and climate in the tropical Atlantic sector, *Mon. Weather Rev.*, *112*(6), 1097–1107.

Hastenrath, S. (1991), Interannual variability of the atmosphere–ocean system, in *Climate Dynamics of the Tropics*, edited, pp. 322–329, Kluwer Acad., Dordrecht, Netherlands.

Hazeleger, W., et al. (2010), EC-Earth: A Seamless Earth-System Prediction Approach in Action, *Bull. Am. Meteorol. Soc.*, *91*(10), 1357.

Hazeleger, W., et al. (2012), EC-Earth V2.2: Description and validation of a new seamless earth system prediction model, *Clim. Dyn.*, *39*(11), 2611–2629.

Hirst, A. C., and S. Hastenrath (1983), Diagnostics of hydrometeorological anomalies in the Zaire (Congo) basin, *Q. J. R. Meteorol. Soc.*, *109*(462), 881–892.

Houze, R. A. (2004), Mesoscale convective systems, *Rev. Geophys.*, *42*, RG4003, doi:10.1029/2004RG000150.

Huntingford, C., P. Zelazowski, D. Galbraith, L. M. Mercado, S. Sitch, R. Fisher, and M. Lomas (2013), Simulated resilience of tropical rainforests to $CO_2$-induced climate change, *Nat. Geosci.*, *6*(4), 268–273.

Hurrell, J. W., J. J. Hack, D. Shea, J. M. Caron, and J. Rosinski (2008), A new sea surface temperature and sea ice boundary data set for the community atmosphere model, *J. Clim.*, *21*(19), 5145–5153.

International Institute for Applied Systems Analysis (2009), RCP Database (version 2.0). [Available at http://www.iiasa.ac.at/web-apps/tnt/RcpDb.]

Ines, A. V. M., and J. W. Hansen (2006), Bias correction of daily GCM rainfall for crop simulation studies, *Agric. Forest Meteorol.*, *138*(1–4), 44–53.

Intergovernmental Panel on Climate Change (IPCC) (2007), Climate change 2007: Impacts, adaptation and vulnerability: Contribution of working group II to the fourth assessment report of the Intergovernmental Panel on Climate Change, *Rep.*, 976 pp., Intergovernmental Panel on Climate Change, Cambridge, U. K.

Iversen, T., et al. (2012), The Norwegian Earth System Model, NorESM1-M - Part 2: Climate response and scenario projections, *Geosci. Model Dev. Discuss.*, *5*(3), 2933–2998.

Jackson, B., S. E. Nicholson, and D. Klotter (2009), Mesoscale convective systems over western equatorial Africa and their relationship to large-scale circulation, *Mon. Weather Rev.*, *137*(4), 1272–1294.

Janiga, M. A., and C. D. Thorncroft (2014), Convection over tropical Africa and the east Atlantic during the West African monsoon: Regional and diurnal variability, *J. Clim.*, *27*, 4159–4188.

Kent, C., R. Chadwick, and D. P. Rowell (2015), Understanding uncertainties in future projections of seasonal tropical precipitation, *J. Clim.*, *28*(11), 4390–4413.

Knutti, R., and J. Sedlacek (2013), Robustness and uncertainties in the new CMIP5 climate model projections, *Nat. Clim. Change*, *3*, 369–373.

Knutti, R., R. Furrer, C. Tebaldi, J. Cermak, and G. A. Meehl (2010), Challenges in combining projections from multiple climate models, *J. Clim.*, *23*(10), 2739–2758.

Laing, A. G., and J. M. Fritsch (1993), Mesoscale convective complexes in Africa, *Mon. Weather Rev.*, *121*(8), 2254–2263.

Laporte, N. T., J. A. Stabach, R. Grosch, T. S. Lin, and S. J. Goetz (2007), Expansion of industrial logging in central Africa, *Science*, *316*(5830), 1451.

Lawrence, P. J., et al. (2012), Simulating the biogeochemical and biogeophysical impacts of transient land cover change and wood harvest in the Community Climate System Model (CCSM4) from 1850 to 2100, *J. Clim.*, *25*(9), 3071–3095.

Li, G., and S. Xie (2014), Tropical biases in CMIP5 multimodel ensemble: The excessive equatorial Pacific cold tongue and double ITCZ problems, *J. Clim.*, *27*(4), 1765–1780.

Li, H., J. Sheffield, and E. F. Wood (2010), Bias correction of monthly precipitation and temperature fields from Intergovernmental Panel on Climate Change AR4 models using equidistant quantile matching, *J. Geophys. Res.*, *115*, D10101, doi:10.1029/2009JD012882.

Lin, J.-L. (2007), The double-ITCZ problem in IPCC AR4 coupled GCMs: Ocean–atmosphere feedback analysis, *J. Clim.*, *20*(18), 4497–4525.

Lobell, D. B., M. B. Burke, C. Tebaldi, M. D. Mastrandrea, W. P. Falcon, and R. L. Naylor (2008), Prioritizing climate change adaptation needs for food security in 2030, *Science*, *319*(5863), 607–610.

Maurer, E. P., and H. G. Hidalgo (2008), Utility of daily versus monthly large-scale climate data: An intercomparison of two statistical downscaling methods, *Hydrol. Earth Syst. Sci.*, *12*(2), 551–563.

Meehl, G. A., et al. (2012), Climate system response to external forcings and climate change projections in CCSM4, *J. Clim.*, *25*(11), 3661–3683.

Meehl, G. A., W. M. Washington, J. M. Arblaster, A. Hu, H. Teng, J. E. Kay, A. Gettelman, D. M. Lawrence, B. M. Sanderson, and W. G. Strand (2013), Climate change projections in CESM1(CAM5) compared to CCSM4, *J. Clim.*, *26*(17), 6287–6308.

Mitchell, T. D., and P. D. Jones (2005), An improved method of constructing a database of monthly climate observations and associated high-resolution grids, *Int. J. Climatol.*, *25*(6), 693–712.

Mitchell, T., and J. M. Wallace (1992), The annual cycle in equatorial convection and sea surface temperature, *J. Clim.*, *5*(10), 1140–1156.

Monerie, P. A., B. Fontaine, and P. Roucou (2012), Expected future changes in the African monsoon between 2030 and 2070 using some CMIP3 and CMIP5 models under a medium-low RCP scenario, *J. Geophys. Res.*, *117*, D16111, doi:10.1029/2012JD017510.

[Figure]
 **Journal of Geophysical Research: Atmospheres**  10.1002/2015JD023656

Moss, R. H., et al. (2010), The next generation of scenarios for climate change research and assessment, *Nature*, *463*(7282), 747–756.

Murphy, A. H. (1988), Skill scores based on the mean square error and their relationships to the correlation coefficient, *Mon. Weather Rev.*, *116*(12), 2417–2424.

New, M., M. Hulme, and P. Jones (2000), Representing twentieth-century space–time climate variability. Part II: Development of 1901–96 monthly grids of terrestrial surface climate, *J. Clim.*, *13*(13), 2217–2238.

Nicholson, S. E., and A. K. Dezfuli (2013), The relationship of rainfall variability in western equatorial Africa to the tropical oceans and atmospheric circulation. Part I: The boreal spring, *J. Clim.*, *26*(1), 45–65.

Nicholson, S. E., and J. P. Grist (2003), The seasonal evolution of the atmospheric circulation over West Africa and equatorial Africa, *J. Clim.*, *16*(7), 1013–1030.

Nkem, J., F. B. Kalame, M. Idinoba, O. A. Somorin, O. Ndoye, and A. Awono (2010), Shaping forest safety nets with markets: Adaptation to climate change under changing roles of tropical forests in Congo Basin, *Environ. Sci. Policy*, *13*(6), 498–508.

Notz, D., F. A. Haumann, H. Haak, J. H. Jungclaus, and J. Marotzke (2013), Arctic sea-ice evolution as modeled by Max Planck Institute for Meteorology's Earth system model, *J. Adv. Model. Earth Syst.*, *5*(2), 173–194.

Pearson, K. J., G. M. S. Lister, C. E. Birch, R. P. Allan, R. J. Hogan, and S. J. Woolnough (2014), Modelling the diurnal cycle of tropical convection across the 'grey zone', *Q. J. R. Meteorol. Soc.*, *140*(679), 491–499.

Phillips, T. J., and P. J. Gleckler (2006), Evaluation of continental precipitation in 20th century climate simulations: The utility of multimodel statistics, *Water Resour. Res.*, *42*, W03202, doi:10.1029/2005WR004313.

Piani, C., G. P. Weedon, M. Best, S. M. Gomes, P. Viterbo, S. Hagemann, and J. O. Haerter (2010), Statistical bias correction of global simulated daily precipitation and temperature for the application of hydrological models, *J. Hydrol.*, *395*(3–4), 199–215.

Pierce, D. W., T. P. Barnett, B. D. Santer, and P. J. Gleckler (2009), Selecting global climate models for regional climate change studies, *Proc. Natl. Acad. Sci. U.S.A.*, *106*(21), 8441–8446.

Randall, D. A., et al. (2007), Cilmate models and their evaluation, in *Climate Change 2007: The Physical Science Basis. Contribution of Working Group I to the Fourth Assessment Report of the Intergovernmental Panel on Climate Change*, edited by S. Solomon et al., Cambridge Univ. Press, Cambridge, U. K., and New York.

Revenga, C., S. Murray, J. Abramovitz, and A. Hammond (1998), *Watersheds of the World: Ecological Value and Vulnerability*, World Res. Inst., Washington, D. C.

Richardson, A. D., T. F. Keenan, M. Migliavacca, Y. Ryu, O. Sonnentag, and M. Toomey (2013), Climate change, phenology, and phenological control of vegetation feedbacks to the climate system, *Agric. Forest Meteorol.*, *169*, 156–173.

Richter, I., S.-P. Xie, S. Behera, T. Doi, and Y. Masumoto (2014), Equatorial Atlantic variability and its relation to mean state biases in CMIP5, *Clim. Dyn.*, *42*(1–2), 171–188.

Rogelj, J., M. Meinshausen, and R. Knutti (2012), Global warming under old and new scenarios using IPCC climate sensitivity range estimates, *Nat. Clim. Change*, *2*(4), 248–253.

Rotstayn, L. D., M. A. Collier, M. R. Dix, Y. Feng, H. B. Gordon, S. P. O'Farrell, I. N. Smith, and J. Syktus (2010), Improved simulation of Australian climate and ENSO-related rainfall variability in a global climate model with an interactive aerosol treatment, *Int. J. Climatol.*, *30*(7), 1067–1088.

Salathé, E. P., Jr., P. W. Mote, and M. W. Wiley (2007), Review of scenario selection and downscaling methods for the assessment of climate change impacts on hydrology in the United States pacific northwest, *Int. J. Climatol.*, *27*(12), 1611–1621.

Sandjon, A. T., A. Nzeukou, and C. Tchawoua (2012), Intraseasonal atmospheric variability and its interannual modulation in central Africa, *Meteorol. Atmos. Phys.*, *117*(3–4), 167–179.

Schmidt, G. A., R. Ruedy, J. E. Hansen, I. Aleinov, N. Bell, M. Bauer, S. Bauer, B. Cairns, V. Canuto, and Y. Cheng (2006), Present-day atmospheric simulations using GISS ModelE: Comparison to in situ, satellite, and reanalysis data, *J. Clim.*, *19*(2), 153–192.

Schneider, T., T. Bischoff, and G. H. Haug (2014), Migrations and dynamics of the Intertropical Convergence Zone, *Nature*, *513*(7516), 45–53.

Sheffield, J., G. Goteti, and E. F. Wood (2006), Development of a 50 year high-resolution global data set of meteorological forcings for land surface modeling, *J. Clim.*, *19*(13), 3088–3111.

Sheffield, J., E. F. Wood, and F. Munoz-Arriola (2010), Long-term regional estimates of evapotranspiration for Mexico based on downscaled ISCCP data, *J. Hydrometeorol.*, *11*(2), 253–275.

Solomon, S., D. Qin, M. Manning, Z. Chen, M. Marquis, K. B. Averyt, M. Tignor, and H. L. Miller (Eds.) (2007), *Climate change 2007: The Physical Science Basis: Contribution of Working Group I to the Fourth Assessment Report of the Intergovernmental Panel on Climate Change*, Cambridge Univ. Press, Cambridge, U. K.

Sulis, M., C. Paniconi, M. Marrocu, D. Huard, and D. Chaumont (2012), Hydrologic response to multimodel climate output using a physically based model of groundwater/surface water interactions, *Water Resour. Res.*, *48*, W12510, doi:10.1029/2012WR012304.

Suzuki, T. (2011), Seasonal variation of the ITCZ and its characteristics over central Africa, *Theor. Appl. Climatol.*, *103*(1), 39–60.

Taylor, K. E. (2001), Summarizing multiple aspects of model performance in a single diagram, *J. Geophys. Res.*, *106*(D7), 7183–7192, doi:10.1029/2000JD900719.

Taylor, K. E., V. Balaji, S. Hankin, M. Juckers, and B. Lawrence (2009a), *CMIP5 and AR5 Data Reference Syntax (DRS)*, 7 pp., World Clim. Res. Program.

Taylor, K. E., R. Stouffer, and G. Meehl (2009b), A summary of the CMIP5 experiment design, pp. 33. [Available at http://cmip-pcmdi.llnl.gov/cmip5/docs/Taylor_CMIP5_design.pdf, (last access: May 2012).]

Taylor, K. E., R. Stouffer, and G. Meehl (2012), An overview of CMIP5 and the experiment design, *Bull. Am. Meteorol. Soc.*, *93*(4), 485.

Teutschbein, C., and J. Seibert (2012), Bias correction of regional climate model simulations for hydrological climate-change impact studies: Review and evaluation of different methods, *J. Hydrol.*, *456–457*, 12–29.

Thrasher, B. L., E. P. Maurer, C. McKellar, and P. B. Duffy (2012), Technical note: Bias correcting climate model simulated daily temperature extremes with quantile mapping, *Hydrol. Earth Syst. Sci. Discuss.*, *9*(4), 5515–5529.

Todd, M. C., and R. Washington (2004), Climate variability in central equatorial Africa: Influence from the Atlantic sector, *Geophys. Res. Lett.*, *31*, L23202, doi:10.1029/2004GL020975.

United Nations Environment Programme (2006), Africa environment outlook 2:Our environment, our wealth *Rep.*, 576 pp., United Nations Environment Programme (UNEP), Nairobi, Kenya.

Voldoire, A., et al. (2012), The CNRM-CM5.1 global climate model: description and basic evaluation, *Clim. Dyn.*, 1–31.

Volodin, E. M., N. A. Dianskii, and A. V. Gusev (2010), Simulating present-day climate with the INMCM4.0 coupled model of the atmospheric and oceanic general circulations, *Izv. Atmos. Ocean. Phys.*, *46*(4), 414–431.

Wang, A. H., D. P. Lettenmaier, and J. Sheffield (2011), Soil moisture drought in China, 1950–2006, *J. Clim.*, *24*(13), 3257–3271.

Washington, R., R. James, H. Pearce, W. M. Pokam, and W. Moufouma-Okia (2013), Congo Basin rainfall climatology: Can we believe the climate models?, *Phil. Trans. R. Soc. B*, *368*(1625).

Watanabe, M., et al. (2010), Improved climate simulation by MIROC5: Mean States, variability, and climate sensitivity, *J. Clim.*, *23*(23), 6312–6335.

[Figure]
 **Journal of Geophysical Research: Atmospheres** 10.1002/2015JD023656

Watanabe, S., T. Hajima, K. Sudo, T. Nagashima, T. Takemura, H. Okajima, T. Nozawa, H. Kawase, M. Abe, and T. Yokohata, (2011), MIROC-ESM: Model description and basic results of CMIP5-20c3m experiments, *Geosci. Model Dev. Discuss.*, *4*, 1063–1128.

Watanabe, S., S. Kanae, S. Seto, P. J. F. Yeh, Y. Hirabayashi, and T. Oki (2012), Intercomparison of bias-correction methods for monthly temperature and precipitation simulated by multiple climate models, *J. Geophys. Res.*, *117*, D23114, doi:10.1029/2012JD018192.

Wilkie, D., G. Morelli, F. Rotberg, and E. Shaw (1999), Wetter is not better: Global warming and food security in the Congo Basin, *Global Environ. Change*, *9*(4), 323–328.

Wilks, D. S. (2006), *Statistical Methods in the Atmospheric Sciences*, 627 pp., Academic Press, Burlington, Mass.

Wood, A. W., L. R. Leung, V. Sridhar, and D. P. Lettenmaier (2004), Hydrologic implications of dynamical and statistical approaches to downscaling climate model outputs, *Clim. Change*, *62*(1), 189–216.

Wu, T., et al. (2013), Global carbon budgets simulated by the Beijing Climate Center Climate System Model for the last century, *J. Geophys. Res. Atmos.*, *118*(10), 4326–4327.

Xin, X.-G., T.-W. Wu, and J. Zhang (2013), Introduction of CMIP5 experiments Carried out with the Climate System Models of Beijing Climate Center, *Adv. Clim. Change Res.*, *4*(1), 41–49.

Yukimoto, S., A. Noda, A. Kitoh, M. Hosaka, H. Yoshimura, T. Uchiyama, K. Shibata, O. Arakawa, and S. Kusunoki (2006), Present-day climate and climate sensitivity in the Meteorological Research Institute coupled GCM version 2.3 (MRI-CGCM2.3), *J. Meteorol. Soc. Jpn.*, *84*(2), 333–363.

---

## Author Comment (AC2) · 16 Aug 2016

Thank you for pointing out the mistake. It has been corrected. Noel Aloysius

---

## Author Comment (AC3) · 17 Aug 2016

**General comments**

The authors address future change in water availability in the Congo Basin. This topic is welcomed given the relative lack of research for this important region. The authors have embarked on a thorough analysis using projections from 50 climate model experiments which are bias corrected and downscaled and run through a hydrologic model. As with any impacts study of this nature, there are a host of uncertainties and methodological choices which can influence the outcomes, and it is challenging to distill information about future impacts in this context. There are also different views amongst scientists as to the best way to approach these uncertainties. However, personally I feel that the balance of emphasis on uncertainties is not quite right in this study, and would like to see more discussion/emphasis on the climate model uncertainty (and less emphasis on the multi-model mean), as well as more analysis of observational uncertainty. Therefore I suggest major revisions. Please note that my background is in climate science so I will mainly comment on this component of the study, and do not have the relevant expertise to comment on the hydrological modelling.

1. Model uncertainty: Given the uncertainties associated with future climate, I think that the comments on the implications of the findings, particularly in the abstract and conclusions, are too strong. The "challenges" described for planners in the abstract occur only if the projections are valid (which we won't know for 50 years). The authors also make several comments about the importance of providing "details" for planners. I disagree. I think it is more important that planners are aware of uncertainties in future climate, and would benefit more from information about the range of future projections than the multi-model mean. (This is in line with a body of researchers and literature discussing such issues e.g. Weaver et al. 2011; Dessai et al. 2009; Knutti et al. 2008).

The authors have quite a large ensemble of projections from their modelling which could be made much more useful in this regard. I think it would be more useful if they commented on the size of the uncertainty and what this means for planning – are there any regions for which there is not a great deal of model uncertainty, where planners can prepare for wetter or drier conditions? Or is there also uncertainty in the direction of change which might mean that adaptive/robust planning strategies are more appropriate? How does the uncertainty from climate models compare to other uncertainties e.g. if a different hydrological model were used?

In general I think they should put more emphasis on understanding the range (e.g. Figure 7 which is useful) and less on the multi model mean (e.g. Figure 6, which should include a measure of uncertainty).

We have revised the text and abstract to highlight the projection ranges and the uncertainties planners will encounter. Figure 6 has been modified to show model projection agreement in runoff change. This figure also highlights the spatial variability in the direction of change.

The historical simulation period is from 1950-2008. The model was calibrated during the early part of the simulation period in order to take advantage of available observed river flow data at 30 gage locations within the basin. The model simulations were validated outside the calibration period at 30 gage locations (Figure 1 and 2). The region had sufficiently detailed data during early part of the

simulation period [*Alsdorf et al.*, 2016; *L'vovich*, 1979]. Satellite measurements, sparse ground-based measurements and reanalysis products provide the most reliable climate data for the reminder of the simulation period [*Alsdorf et al.*, 2016; *Munzimi et al.*, 2014].

We only used one hydrological model. However, recent research suggest that projection uncertainties dominate compared to other sources of uncertainties (e.g. model structure and parameters) in hydrologic projections [*Maurer and Pierce*, 2014]. Suggested references have also been used to improve the discussion section.

2. Observational uncertainty. The first sentence of the paper states that efforts to understand the impacts of climate change in the Congo Basin are hindered by data availability. However, the authors do not make clear in the paper how they have overcome this, or the extent to which their findings are valid given observational uncertainty. They use an observational dataset from Sheffield et al. and (I think) use this for (a) bias correction (b) temporal downscaling to daily data and (c) sub-selecting climate models based on their ability to represent the region. Therefore, the observations might have a very important influence on their findings.

It is generally accepted (e.g. Washington et al. 2013) that availability of observed climate data in this region is a huge problem which might prevent subselection of models or bias correction. How can we say which model is more valid when there are basic questions remaining about the quantity of precipitation or where the precipitation maximum occurs? What dataset should we use assess and correct biases when there are large differences between the observational datasets used? The Sheffield et al. dataset does sound like an impressive undertaking and an important initiative but in the absence of rain gauge records it is difficult to validate it for this region, so it it still just one estimate of the observed state. I think the authors should, as a minimum, comment on the extent to which this dataset is reliable for the region and the extent to which their results might be influenced by observational uncertainties. They could also repeat their correction analyses with an alternative observational estimate and see whether this influences their results.

I am particularly concerned about the temporal downscaling to daily data, and think the authors should comment on the extent to which this is reliable, give that our understanding of day-to-day variability in precipitation/organisation of convection/meso-scale convective systems in this region is just beginning.

As mentioned in the earlier response, the region had sufficiently detailed ground-based observational data (e.g. precipitation and river flows) during early part of the simulation period. Satellite-based and limited ground-based observations are used to develop historical precipitation data used in our study. The dataset is developed and evaluated using multiple observation-based and reanalysis products (TRMM, GPCP, CRU, NCEP-NCAR and the second Global Soil Wetness Project) [*Sheffield et al.*, 2006]. During the development of the this dataset, the NCEP-NCAR precipitation product was examined and corrected for total monthly precipitation and monthly rain day statistics using CRU, GPCP and a 15-year gage-based dataset. The downscaling process also took into consideration the spatial consistency.

The lack of observational data (both precipitation and river flow) during the late 1970s and 1980s is a constrain and a limitation in this region. We have discussed these limitations and constraints in the manuscript.

**Specific comments**

p. 4. Line 50. "require detailed information" – perhaps rephrase. If the information is not credible then details could be counterproductive. So I think better to say "would benefit from detailed information"

**Revised**

p. 4. Line 54. "predictive" and "forecast" – suggest change to "project" since we cannot forecast or predict on these timescales, only "project" what if under certain emissions scenarios. Suggest changing throughout.

**Revised**

p. 9. Line 162 – I am not sure what is meant by "medium mitigation" for RCP4.5.

The phrase is revised as "mid-range mitigation emission". The paragraph is revised to make clear the two emission scenarios.

p. 11. Line 190 – does this refer to bias corrected precip? If so I think this should be highlighted. Does it mean much if bias corrected precip fits with observations? Since it has been corrected using these observations?

The GCM-simulated annual precipitation refers to the bias-corrected values. The paragraph has been revised to make this point clear. We used a statistical bias-correction method to correct monthly GCM-outputs [*Li et al.*, 2010]. The procedure is described in the methods section and in the SI.

p. 11. Line 193. "The modeled inter-annual variability among the climate models (vertical bars in Figure 2) lies within the range of the observed variability" – Looking at the figure I am not sure this is strictly true. I can see a few examples where the error bar for the models is larger than for the observed.

In Figure 2, we show, among the 25 GCM outputs, the largest (red vertical bars) and smallest (blue vertical bars) values. As noted, there are some GCM outputs that show larger variabilities.

Figure 2 – please clarify the meaning of the modelled error bar. The caption states that it is based on the minimum and maximum range of interannual variability from the models. Is there anything to show the range of mean/climatological values for the models? And how does this compare? (similar comment for Figure 3b)

The vertical bars show mean  $\pm$  one standard deviation of GCM-simulated annual precipitation during the historical period (1950-2005). The red bars denote the largest variability (highest value of std. dev.) within the 25 GCM outputs, and the blue bars denote the smallest. The horizontal bars shows the mean  $\pm$  one standard deviation for the observed precipitation during the same historical period. Each black point indicates the mean annual precipitation within the drainage areas at gage locations showed in Figure 1. The text and figure captions have been revised.

p. 13. Line 239 – why is the IQR used? What is the full range?

We chose to present the inter quartile range to highlight where the bulk of the projection values lie. The full range of precipitation projections varies between a 3% decrease to a 6% increase in the near-term (2016-2035). The mid-term (2046-2065) changes are -5% to 7.6% for RCP4.5 and -6% to 9% for RCP8.5, respectively.

Figure 7 – nice figure. Is there a way to make historical plot clearer?

Figure has been revised.

p. 16 Line 293 – I am not sure that MMEs reduce uncertainty. It's more that they help explore and reveal uncertainty.

We have revised as per the reviewers suggestions.

p. 16 Line 304 – I think it is overstating it to say that these models reliably simulate regional climate. We don't have good enough observations of the regional climate to judge this. And, in any region, subselecting models is usually about taking the ones which most reliably simulate regional climate, rather than being confident that they are good enough.

We evaluated the annual, seasonal and monthly simulations of precipitation and temperature by the 25 GCM in the Central African region in a separate manuscript [*Aloysius et al.*, 2016]. Previous works in the Central Africa region highlight that model skill in simulating precipitation are partly dependent on how they replicate teleconnections with sea-surface temperature (SST) departures, particularly in the North Atlantic and Indian Ocean sectors (e.g. *Balas et al.* [2007]; *Dezfuli and Nicholson* [2013]; *Hirst and Hastenrath* [1983]; *Suzuki* [2011]). Our companion manuscript [*Aloysius et al.*, 2016] explored the linkages between precipitation and SST departures, and identified a subset of GCMs that simulate precipitation well.

We revised the discussion taking into consideration i) the above points and ii) the reviewer's comments.

p. 17 Line 326. This is quite an odd paragraph which starts of talking about implications of findings (from MM and SM?) and then finishes by saying we can

reduce the range of projections from MMEs. Perhaps this should be reconsidered to suggest more nuanced conclusions about the implications of the findings which incorporate un- certainty? It is also not clear, when the author is discussing the potential to constrain model ensembles using knowledge of mechanisms that moderate the regional climate system, whether this is something they feel they have already done, or something that needs to be done. If the former, I'd suggest that their subselection procedure warrants further attention in the paper.

**This section has been revised.**

Figure 8 – quite a lot of information here. Could it be distilled to extract the main message?

**Figure is revised.**

p. 19 Line 363. "with sufficient details". I disagree. Providing details to planners may be misleading if there is too much uncertainty to give details. Better to help planners understand the uncertainty?

This section has been revised to highlight projection uncertainties between GCMs and emission scenarios.

p. 20 Line 377. "The analyses presented in our work increase the degree of confidence in using the results for policy and management." This is unsubstantiated.

**References**

- Aloysius, N., J. Sheffield, J. E. Saiers, H. Li, and E. F. Wood (2016), Evaluation of historical and future simulations of precipitation and temperature in Central Africa from CMIP5 climate models, *Journal of Geophysical Research - Atmospheres*, 121(1), 130-152, doi: 10.1002/2015JD023656.
- Alsdorf, D., E. Beighley, A. Laraque, H. Lee, R. Tshimanga, F. O'Loughlin, G. Mahé, B. Dinga, G. Moukandi, and R. G. M. Spencer (2016), Opportunities for hydrologic research in the Congo Basin, *Reviews of Geophysics*, 54(2), 378-409, doi: 10.1002/2016RG000517.
- Balas, N., S. E. Nicholson, and D. Klotter (2007), The relationship of rainfall variability in West Central Africa to sea-surface temperature fluctuations, *International Journal of Climatology*, *27*(10), 1335-1349, doi: 10.1002/joc.1456.
- Dezfuli, A. K., and S. E. Nicholson (2013), The Relationship of Rainfall Variability in Western Equatorial Africa to the Tropical Oceans and Atmospheric Circulation. Part II: The Boreal Autumn, *Journal of Climate*, *26*(1), 66-84, doi: 10.1175/JCLI-D-11-00686.1.
- Hirst, A. C., and S. Hastenrath (1983), Diagnostics of hydrometeorological anomalies in the Zaire (Congo) basin, *Quarterly Journal of the Royal Meteorological Society*, 109(462), 881-892, doi: 10.1002/qj.49710946213.
- L'vovich, M. I. (1979), *World water resources and their future*, American Geophysical Union.
- Li , H., J. Sheffield, and E. F. Wood (2010), Bias correction of monthly precipitation and temperature fields from Intergovernmental Panel on Climate Change AR4 models using equidistant quantile matching, *Journal of Geophysical Research -Atmospheres*, *115*(D10), D10101, doi: 10.1029/2009jd012882.
- Maurer, E. P., and D. W. Pierce (2014), Bias correction can modify climate model simulated precipitation changes without adverse effect on the ensemble mean, *Hydrol. Earth Syst. Sci.*, *18*(3), 915-925, doi: 10.5194/hess-18-915-2014.
- Munzimi, Y. A., M. C. Hansen, B. Adusei, and G. B. Senay (2014), Characterizing Congo Basin Rainfall and Climate Using Tropical Rainfall Measuring Mission (TRMM) Satellite Data and Limited Rain Gauge Ground Observations, *Journal of Applied Meteorology and Climatology*, *54*(3), 541-555, doi: 10.1175/JAMC-D-14-0052.1.
  Sheffield, J., G. Goteti, and E. F. Wood (2006), Development of a 50-year highresolution global dataset of meteorological forcings for land surface modeling, *Journal of Climate*, *19*(13), 3088-3111, doi: 10.1175/JCLI3790.1.

Suzuki, T. (2011), Seasonal variation of the ITCZ and its characteristics over central Africa, *Theoretical and Applied Climatology*, *103*(1), 39-60, doi: 10.1007/s00704-010-0276-9.

---

## Referee Report (RR2)

[revised manuscript text omitted]

1-3% increase in precipitation (20mm – 45mm) and a 4-9% increase in total runoff (15mm-34mm) within the CRB in the near-term (2016-2035) relative to reference period (1985-2005) for MM and SM, respectively. In the mid-term (2036-2065), on the other hand, projections are GCM and emission-scenario dependent, with the high emission

RCP8.5 scenario showing the highest increases in precipitation (2-5%, 30mm – 70mm)

and runoff (7-14%, 25mm – 50mm) for MM and SM, respectively. Modeled projections also exhibit substantial    inter-model variability with projected changes varying between

-3% and 9% for precipitation and -12% and 24% for total runoff from the CRB between the mitigation and business-as-usual greenhouse gas emission scenarios. Regionally, both

MM and SM project decreasing precipitation and runoff in parts of southern and northern headwater regions of the CRB.

[revised manuscript text omitted]
 | 0.04 (17.7) | -4.1 (19.4) | 2.5 (9.3)* | 1.1 (8.5) | 5.7 (7.2)* | 3.3 (7.7) | 5.6 (11.2)* | 3.1 (12.6) |

---

## Author Response (AR2)

**Response to Reviewer Comments (Aloysius and Saiers)**

Our manuscript has benefitted from the comments and suggestions of the two reviewers. We have revised and rewritten sections of the manuscript. The comments of the reviewers are provided below in *italicized* font; our responses are in normal text. The track changes enabled version of the manuscript highlights revisions made to the manuscript.

Reviewer #1:

1) *Details are missing from the SWAT model development: version and revision of SWAT used and a table of parameters that were changed in calibration of the model (can be in SI).*

We have updated the Methods section and Supplemental Information as per the reviewer's suggestions. A table with adjusted parameters during calibration is also included in the SI (SI Table S5). We used version 488 of the model source code. We have also revised the Supplemental Figure S3 to show simulated v. observed hydrographs of all 30 gages used in calibration and validation.

2) *Was the CO2 level changed in SWAT? If so, what was it changed to; if not, why was it not changed?*

Due to the lack of information on the effect of $CO_2$ on the 16 land cover classes simulated, the ambient $CO_2$ concentration was maintained at 330 ppm throughout the simulation period. A recent study also suggest that hydrologic partitioning in tropical rainforest catchments is largely unaffected by elevated $CO_2$ (line 230-235).

The methods section (section 2.4) is updated accordingly.

3) *In general the figure captions need to be more detailed so that they can be stand-alone. For example, Figure 3B you should clarify if the GCM-simulated climate is the statistically downscaled and bias corrected data (similar comment for Figure S2).*

Captions of Figure 3B (lines 25-35 in "02_Aloysius_2016_figures.docx") and SI Figure S2 (SI lines 104-109) have been updated as per the reviewer's suggestion.

4) *In lines 195 and 221 you refer to the climate projection simulations going to 2099, but it does not appear this was the case so this number should be changed.*

The model simulation period is 1950-2065 (lines 198 and 230). We have updated the text accordingly.

Reviewer #2:

*1) As in the previous round of reviews, I would like to highlight that I welcome the contribution of this substantial scientific effort to investigate climate change in the Congo Basin, since it is such an important and understudied region. However, I do not believe that the authors have sufficiently addressed my previous comments, and therefore would suggest further major revisions. I think the analysis could be useful, but that the paper requires a substantial re-write to ensure that the results and their implications are represented accurately. Perhaps I can explain my points more clearly to help them to be addressed more systematically. They still center around (1) model uncertainty, and (2) observational uncertainty.*

Inadequately addressing model and observational uncertainty was a significant weakness of the manuscript. To address this issue, we have revised the methods section (lines 216-222) and added a new section "3.4 Sources of Uncertainty". This section covers both model and observational uncertainties as suggested by the reviewer. The observational uncertainties include declining gage-based precipitation observations, particularly in the equatorial region (lines 414-427) and observed runoff data (lines 428-435). We agree with the reviewer that gage-based precipitation coverage is very limited after 1990s. We have quantified the number of gage-based precipitation data that contributed to the development of historical climate observations used in the hydrological model and for statistical bias-correction. Number of gages remained at about 160 during 1950-1980 and had substantially reduced since then (Supplemental Figure S5 and S6). However, satellite-based precipitation data has been used since the 80s. We believe these multiple sources (gage and satellite-based and reanalysis) adequately capture spatial and temporal variability of precipitation in the Congo region. Additional references supporting our claim are mentioned in the main text (line 423-427).

For runoff, we used all the available gages (n = 30) during the study period. The locations of these gages adequately capture climatic, land cover and topographic variability (lines 428-435 and supplemental Figure S3).

For future projections, the largest source of uncertainty is the GCM outputs. We have discussed the potential sources in section 3.4 (lines 436-496). Suggested literature by the reviewer has been incorporated. Figures 6-8 have been revised to highlight model uncertainties. The variability in modelled runoff are presented in Table 3, which show the multi-model mean, standard deviations and fraction of model projections with increasing runoff, by region and by season.

We have revised the abstract to highlight the need to evaluate uncertainties in climate change assessment (lines 32-35)

Specific comments:

*1) First sentence in the abstract: A side point, but is this really true? Compared to other regions there is relatively little research for the Congo Basin.*

We have re-characterized the effects of climate change on CRB water resources as understudied (line 14).

*2) I do not think you can say "elucidate" since we cannot know what the variability in runoff will be I the near and mid 21s century yet.*

Changed to "explore"

*3) All models? Some models? Most models? The mean of the models? Are there any that show decrease?*

The abstract has been revised to include the mean and the range of projections (lines 20-23).

4) Here I think it would be more useful to embed the information about uncertainty into the information about projections. It is not easy to infer this from what is written, but it might be something like:

We revised the abstract according to this suggestion (lines 32-35).

*5) Unclear why this has been changed from "model consensus" to "consensus". Arguably it's important that it is just a model based consensus*

We have removed this phrase in the revised abstract.

*6) I think might and would are important here to tone down so that it is not implying that we know what will happen*

Abstract has been revised as suggested.

*7) This is a bit of a strange statement. Of course the risk attitudes of planners will influence their approach, but perhaps the scientific results can be used to imply the extent to which there is credible information for planning. Personally I think it would be OK to recommend using an approach which takes into account a range of futures, since there are so many uncertainties associated with climate information in the Congo Basin.*

The phrase "risk attitude" no longer appears in the abstract.

*8) Can you instead comment on the challenge of finding a solution that is robust to the range of projected changes?*

Addressed in section 4 (lines 501-524).

*9) Why? This is unsubstantiated and doesn't really make sense. What does it mean to say that the analyses increase the degree of confidence in using the results (since the results are based on the analysis). Suggest removing.*

Removed as suggested.

*10) In general I think it would be important to revise the text of the paper in line with these kind of edits. i.e. if referring to model results, it is important to say that they are model results, and if making inferences, to use "might" or "could" rather than "will". The use of "predict" has been changed in several cases to "project", as advised, but this has not been done consistently. I would suggest removing all references to "predict" and "forecast" when referring to long term climate projections.*

These suggestions have been adopted in the revised text.

*11) "The results presented here show a range of runoff projections under two broad assumptions, that i) individual GCM biases will cancel and that MM mean projections are more likely correct and ii) selection of GCMs that simulate mechanisms reliably is a better option for climate change assessment." However, I do not think these assumptions can be used unless they are justified. I think that both (i) and (ii) are highly questionable. There is quite a bit of work (cited in my previous round of comments) which critiques the idea of using the mean for future projections. And, on point (ii) I do agree that selecting GCMs which simulate mechanisms would be helpful, but what is meant by "mechanisms"? My understanding is that the subselection here is based on the author's previous JGR-A publication, in which models are selected based on observations of key variables like temperature and precipitation, rather than the modelled "mechanisms". Sub-selecting models using observational constraints is an approach which is often adopted, but is also questioned, particularly for regions with such high observational uncertainty. Therefore, I think that if these assumptions are to be stated they must be justified and discussed in a balanced manner which acknowledges for the readers of HESS that many climate scientists would dispute with these assumptions. Alternatively, a better approach would be to re-write the results to focus more on the range of modelled outcomes.*

These assumptions have been revised and rewritten. Section 3.4 and 4 addresses the projection uncertainties. We have provided reasoning for selection of the subset of models (lines 461-472).

*12) It would be interesting to quantify the amount of data available and comment on what is meant by "sufficient". I agree that there is more data available during the early part of the period (when I believe CRU is the only one of the datasets used to modify NCEP reanalysis – based on Sheffield et al. 2006, Table 1), however, based on Washington et al. 2013 Figure 1, there are still max 60 gauges*

*contributing to CRU during this time for the whole Congo Basin, which is very few stations compared to the density of stations over e.g. UK or USA.*

We have added two figures in the Supplemental Information (Figures S5 and S6) and discussed the observational data availability in section 3.4 (lines 406-427).

*13) I cannot see where this discussion has been added? I think it should be discussed in the methods section. Also in results – p. 11, line 219 there is a statement about bias corrected precip from model being in agreement with observations. Wouldn't this be expected if the observations have been used to correct the model output?*

The observational uncertainties are discussed in section 2.4 (lines 216-220) and section 4 (lines 408-427). Results comparing bias-corrected and observed precipitation have been revised (lines 249-251).

[revised manuscript text omitted]

1-3% increase in precipitation (20mm – 45mm) and a 4-9% increase in total runoff (15mm-34mm) within the CRB in the near-term (2016-2035) relative to reference period (1985-2005) for MM and SM, respectively. In the mid-term (2036-2065), on the other hand, projections are GCM and emission-scenario dependent, with the high emission

RCP8.5 scenario showing the highest increases in precipitation (2-5%, 30mm – 70mm)

and runoff (7-14%, 25mm – 50mm) for MM and SM, respectively. Modeled projections also exhibit substantial    inter-model variability with projected changes varying between

-3% and 9% for precipitation and -12% and 24% for total runoff from the CRB between the mitigation and business-as-usual greenhouse gas emission scenarios. Regionally, both

MM and SM project decreasing precipitation and runoff in parts of southern and northern headwater regions of the CRB.

[revised manuscript text omitted]
 | 0.04 (17.7) | -4.1 (19.4) | 2.5 (9.3)* | 1.1 (8.5) | 5.7 (7.2)* | 3.3 (7.7) | 5.6 (11.2)* | 3.1 (12.6) |

---

## Author Response (AR3)

**Response to Reviewer Comments (Aloysius and Saiers)**

Our manuscript has benefitted from the comments and suggestions of the three reviewers. We have updated the manuscript based on comments and suggestions by the third reviewer. The comments of the reviewer is provided below in *italicized* font; our responses are in normal text. The track changes enabled version of the manuscript highlights revisions made to the manuscript.

Reviewer #3:

1) *It's a little strange to have results in your introduction. My preference is to remove this section, but I'm not completely opposed to it being left in.*

We have removed the last paragraph in the Introduction (removed line 93-109).

2) *Please actually use the model's name - SWAT*

We have revised the text as per reviewer's suggestion (Line 142).

3) *This is rather strange to use the earliest part of your simulation period to calibrate the model. Climate and likely land-use have changed since then so I'm wondering if the model is actually representative of the current time period? If this is a limitation of available data, you should describe this.*

We used the time period 1950-1957 due to the limitation of available observed data that captures climatic, land cover and topographic variability within the river basin. However, the results presented in Figure 3A use all the observed data available during the historical simulation period (1950-2008). All streamflow gages had at least 10 years records (line 180). We have revised the methods section as suggested by the reviewer (lines 183-184). Human influenced land cover changes are minimal in the region (lines 123-125).

4) *Replace fields with data here and in subsequent paragraphs.*

We have revised the text as suggested by the reviewer.

5) *Good to reference tables 1 and 2 in this section.*

We have referenced Tables 1 text (lines 237), and Table 2 is referenced in the results section.

6) *It's not clear how you calculated AF. Was this just 70% of total flow, or did you use a threshold value to define flooding, then subtract the flooding out? Please define further.*

We applied a base flow separation method described in Nathan and McMahon 1990 to remove surface runoff events associated with flood events. The remainder is the accessible flow. We have updated the methods sections (lines 241-243).

7) *This figure (Figure 1) would benefit from a location inset map, but it isn't necessary. Also, you clearly have a topographic map underneath the "rainforest, wetlands, lakes" layer, but it actually makes it difficult to view. What are the areas that are not lakes, rainforest, or wetlands? Hard to tell from this map.*

We have updated Figure 1 as suggested by the reviewer. A location map is included. The land cover classes are grouped into four categories as i) rainforest, ii) lakes, iii) wetlands and iv) all other vegetation. Relevant reference is included in the figure caption.

8) *These figures (Figure 3) are nice for overall view, but difficult to read. Would help to have a table associated with this (even if in SI) with NSE for each station.*

We have added the NSE values for each gage station shown in Figure 1 and 3 in Supplemental Information Table S2.

9) *Figure 9 – change to "blue and red bars (RCP 4.5, RCP 8.5, respectively)"*

Figure caption has been modified as suggested by the reviewer.

[revised manuscript text omitted]